# Considering socio-political framings when analyzing coastal climate change effects can prevent maldevelopment on small islands

C. Gabriel David 1,7✉, Arne Hennig2, Beate M. W. Ratter 2,3, Volker Roeber4,5, Zahid6 & Torsten Schlurmann 1

Adapting to climate change and sea level rise is challenging on small islands. False adaptation can lead to adverse impacts on natural and societal dynamics. Therefore, an interdisciplinary perspective on the interaction of natural dynamics, societal demands, and political decisions is crucial. In this sense, this study scrutinizes coastal processes and socio-political dimensions of erosion on the reef island Fuvahmulah, the Maldives. The national government and Fuvahmulah's population have an opposed perception and attribution of the drivers and processes behind Fuvahmulah's most pressing coastal issue – coastal erosion. To review these perceptions, natural dynamics are recreated with process-based methods and discussed regarding present and projected marine pressures. Population surveys and interviews with actors in coastal development complement the physical insights into erosion on Fuvahmulah and describe the socio-political dimension of climate change adaptation on small islands. This interdisciplinary approach demonstrates how small-islands' adaptive capacities are typically impaired and disclose the potential of local knowledge to overcome maldevelopment.

[1] Ludwig-Franzius-Institute for Hydraulics, Estuarine and Coastal Engineering, Leibniz Universität Hannover, Nienburger Straße 4, 30167 Hanover, Germany. [2] Institute for Geography, Department for Earth System Sciences, Universität Hamburg, Bundesstraße 55, 20146 Hamburg, Germany. [3] Helmholtz-Zentrum hereon GmbH, Department of Human Dimensions in Coastal Areas, Max-Planck-Straße 1, 21502 Geesthacht, Germany. [4] E2S UPPA, Chair HPC-Waves, SIAME, Université de Pau et des Pays de l'Adour, Allée du parc Montaury, Anglet, France. [5] University of Hawai'i at Mānoa, Department of Oceanography, 1000 Pope Road, Honolulu, HI 96822, USA. [6] Formerly Maldives Meteorological Service, Hulhule 22000, Maldives. [7] Present address: Leichtweiß-Institute for Hydraulic Engineering and Water Resources, Division Hydromechanics, Coastal and Ocean Engineering, Technische Universität Braunschweig, Beethovenstr. 51a, 38106 Braunschweig, Germany. ✉email: g.david@tu-braunschweig.de

There is high confidence and evidence that climate-change induced sea-level rise increases the risk of small islands to suffer from ocean-borne impacts associated with high tides and swells[1–3]. Small low-lying islands and atolls are especially susceptible to the impacts of sea-level rise, as there is often no higher ground available for the local population to retreat to. In the upcoming decades, stronger and more frequent seaborne hazards will follow global heating, making small islands – in extreme situations – potentially uninhabitable[2,4]. Seaborne hazards on low-lying islands are typically rising sea levels, severe storm surges and swell waves. Likewise, these environmental pressures are constituent parts of the natural dynamics: Coastal environments react to hydrodynamic pressures with a morphodynamic response. Thus, wave-induced currents on the reef platform are an important driver of sediment transport[5,6] and the interplay of waves, sea level, reef platform, and sediment production are major elements of reef island genesis and evolution[5,7–9]: For example, when looking at the Maldives, monsoonal wind-wave events have an impact on locally characteristic island morphology, but, in general, distant-source swell waves are dominant in the regional wave climate[5,8,10]. Swell waves are characterized by a long wave period and play a key role in the island development on reef platforms[11,12]. Together with constant sediment provision by the coral reef[13], the wave-induced sediment transport allows reef islands to mitigate erosion[14] and accrete vertically in response to sea level rise[11,12]. Coral reefs are important coastal protection assets[15], because the reef dissipates most of the incoming wave energy[16]. There is evidence that reefs have grown and can continuously provide protection under rising sea levels[11,17] if they remain healthy. As a consequence, healthy coral reefs are a very important – if not the most important – factor of the island's resilience to withstand sea level rise and marine extreme events. However, global warming and ocean acidification threatens the health of coral reefs[18] and with that also the important ecosystem services coral reefs provide.

For small islands, coastal protection is an important ecosystem service. Owing to their geography, small islands' populations are more exposed to marine hazards and more vulnerable than mainland communities. In addition, marine resources deliver important ecosystem services for island communities, because they provide food and income for the island's inhabitants[19]. However, protection from increasing risk of seaborne hazards as well as connection and exploitation of marine resources require concerted development of coastal areas, followed by construction and maintenance of infrastructure. To date, coastal structures and anthropogenic interventions have often resulted in the proliferation of engineering-type hard-coastal protection systems, defending coastal areas up to a certain level of safety from coastal hazards. These interventions are man-made disruptions of the coastal system, often undermining vital ecosystem services and interfering with the morphodynamic interchange on beaches[20,21]. Nevertheless, such structures are implemented widely[22], even though they interrupt the littoral currents or deteriorate coastal environments to large extents. Anthropogenic interventions in natural dynamics put further pressures on the environment and are driven by political decision-making processes and social interests[23].

In their first assessment report, the Intergovernmental Panel on Climate Change (IPCC) names three general response strategies to adapt to increased coastal risk and sea level rise[24]: protection, accommodation and retreat. In the meantime, further alternatives have emerged and in 2019, the Special Report Ocean and Cryosphere in a Changing Climate (SROCC) of the IPCC complements the response strategies by supporting a broader range of options[3]: protection, accommodation, advance, retreat, ecosystem-based adaptation, or no response. However, especially small island countries are often confronted with a number of issues that further complicate finding sustainable solutions to infrastructure demands[25,26]: on the one hand, decisions are regularly made from afar by the central government without addressing or integrating local knowledge. On the other hand, island states often lack expertize as well as human and financial resources to allow thorough planning and proper implementation[27]. Thus, development decisions are embedded in political structures on multiple levels of governance, determining how decisions are made and what actors are involved in the process[25,28,29].

Against this background, the present study illustrates the problem of interlinked natural and societal dynamics by presenting the case study and example of erosion on the reef island Fuvahmulah, the Maldives. The study site is subject to typical aspects of increased disaster risk on small islands in the face of climate change: Fuvahmulah is located in the center of the Indian Ocean, especially exposed to seaborne hazards, such as sea level rise or wave events. In the Indian Ocean, sea level rise is accelerating and the region is prone to experience extreme climate and weather events more frequently[30,31]. The Maldives government addresses these challenges mainly with hard-engineered protection[20,22,32]. The resulting fortification of the coast is often an anthropogenic disturbance in the dynamic natural reef system, undermining vital ecosystem services and leading to increased vulnerability to suffer from adverse climate change effects[20,22,32]. In that context, this study investigates maladaptive developments and explicates their consequences to facilitate developing adequate and sustainable adaptation strategies – especially in the face of increasing risks due to climate change. This is achieved by an interdisciplinary approach, using methods and results from former studies on natural drivers and responses on small islands[5] and socio-political framing of climate change adaptation in the Maldives[33]. This study synthesized these results with the Drivers, Pressures, State, Impact and Response model of intervention (DPSIR) and focuses on root causes behind erosion on Fuvahmulah. Following the DPSIR's chain of causal links, the framework facilitates identifying the oceanic-climatic and socio-political pressures leading to state changes of sensitive environments. Establishing connections between the DPSIR's constituents allows for upscaling the lessons learned from the case of Fuvahmulah to other examples of top-down implemented coastal infrastructure in sensitive reef environments, because the island's current adaptation pathway is an epitome of maldevelopment on small islands: the attempt to improve the socio-economic situation requires to equip the island with coastal infrastructure, which most often interferes with the natural dynamics. However, instead of addressing these adverse effects adequately, the socio-political preference and constant top-down prescription of generic hard-engineered measures is prone to lead to recurrent maladaptive actions, cascading adverse effects along the coast. Together, the socio-political framing and the top-down approach to implement infrastructure projects are anthropogenic pressures, which gradually diminish the island's natural capacity to adjust to ocean climate-related pressures. Numerous examples exist elsewhere in the Maldives[20,22,34] and in other Small Island Developing States (SIDS)[25,27] with similar symptoms of maldevelopment. With an interdisciplinary perspective of coastal scientists and human geographers, this study (1) introduces the concept of maldevelopment, describing the socio-political framing leading to repeated maladaptive actions, and (2) demonstrates how local knowledge of the drivers and processes in sensitive small island environments supports low-regret development to mitigate adverse impacts on natural and societal dynamics.

## Results

**Maldevelopment.** The IPCC defines action or inaction as maladaptation if it "may lead to increased risk of adverse climate-

related outcomes, increased vulnerability to climate change, or diminished welfare, now or in the future" (WGII AR5[35], Glossary, page 1769). In coastal environments, maladaptation aims at reducing adverse effects to the coastal community now, but has negative implications on future response options under different climate change scenarios[36]. For example, the wide-spread anthropogenic coastal fortification in the Maldives has lead to irreversible changes to the natural coastal system, so that the only remaining future response to rising sea levels and extreme events is further armoring of the reef islands' coasts[22]. These "anthropogenic tipping points"[22] are mainly governed by negative socio-political dynamics, which have grown historically in the Maldivian context within the last decades[26,37]. Despite the success of recent decentralization efforts[38], national development policies aim at improving the socio-economic situation by fostering regional development centers (see the National Population Consolidation Program). These regional development centers are equipped with public and infrastructure facilities, and thus are considered safer and are higher populated[26,37]. However, these development efforts by the national government also encourage top-down implemented infrastructure projects[33]. In a top-down approach to implement infrastructure projects, decisions on these projects are generally made on a national-level, prescribing the implementation of these projects to the local level. Under this current policy pathway, anthropogenic disturbances in the natural dynamics will remain to govern coastal changes on the Maldives considerably in the future – besides climate change related pressures[26]. While recent studies found deficient coastal adaptation policies and policy compliance in the Maldives[39,40], the results of this study illustrate the impact of such deficits in practice and reveal the framing of recurrent and ongoing maladaptation, which is structurally embedded within the socio-political system. In contrast to maladaptation, this "maldevelopment" is not an inadequate adaptation action leading to climate change related risks, but the socio-political driver behind repetitive maladaptation. Besides the maladaptive impact on the natural system, maldevelopment emphasizes the socio-political aspects of recurrent maladaptive actions: on the one hand, the societal factors, being the local population's perception of and their attachment to place, their interest in economic development, and environmental protection. On the other hand, the political aspect of responsible authorities, aiming to balance the need for socio-economic development with environmental protection and issues of sustainability by means of decisions and policies. In this sense, maldevelopment is characterized by decisions or policies which are in constant, repeatedly or deliberate favor of maladaptive actions and trade-offs towards future climate change related risk – as epitomized by current coastal adaptation efforts on the reef island Fuvahmulah.

**Study site.** Fuvahmulah is an island in the south of the Maldives, located approximately 30 km south of the equator (latitude: −0.30°, longitude: 73.43°, see Fig. 1). Unlike most other inhabited islands of the Maldives, Fuvahmulah is not part of a ring-shaped atoll and lacks the distinct protection features of oceanward islands[8]. Fuvahmulah consists of only one main island and its fringing reef. The main island has a size of about 4.4 km by 1.0 km and a coastline of ~11 km length. The island is surrounded by ridges, which are located landward behind the beaches and predominantly vegetated by a fringing palm forest. The island's ridges reach a height of up to about 4 m above mean sea level. Together with the reef, this natural barrier mostly protects the island from storm surges, swell waves and helps mitigating overwash. Behind these natural coastal protection systems, Fuvahmulah's inland is at mean sea level and hosts two freshwater lakes.

Among islands of the Maldives, the landmark of Fuvahmulah is its sandy beach Thoondu, a coastal spit situated at the north of the island. As wave direction changes with the seasons, Thoondu responds to these natural dynamics[5]: it seasonally adjusts its landform shape, alters its sediment volume and regularly meanders from the northern part of the headland in wet season towards the north-east of the island as well as to the northwestern Geiymiskih beach in the dry season. The constant change of Thoondu is well known by locals and contributes to its uniqueness. As consequence, in 2020, the United Nations Educational, Scientific and Cultural Organization (UNESCO) designated Fuvahmulah as a Biosphere reserve.

The closest neighboring island is Hulhumeedhoo (part of Addu Atoll) located about 40 km southwest of Fuvahmulah. The 8510 inhabitants[41] of Fuvahmulah are exceptionally isolated – even when considering all 26 atolls in the Maldives, stretching over about 870 km length (north to south) and about 130 km width (west to east). The isolated location and with the reef and vegetated coastal ridges serving as the only natural coastal protection, Fuvahmulah is particularly susceptible to environmental forcing. Fuvahmulah's location close to the equator makes it less likely to experience cyclones[10,42] but the island is considered to be highly exposed to monsoon winds[43] and associated wind-waves, as well as to distant-source swells[5,8,10,26].

The isolated position makes Fuvahmulah dependent on transportation infrastructure, providing a safe connectivity, an access to the outside world. The transport infrastructure on the island endows the local economy and supply – especially through the shipping of goods. However, until 2002, Fuvahmulah was only accessible by small fishing boats ("dhonis"), which had to navigate through the breaking waves over the reef plateau. The "dhonis" acted as feeders, transporting people, goods, and cargo from Fuvahmulah to larger ships that were waiting offshore. In this procedure, numerous boats capsized and people died. For decades, the people of Fuvahmulah have been requesting the construction of a safe port. The new seaport was constructed on the southeast of the island opening in 2002. The seaport was followed by the construction of an airport for domestic flights in the southwest of the island that opened in 2011.

**Measured coastline erosion on the east side.** Ever since the port has been built, people on the island observe and report severe erosion on the east-side of Fuvahmulah[44]. The Digital Elevation Models (DEMs), derived from aerial imagery, prove this observation and quantify the observed erosion rates and its spatial extent along the east coast. Between 2017 and 2019, the cross-sections along the east side show distinct differences in morphologic changes (Fig. 2): while the northern area close to Thoondu (Fig. 2a) shows a slight increase in beach volume, the other transects depict ongoing erosion (Fig. 2b–d). The increase in sediment volume in the north at Thoondu beach is due to the typical dynamic behavior of the northern beaches in response to varying seasonal wave conditions: during dry season, Thoondu beach moves to the northeastern coast[5]. The cross-section at the end of Thoondu beach (Fig. 2a) shows the calcareous bulk material forming the beach face. When moving further south, the coast of the central east side has a steeper profile than the northern beaches and consists of gravel to cobblestone material (Fig. 2b) or larger boulders. The two southernmost transects near the seaport have steep edges, where the mainland meets the reef flat (Fig. 2c, d). These transects are located about 700 m (Fig. 2c) and 190 m (Fig. 2d) in the north-west of the seaport's entrance. At these locations, the subsurface insular bedrock becomes visible. The erosion process is already noticeable from field observations and has also been widely recognized and reported by locals.

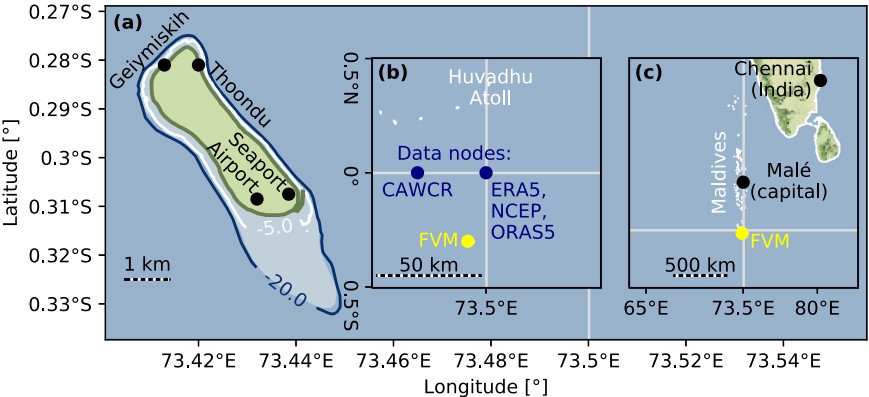

**Fig. 1 Map of Fuvahmulah (FVM). a** Shows the island, the seaport, and airport location, as well as the beaches Geiymiskih and Thoondu. Contour lines show the fringing reef depth. **b** Location of Fuvahmulah and data nodes of the hindcast and reanalysis model, from which the wave climate was derived. **c** Location of Fuvahmulah within the Maldives and the Indian Ocean. Maps were generated with the Python module cartopy, using open-access © OpenStreetMap contributors data (open-access Geofabrik download server data and Stamen map tiles), as well as data from the field measurements (Fuvahmulah's coastline and reef bathymetry).

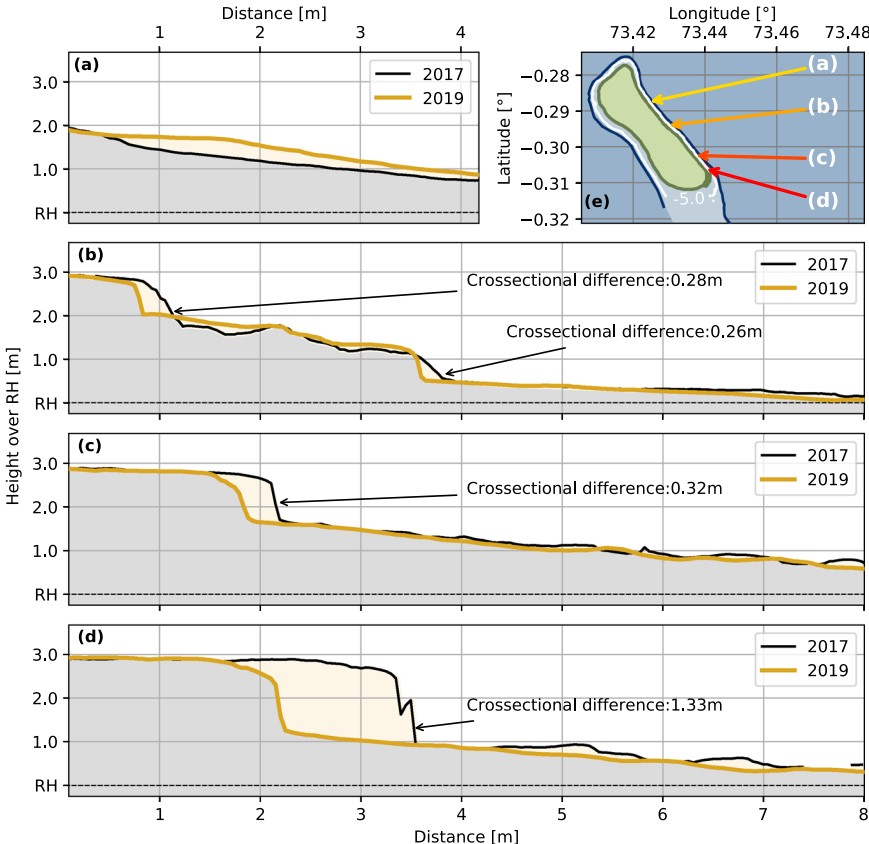

**Fig. 2 Erosion along the east coast of Fuvahmulah.** Cross-sections of four locations, showing typical coastal profiles along the east side of Fuvahmulah. The profiles originate from the Digital Elevation Models (DEMs) of the dry season of 2017 and 2019. The DEMs were produced by Agisoft Photoscan with an Structure from Motion-MultiView Stereo (SfM-MVS) approach. **a** marks the beginning of the beach Thoondu, showing a slight increase of the sandy beach face. **b** shows a transect from the central area of the east coast, suffering slight erosion of about 0.26 m to 0.28 m at the toe of the beach. Erosion like this can occasionally be measured in other areas of the central east side but it is not constantly detectable along this section of the coast. **c** and **d** show profiles of the southeastern coast adjacent to the seaport. The steep edge is the transition of the main island into the reef flat. Here, the coast suffers from constant, structural erosion of about ~0.3 m with maximum values of ~1.33 m between 2017 and 2019 (see Fig. 3). The map in **e** shows the locations of the profiles on Fuvahmulah's east coast. The map is based on data from field measurements and adapted publicly available © OpenStreetMap contributors data, accessed through the open-access Geofabrik download server.

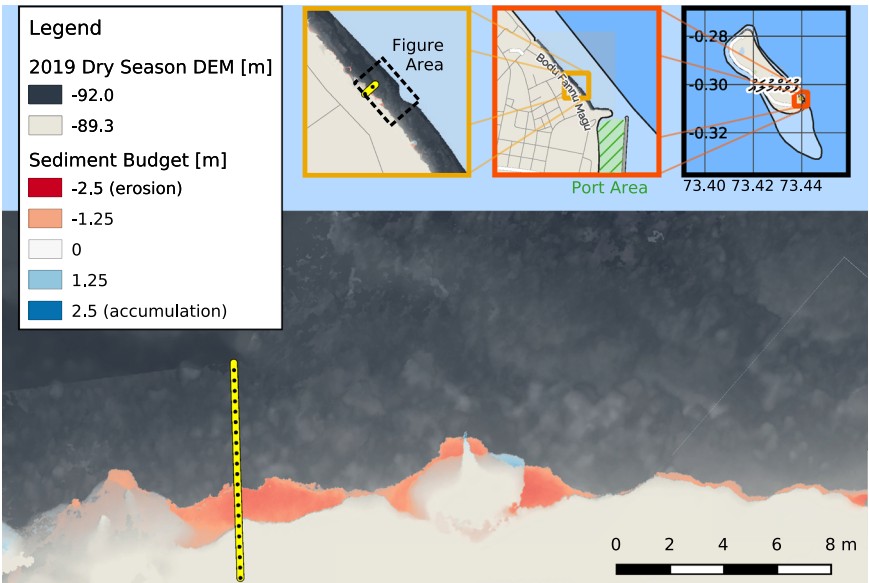

**Fig. 3 Erosion adjacent to the port.** The Digital Elevation Model (DEM) of the dry season 2019 on the southeastern coast of Fuvahmulah, adjacent to the seaport. The figure shows erosion (red) and accumulation (blue) when compared to the DEM of the dry season 2017. The yellow dotted line is the most southern transect of this study with cross-sectional differences of ~1.33 m (see Fig. 2d). Maps contain data from field measurements, mini maps also use adapted publicly available © OpenStreetMap contributors data, accessed through the open-access Geofabrik download server.

Along the southern stretches of the east coast, there are further signs of erosion, such as uprooted trees. For example, adjacent to the most southern cross-section was a palm tree that later fell onto the reef (see Fig. 3). Large parts of Fuvahmulah's coastal ridge have a coastal forest ("heylhi"), reinforcing the sandy beach profile with its roots. On the east side, waves and currents carve out the sediment under the roots and dislocate broken bedrock material. These observations show that erosion on the east side of Fuvahmulah starts above the reef flat at the bottom of the beach profile. These observations also help to put erosion into context: while the northern transect shows slight sediment accretion, the central part shows shoreline retreat with maximum values of 0.26 m to 0.28 m between 2017 and 2019 (Fig. 2d). However, shoreline retreat was only occasionally measurable and not present over the entire central coastline – in contrast to the southeastern coast, adjacent to the port. This area experiences substantial ongoing erosion along its entire coastline in the order of 0.3 m between 2017 and 2019 with maximum values ~1.33 m (see Fig. 3). The field data quantifies erosion on the southeast of Fuvahmulah in the order of decimeters over two years. Erosion rates on the east coast decline towards the north. In the northeast of Fuvahmulah, the highly dynamic morphology of Thoondu stabilizes the coast.

**Wave climate of the southern Maldives**. Both, erosion adjacent to the harbor and sediment dynamics on Thoondu beach, are effects of sediment transport dynamics on the reef. Here, the main driver behind sediment transport are wave-induced currents[5,7]. Therefore, the local wave climate and possible changes of the wave climate are of particular importance when assessing shoreline changes. As measured wave data for the Maldives is scarce, global wave hindcasts or reanalysis models are the only sources containing long-term wave data of directional sea states for the islands of the Maldives. Information on wave data is provided by several meteorological services, such as the National Centers for Environmental Protection (NCEP) of the American American National Oceanic and Atmospheric Administration (NOAA), the European Centre for Medium-Range Weather Forecasts (ECMWF) or the Collaboration for

Australian Weather and Climate Research (CAWCR) (see Fig. 4a). Annual time series from the considered services agree very well among each other, with mean annual correlation coefficients of $R \geq 0.89$ (Fig. 4c). To validate the hindcast and reanalysis data, this study uses harmonized and inter-calibrated Satellite Radar Altimetry (SRA) measurements from the Altimeter Data System (ADS) of the Helmholtz Centre Potsdam (GFZ)[45] and compares these to the three data sets considered for this study (Fig. 4a). The SRA measurements are available between 1993-04-25 and 2018-06-15. Wave heights from ECMWF's fifth generation atmospheric reanalysis of the global climate (ERA5) data set are closest to the measured SRA wave height in the region of Fuvahmulah. Even though the locations of the CAWCR output nodes differ from the other data sets, analyzing the spatial variation of the CAWCR time series shows a very high correlation of adjacent data nodes ($R \geq 0.977 \pm 0.007$, see Fig. 4d). The spatial similarity following from this high correlation of wave parameters allows the data sets to be compared among each other – despite the spatial distance. Likewise, this shows the wave climate on Fuvahmulah is typical for atolls in the south of the Maldives – as well as the associated exposure and impacts. All further analyses of this study make use of the CAWCR data, because the output node is closest to Fuvahmulah and CAWCR provides significant wave heights under Representative Concentration Pathway (RCP) 4.5, and RCP 8 for the twenty-first century[46]. These future projections allow this study to scrutinize future states of wave climate on the island as well as their impacts and triggered effects.

The wave climate in the area around Fuvahmulah reflects the dry and wet season both by changed wave heights and wave direction. Median wave heights range between 0.98 m in March, as well as 1.71 m in July with a maximum wave height $H_{s,\max}$ of 3.32 m on June 22nd, 1987 (for more statistical data of Fig. 5a, f, for example on boxplot mean and IQR values, see Supplementary Information File). This study defines the period between November to February as dry season and the period between April and September as wet season (Fig. 5a). April and October are considered transition months.

The waves are smaller in the dry season than in the wet season and are dominantly approaching the island from SSW (202.5°, see

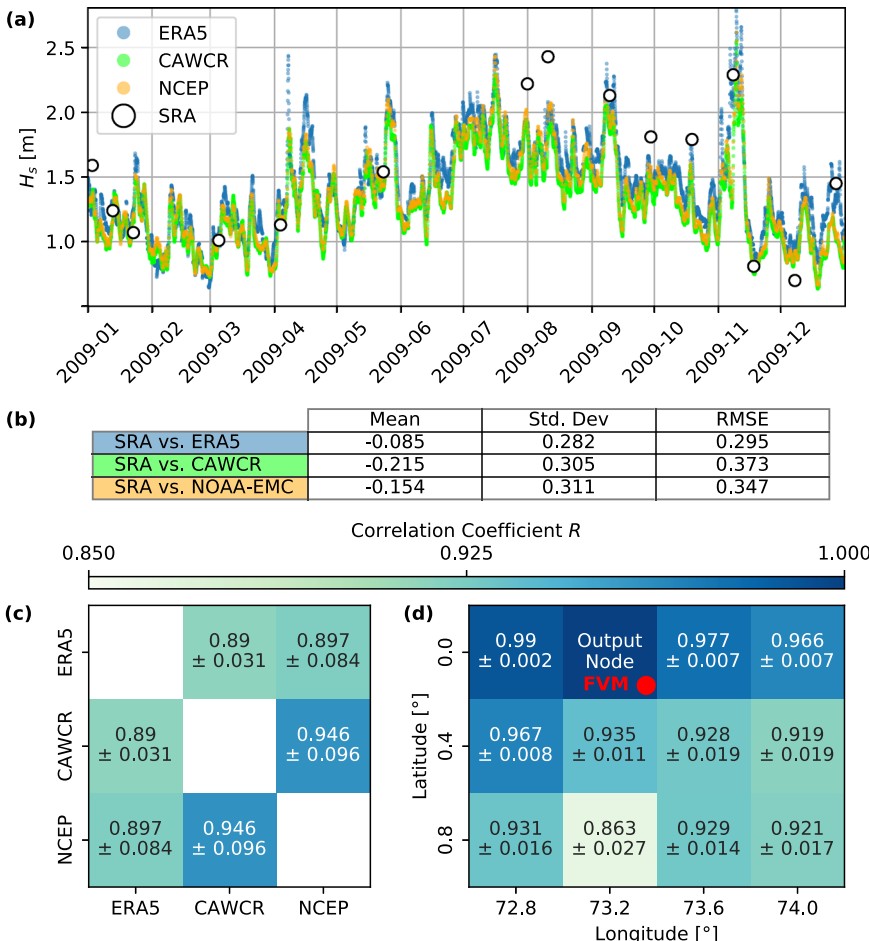

**(b)**

| | Mean | Std. Dev | RMSE |
|---|---|---|---|
| SRA vs. ERA5 | -0.085 | 0.282 | 0.295 |
| SRA vs. CAWCR | -0.215 | 0.305 | 0.373 |
| SRA vs. NOAA-EMC | -0.154 | 0.311 | 0.347 |

**Fig. 4 Time series of local wave data in comparison to regional and projected future wave data. a** Typical time series of the significant wave height $H_s$ for each data set over one year (here 2009). The data sets are provided through the climate data repositories of the European Centre for Medium-Range Weather Forecasts (ECMWF), the Collaboration for Australian Weather and Climate Research (CAWCR) and the National Center for Environmental Protection (NCEP) of the United States National Oceanic and Atmospheric Administration (NOAA). White dots are Satellite Radar Altimetry (SRA) measurements of $H_s$, recorded in the vicinity of the data output node. **b** shows the deviation between data sets and the SRA measurements **c** is a cross-correlation matrix for $H_s$ from each considered data repository between 1980 and 2019-05 (sample size $n = 345,504$ for ECMWF's and CAWCR's hourly data; sample size $n = 115,168$ for NCEP's 3-h data). **d** is a spatial cross-correlation matrix for data nodes of the CAWCR data set adjacent to the output node considered in this study. The data nodes are in the center of each cell. Values in the matrix are mean $\mu_R$ and standard deviation $\pm \sigma_R$ of the correlation coefficient $R$. The red dot marks the location of Fuvahmulah within the matrix.

Fig. 5b–e). There are also other southern and south western portions between November to February, but in terms of occurrence they play a minor role. Significant wave heights $H_s$ are mostly smaller than 1.7 m in the dry season (see Fig. 5c). In the wet season, waves approach the island dominantly from southeast and south-southwest (135° and 202.5°). In the rainy season, significant wave heights increase when compared to the dry season and range between $H_s = 1\,\text{m} - 2\,\text{m}$. Considering the future wave climate in the region, by using the projections of the Coupled Model Intercomparison Project Phase 5 (CMIP5), the data does not significantly differ under the respective RCPs, when comparing wave parameters between the reference time frame 1986–2005 and the last two decades of the twenty-first century (Fig. 5f; reference time frame according to the fifth assessment report of the IPCC[47]). These results are in line with findings of another study using CAWCR wave projection data on a global scale[48].

Considering the results of the statistical wave climate analysis, this study concentrates on scrutinizing waves which propagate from SE and SSW into the computational domain of the numerical models in the following. These wave directions represent the two dominant shares of the wave rose for Fuvahmulah in both seasons. Considering today's and future significant wave heights $H_s$, the numerical models use the 99th percentile from the reanalysis data set ($H_{s,\text{model}} = 2.3\,\text{m} \approx H_{s,99\%} = 2.27\,\text{m}$) as boundary conditions. All conditions are therefore modeled for storm conditions.

**Natural morphodynamics on the reef and anthropogenic interventions.** Results from the wave climate analysis serve as boundary conditions for the regional wave models[5]. The numerical wave models also utilize the DEM data and set the observed and measured morphodynamics into broader context by considering the wave climate's hydrodynamic forcing. With that, the wave models test the assumption of the harbor infrastructure being the root cause of erosion on the southeast coast, adjacent to seaport. While Delft3D (D3D) calculates the general sediment movement on the Fuvahmulah reef platform for different wave directions, the depth-integrated (2DH) Boussinesq-type wave model gives more detailed information on the role of the port as anthropogenic interference in the natural sediment transport system of the reef (a preceding study gives

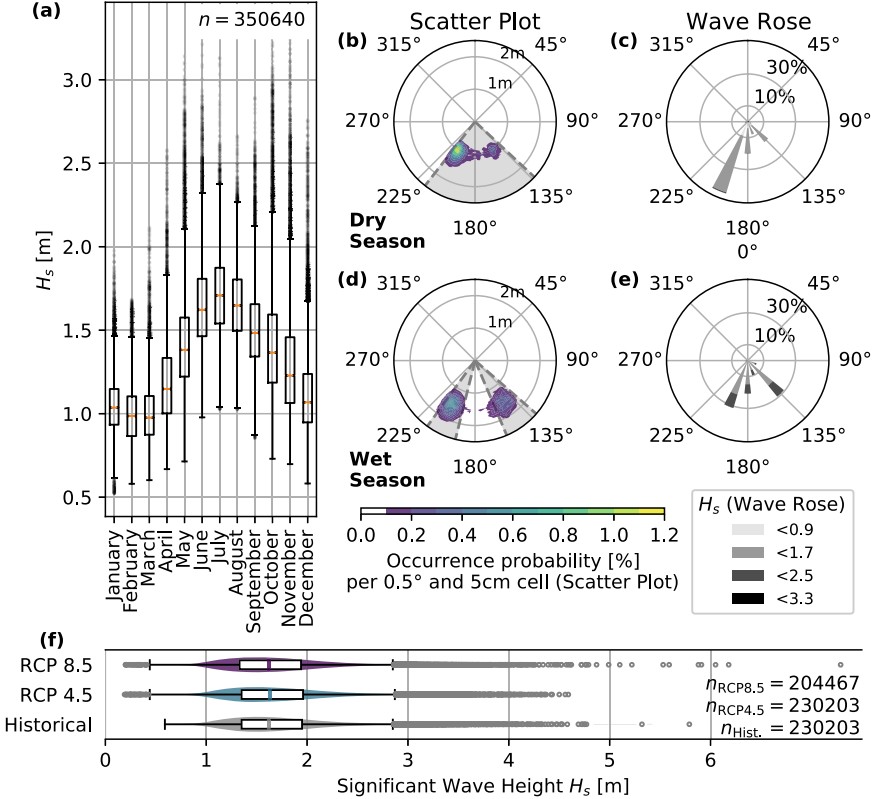

**Fig. 5 Wave statistics for Fuvahmulah. a** Annual significant wave height distribution for hindcast data between 1980 to 2019 (sample size $n = 350{,}640$). **b** shows the discrete marginal distribution between peak wave direction $\theta_p$ and significant wave height $H_s$ for the dry season, while **d** contains results for the wet season. Here, the colorbar shows the occurrence probability of each 0.5° wave direction and 0.05 m wave height cell. The gray areas visualize the significant peak direction range $\theta_{p,r}$ for each subset, containing the directions which, when combined, have the highest 33% occurrence probability. **c** and **e** show wave roses for the dry and wet season. **f** are significant wave heights for 2010–2019 (here labeled as "Historical"; sample size $n = 204{,}467$; undefined "NaN", or "Not a Number", values remain unconsidered) and Representative Concentration Pathway (RCP) 4.5 and 8.5 (sample size $n = 230{,}203$; undefined "NaN" values remain unconsidered). Boxplots in **a** and **f** are done with the corresponding boxplot-function in Python's module matplotlib. Here, the orange line of the boxplot is the median, the box limits are the upper and lower quartiles while whiskers mark the range of the non-outlier data and extend the box limits by $1.5 \cdot$ IQR.

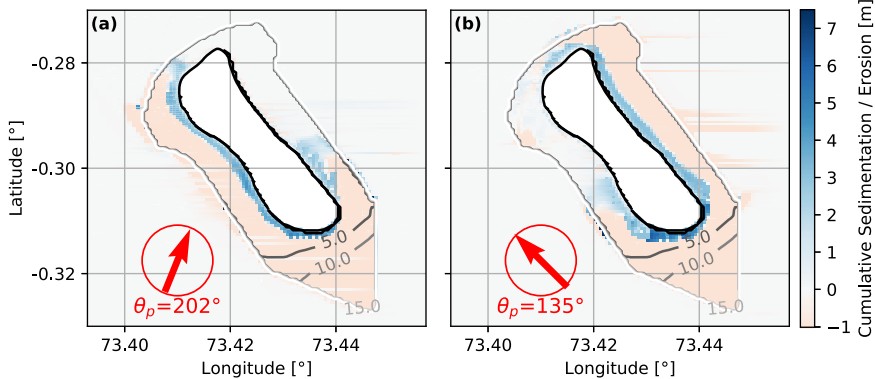

**Fig. 6 Modeled sediment transport.** Sediment relocation calculated with Delft3D (D3D) for constant waves of one month with significant wave height $H_s = 2.3$ m, peak period $T_p = 17$ s approaching from **a** $\theta_p = 202°$ and **b** $\theta_p = 135°$. The offshore depth is truncated to 100 m. The idealized reef has a depth of 5 m to 17.5 m. Gray contour lines mark the water depth on the reef between 5 m to 15 m in 5 m increments. The erodible sediment layer is 1 m throughout the domain. Sediment accumulation is blue, while erosion is red. The maximum erodible depth is equal to the erodible sediment layer thickness. Map data is idealized, based on measurements.

further insights into the more general dynamic pattern of seasonal and annual sediment transport on the fringing reef of Fuvahmulah[5]).

D3D confirms that the interplay of sediment supply[13] and wave-induced currents provides sediment for the island (Fig. 6). It further indicates where to expect sediment deposits under the given hydrodynamic forcing. As waves approach from $\theta_p = 202°$, sediment moves towards the island and settles on the west side (Fig. 6a). A smaller sediment depot also forms on the east and southeast of the island under the given premises. Waves from $\theta_p = 135°$ will take sediment from the offshore reef and transport it towards the island (Fig. 6b). The computations for waves from

$\theta_p = 135°$ disclose that the sediment stream splits at the southeastern tip of the island, approximately at today's harbor location. The wave-induced currents distribute the sediment along the east coast of the island as well as on the southwest side. A small portion of sediment also moves around the northern tip of the island onto the lee side.

In general, the D3D computations reveal that wave-induced currents take sediment from the southern reef and transport it towards the south and east coast (without seaport structures being present). D3D also indicates the particular importance of the island's southeast part for sediment transport along the entire east coast: waves from $\theta_p = 202°$ pick up sediment at the southern reef, transport it towards the east and the southeast coast of Fuvahmulah where it finally accumulates. Then, over seasonal cycles, waves from $\theta_p = 135°$ can take away the accumulated sediment and distribute it along the east side.

The 2DH model highlights the difference in wave driven current patterns leading to sediment transport around the port area. The models also facilitate scrutinizing the changes in sediment transport with and without the harbor infrastructure being present. These computations confirm the processes outlined with the D3D computations and further highlight two factors, contributing to the erosion along the east coast of the island:

The first factor is the available sediment. In both cases – with and without the port structures (breakwater and headland) – waves from $\theta_p = 202.5°$ create a current in front of the port (Fig. 7a, b), transporting sediment over the reef. At the same time, in the area of today's harbor entrance, the current decelerates and thus allows for the sediment to accumulate in this area. However, with the breakwater present, the structure interrupts the sediment transport and sediment cannot enter this area. In addition, the breakwater reaches up to the reef's edge. As a consequence, it deflects the current and thus redirects the sediment off the reef into deeper waters.

The second factor is the transport capacity of the east coast's current: without breakwater, waves from $\theta_p = 135°$ induce a northward current to the area of today's port entrance (see Fig. 7). This is also the area, where sediment was able to settle from the $\theta_p = 202.5°$ component. Instead, with the harbor present, the breakwater obstructs the emerging current in this area and deflects the velocity momentum off-shore. Yet, a wave-induced longshore current is still present along the east coast independently of the port: when waves from $\theta_p = 135°$ approach the reef and break, they induce radiation stresses and subsequently create this longshore current. With missing sediment from the reef, the currents will likely take sediment from the coast, leading to erosion.

**Governmental and societal framing of coastal development.** On Fuvahmulah 93.8% of residents are citizens of the Maldives and as on most other inhabited islands, tourism plays a minor role[41]. Fuvahmulah's island dwellers are mainly employed in education and commercial services[41]. Thus, together with the insights of the field campaigns, Fuvahmulah can be considered a local's island. In the household survey, the local community perceives erosion as the most pressing issue (closed question, 27% of 345 mentions, see Fig. 8 and Supplementary Information File). At the same time, both the interviews with government officials as well as the recently issued Environmental Impact Assessments (EIAs) on coastal protection on Fuvahmulah reveal that administrations on the national and local level acknowledge erosion as a high priority issue on the island. However, while there is consensus on the need for action among all interviewed actor groups, attribution of the erosion's root cause varied between the national government

representatives and the affected community. On the one hand, the Maldives' dominant adaptation challenge is sea level rise and the associated impacts. In fact, the Maldives national government actively promotes this narrative of being highly vulnerable to climate change induced sea level rise[49] – even in the case of Fuvahmulah. On the other hand, Fuvahmulah's population is skeptical about the national government's narrative. The household survey contained an open question (without pre-formulated responses) asking the participants to name visible changes of the natural environment on the island. Here, about 36% of all responses mentioned erosion and of these, 20% also attributed the erosion processes on Fuvahmulah specifically to the harbor construction (see Fig. 8 and Supplementary Information File). These diametrically opposed perceptions on root causes behind coastal adaptation originate from the historical context of national politics and local society.

First experiences with coastal infrastructure in the Maldives dates back to the 1970s, when the first modern coastal protection constructions were built on the islands[34]. However, coastal protection management was not professionalized before 1987 and 1988 when strong coastal floods caused massive damage to the Maldives' capital Male' and on several other islands in the middle of the country (interview with representatives of the Ministry of Environment and Energy (MEE) in 2017). At the same time, establishing coastal protection in the Maldives was accompanied with a rapid increase in population, especially in the Maldives capital Male'. In accordance with the Maldives Decentralization Act in 2010, the national government shifted its focus onto regional development centers, equipped with the necessary infrastructure[26] and limiting the migration to the capital[50]. However, the interviewees still highlight today's accumulation of resources – economic and professional expertize – in Male'. Decisions are made at the highest level and have an impact down to the local scale. When regarding the implementation of coastal development projects in the Maldives, the planning, implementation and decision-making process are centrally executed and ministerially anchored in the national government without significant involvement of local capacities on the islands. According to actors involved in coastal governance, in general, such top-down processes in decision and implementation are applied for coastal infrastructure projects, for example seaports as well as coastal protection structures. Regional and local government institutions, such as the city council on Fuvahmulah, lack influencing power in the decision-making process. According to the interviewees, the role of local government institutions is limited to informing national-level actors about coastal problems on their island. The lack of power is also expressed by missing financial resources for coastal projects at a council-level (interview with a representative of a state environmental agency in 2017). In addition, infrastructure projects usually require external financial and knowledge resources, provided by international organizations. However, international organizations are legally bound to use the national government as an entry point and cannot initiate projects below the central national-level. Altogether, this stands in contrast to the republic's decentralization efforts and the corresponding strengthening of local communities.

Moreover, since 2012, the EIA Regulation of the Maldives requires to assess the (adverse) impact of infrastructure projects on the environment. However, quality and compliance to EIA policies – also for example in the tourism sector[39] – are traditionally weak[40] and have extensively promulgated technical fortification of many inhabited islands in the Maldives[20,22,26].

In contrast to the national politic perspective on coastal infrastructure, the traditions and knowledge of Fuvahmulah's inhabitants on their environment as well as experience with

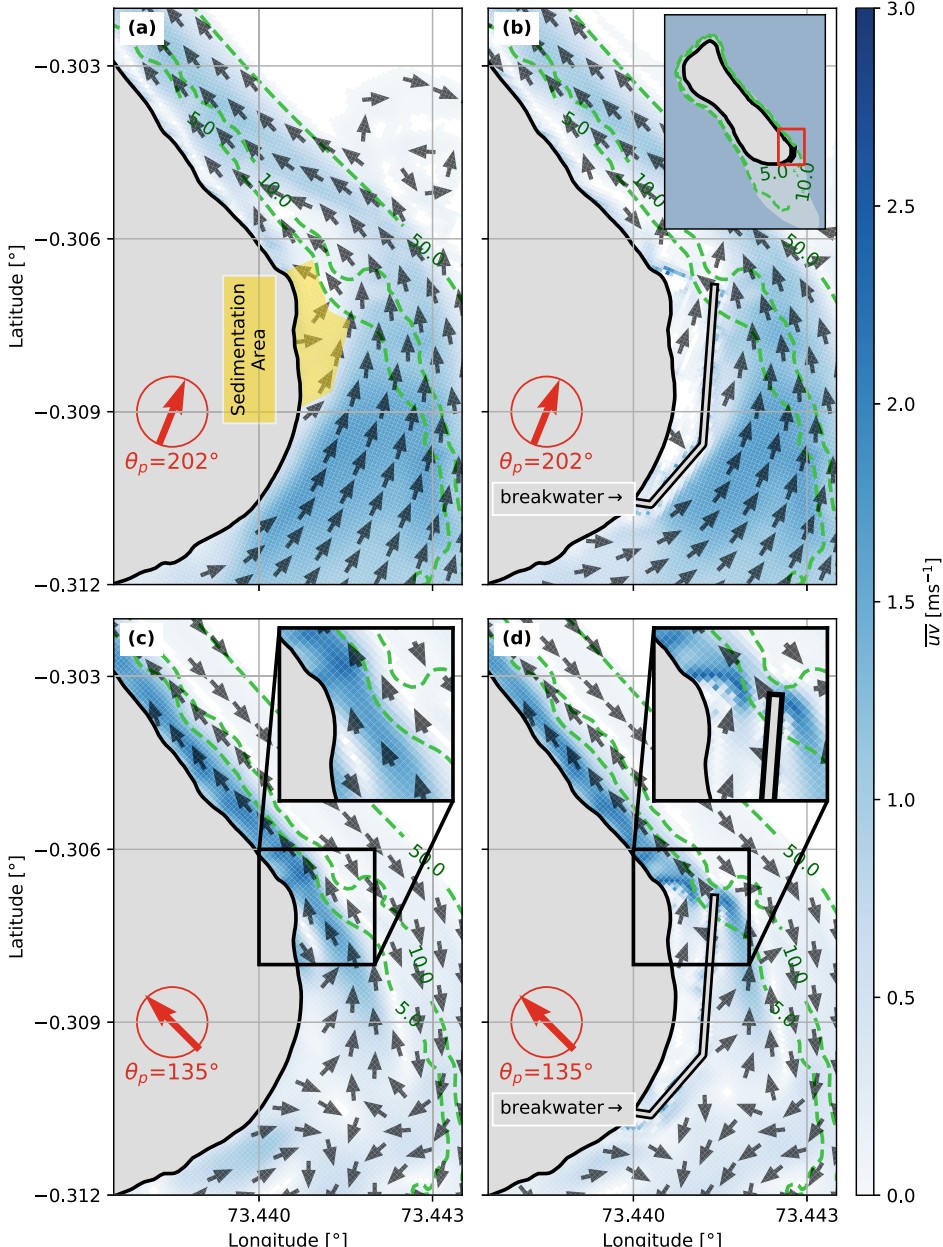

**Fig. 7 Wave-induced currents and impact of the seaport on the natural hydro- and morphodynamics.** Wave-induced currents $\overline{uv}$ from the depth-integrated (2DH) model for the seaport area in the southeast of Fuvahmulah, where the colorbar shows the magnitude of the current between $0.0\,\text{ms}^{-1}$ (white) to $3.0\,\text{ms}^{-1}$ (blue), while arrows depict the current direction. **a** shows velocity fields from waves with a peak direction of $\theta_p = 202°$ for Fuvahmulah without the harbor, while **b** includes seaport infrastructure. Similarly, **c** shows wave-induced currents before the port construction, while **d** presents today's situation with the harbor present for waves approaching from $\theta_p = 135°$. Green dashed contours show the water depths of the reef platform for 5 m, 10 m, and 50 m. Maps are based on measurements and adapted publicly available © OpenStreetMap contributors data, accessed through the open-access Geofabrik download server.

coastal hazards led to a different perception on coastal adaptation. More than half of the local respondents feel safe on their island with regard to sea level rise (52% of the respondents, see Fig. 8). The interviews have shown this sense of safety evolves from the awareness that the lower-lying center of Fuvahmulah is protected from seaborne extreme events by the island's fringing coastal ridges. Locals even believe Fuvahmulah to be comparatively safer than other islands in the Maldives. Some also mention the protective functions of the reef: one respondent answers an open question, asking which elements of the island's natural environment are perceived as important: "*most important is the reef. What I'm saying is that we are protected by the reef. It's not like*

the [other islands in the Maldives] *that are protected by more islands in the atoll*" (Survey 1 participant: 080). Others have argued about the protective function of the "heylhi", which is the natural green belt of vegetation around the entire coastline of Fuvahmulah, stabilizing the shore and protecting local inhabitants from seaborne hazards (see Supplementary Information File). In addition, building houses was traditionally only allowed in the center of the islands behind the green belt, consisting of local trees and shrubs. However, in 1993, a 86 m long seawall was built on the central east-side as the first protection measure on the island, followed by noticeable erosion[44]. Today, the seawall is detached from the coast, substantiating this study's earlier

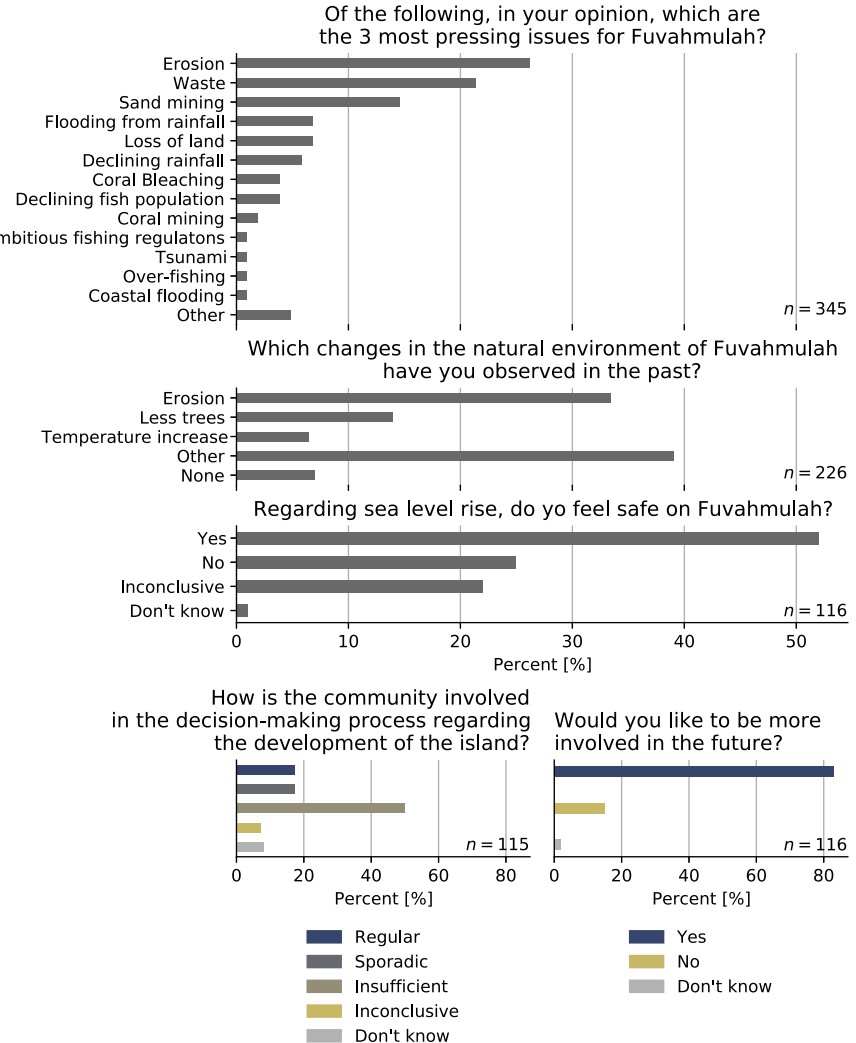

**Fig. 8 Results from the population survey.** Selected questions and answer frequency from the population survey. See Supplementary Information File for the question type (open or closed question) as well as associated codes and coding rules.

findings and a general propensity of adverse impacts of human-made interventions in the coastal environment. While there hasn't been a documented response to the seawall's impact at the time, today's erosion adjacent to Fuvahmulah's seaport is more severe and has a higher impact on the society. For example interviewees of the second survey discussed the effects of erosion as well as the impact on their personal lives, for example: "[…] I am very worried about the soil erosion. I think around 20 feet has gone. So, it is a huge risk for our people in our community. So, I am very much worried about it." (Survey 2 participant: 039). Fuvahmulah has gained more attention by the national government in the last two decades, manifested through numerous development projects in that time, thus erosion – being a high priority issue on the island – is a matter dealt with on the highest governmental level nowadays.

**Coastal adaptation in practice**. In 2014, government actors of the MEE agreed on taking action against erosion adjacent to Fuvahmulah's seaport. They signed a grant arrangement for an internationally sponsored infrastructure development fund and initiated the project "Coastal Protection at Gn. Fuvahmulah, Maldives". This project includes "the development, implementation and maintenance of sustainable coastal protection to prevent erosion and flooding on […] Fuvahmulah. The main objective of

this project is to decrease erosion and flooding through a possible […] combination of hard and soft coastal engineering interventions […] to protect the island of Fuvahmulah against flooding due to ongoing coastal erosion and rising sea levels"[50]. An international consultant supported the MEE together with a local sub-contractor in the site investigation, the feasibility study, the technical design[50] and the draft of the legally mandatory Environmental and Social Impact Assessment (ESIA)[51].

The technical design report finds human interventions and changes in natural processes disturb the natural balance of the sediment supply along the coastline[50]. According to the report, these human interventions are more specifically the harbor construction, sand-mining and earlier reef blasting for beach access, while natural processes disturbing the natural balance are "changes in the natural climate"[50], but also "may"[50] be a possible deterioration of the coral reef. The report also mentions a lacking local sediment source for the coast and predicts the fringing ridge of Fuvahmulah to breach in the next decades, causing "tremendous flood damage to the island"[50]. Because of this, the report concludes, coastal protection is required and the design should "immediately stop the ongoing erosion"[50]. A preceding feasibility study was cited in the report, identifying two possible solutions against the ongoing erosion and rendering them reasonable while considering their environmental and social

impacts – an offshore breakwater and an onshore revetment. The technical report concludes the revetment to be the most viable option, after presenting preliminary designs and a cost-benefit analysis of both alternatives. The final revetment is about to cover approximately the southern 2.6 km of the 4.0 km long east side and is designed with a lifetime of 50 years, withstanding hazardous seaborne events with a return period of 100 years.

The ESIA[51] builds on the technical design report and further underlines the project's significance in the context of future climate change risks, as well as for the national and local economy. The report also finds the project would enhance the "sense of security and safety" of Fuvahmulah's island dwellers[51]. The ESIA initially gives five approaches to coastal adaptation, but finds ecological methods would contain a high uncertainty and that they would be ineffective. Also, the report further dismisses soft measures (also known as sediment-based measures) as unsuitable because a proper (local) sediment source is supposedly missing. The impact assessment report finally presents the same coastal protection alternatives that have been suggested in the technical design report[50]. However, in addition to the technical design report, the ESIA also contains stakeholder consultations on the project. Resulting from these consultations, the chapter "project risk mitigation" in the ESIA finds "acceptability of the Project by all stakeholders is an important factor for smooth implementation of any project"[51]. The first local stakeholders consultation took place in July 2016 on Fuvahmulah and addressed the off-shore and on-shore protection alternatives. The discussions revealed locals prefered the off-shore alternative, but the ESIA objects as "the risk of affecting [Thoondu] beach was not well understood [by the local stakeholders] at that time". Therefore, the ESIA mentions another consultation with the local government and community in October 2016 "to explain to the stakeholders the preferred alternative that emerged following the feasibility study and proposed in the EIA report" - with the revetment being the "preferred" and to be implemented alternative. In summary, both reports dismiss feasible soft solutions due to false assumptions (missing sediment source, however the reef naturally provides marine aggregates[5,9]) and undermine the participatory process by promoting the less popular alternative in the end.

In principle, public participation is a key component of EIAs[39] and when taken into the decision-making process, public participation also leads to more sustainable implementation of development projects. However, the population survey of this study from March 2017 reveals an unutilized potential to improve the participatory process based on two questions: First, when asking how far politicians work together with the community (open question), 49% of the survey's respondents found insufficient cooperation of politicians with the community in development projects – while 19% answer to be sporadically involved and 17% report to be regularly involved (see Fig. 8 and Supplementary Information File). Second, a large majority of 83% endorses more involvement in the decision-making process regarding the development of the island, while only 15% of the interviewees do not wish for more participation in this matter. These results disclose an ongoing dissatisfaction of Fuvahmulah's people and the lacking consideration of their interests, opinions, and knowledge – despite or because of insufficient earlier attempts to include their perspective in the ESIA.

## Discussion

Small island policy aims for an improved economic situation and at the same time for better provision of access. To improve this situation for the island dwellers of Fuvahmulah, a sea- and an airport were constructed at the time. As of today – almost 20 years later – climate change and sea level rise causes marine pressures and these pressures dominate the discourse on small islands. As a result, small island state governments prepare their nations for the associated adverse impacts. Coastal erosion is a pressing issue of inhabited reef islands like Fuvahmulah and adaptation measures need to address these issues under deeply uncertain future projections. Following the early example of the capital Male', the centralized government is in favor of generic adaptation measures and implements these measures in a top-down approach[26]. In addition, the government of the Maldives – as other SIDS[27] – most often relies on external, technical knowledge and funding to realize climate change adaptation and vital development projects. As a consequence, the resulting planning efforts and constructive responses are mostly based on external design guidelines and building codes. In general, experiences from the past are mainstreamed into international design guidelines and building codes[52,53], and then guidelines and building codes give best-practice examples under certain environmental conditions. In case of coastal structures these environmental conditions are predominantly those of mainland coasts. Endorsing hard coastal protection is common in SIDS[27] and based on both, their top-down governance structure as well as their dependency on international funding – two aspects, that are also a legacy from the colonial history of SIDS. In the Maldives, consequent consideration of hard structures and acceptance of their associated adverse impacts on the environment is historically grown, and ill-prepared policies and policy compliance is projected to continually exist[26,40]. In both cases, transcribing inappropriate mainland design criteria to sensitive reef environments have caused an insufficient level of coastal protection on small islands in the past. In most cases, there is a straight forward transfer from external knowledge and guidelines to local circumstances, overlooking the special features and services of the fringing coral reef. Yet, corals are important for the livelihood and protection of the accommodated reef islands. In terms of protection, the main ecosystem service of healthy coral reefs is their ability to attenuate waves[16] and to supply the reef island with coral sediments[5,54]. Generic hard-engineered coastal protection have led to disturbances of natural dynamics and the associated ecosystem services in the Maldives[20,22,26,34] and in other SIDS[25,55]. In a worst case scenario, generic hard-engineered approaches to coastal protection and adaptation even fail to alleviate but aggravate adverse effects on the natural environment. Also future plans for coastal protection on the Maldives[22] fail to implement a local perspective on dealing with changes. Instead, the propensity is to further fortify the island's perimeter with traditional hard-engineered measures[26] – as in the field case example of Fuvahmulah.

The example of Fuvahmulah epitomizes planning inadequate coastal structures in a reef environment underlies special environmental features and requirements beyond conventional, generic infrastructure design rules and solutions (see the results section "Natural morphodynamics on the reef and anthropogenic interventions" and Fig. 7). Traditionally, human intervention beyond the natural green belt of Fuvamulah used to be highly limited, leaving the protection feature and natural sediment distribution on the reef intact. However, the first hard-engineered protection measures have already led to complications[44]. Sand-mining is discussed as a factor and cause for coastal erosion on Fuvahmulah. Sand-mining can potentially put anthropogenic pressure on small islands[25] and on Fuvahmulah, sand-mining for private purposes was frequently witnessed within the field campaigns around the island. A further reason for increased erosion could be climate variability. However, this study does not include an in-depth assessment on recent climate variability and possible effects on the east-coast of Fuvahmulah, albeit there are

tendencies of changing climate impacts (for example sea surface temperature oscillations in the Indian Ocean), which could have affected the formation of Thoondu beach in 2019[5]. However, the most visible and distinct adverse changes on Fuvahmulah's coasts is local erosion, located adjacent to the seaport in the southeast. Based on the field campaigns and the recorded high-resolution data, this study finds coastal erosion on Fuvahmulah to be a negative consequence from the seaport. The seaport is an anthropogenic intervention into the coral reef system adversely impacting the natural sediment transport and the associated wave-induced redistribution pattern around the island – much more than sand-mining or climate variability.

Together with today's impact of the seaport, future modifications to the natural reef system will most likely have even more significant adverse effects for greater parts of the coast – especially in spatio-temporal dimensions of the project "Coastal Protection at Gn. Fuvahmulah, Maldives". This study has shown that in the case of Fuvahmulah, there are distinctive hot-spots, which are important for the sediment supply of the island's coast. The seaport is located at one of these hotspots, disturbing the natural sediment transport around the island (Fig. 7). The seaport construction was arguably a trade-off for economic development and societal well-being at the expense of natural conservation. Nevertheless it has triggered maldevelopment on Fuvahmulah, when assessing how the negative consequences on the environment are being dealt with: instead of governing this conflict[56], future plans are going to respond to erosion associated with the seaport's construction with further generic main-streamed adaptation measures[50,51] rather than acknowledging and compensating for the root causes of erosion (also being recognized by Fuvahmulah's island dwellers). Concluding from other studies' insights[22,26,32] and findings from this study, these measures fail to address processes of the reef's sediment transport and thus drivers behind erosion adequately. Incorporating the topographic and reef measurements as well as the wave climate in numerical wave models helps to scrutinize the root cause behind the measured erosion on Fuvahmulah's east coast: the seaport intervenes with the natural sediment transport and acts as barrier, deflecting suspended sediment off the reef (Fig. 7). Bypassing this barrier would be a suitable remedy resurrecting the natural sediment transport along the east side[57]. A bypass would nourish the beaches and act as "low-regret" adaptation to the anthropogenic disturbance, because it limits the impact on future response options under different climate change scenarios (sediment bypassing is described in typical coastal engineering manuals, such as the Shore Protection Manual[52] or the Coastal Engineering Manual[53]). However, instead of providing a more robust and flexible approach towards coastal adaptation, the current adaptation plan is in favor of fortifying the entire coastline – a widespread approach in the Maldives and common pitfall, which is known to undermine the islands' natural capacity to adjust to ocean climate-related pressures and a crucial step towards reaching an "anthropogenic tipping point"[22]. Hard-engineered responses can undermine the natural dynamics feeding the reef island with coral sediment. This study illustrates this on Fuvahmulah, but the findings are scalable to further examples in the Maldives[26] and on other SIDS[27].

The current decisions behind the present adaptation plan on Fuvahmulah lead to a repeated violation of the reef environment's natural prerequisites and omission of societal demands. Following the DPSIR chain of causal links, this continuously affects the current state adversely and leads to the associated maladaptive impacts on the environment and society. Scrutinizing the case study of Fuvahmulah with an interdisciplinary approach documents the significance of societal demands and political decisions leading to maladaptation. It shows that maladaptation is implemented in a socio-political context, aiming to improve the economic or societal situation by consistently trading-off ecosystem services. However, instead of compensating and mitigating these trade-offs seriously, the socio-political responses lead to cascading maladaptive activities. In the Fuvahmulah example, the constellation of funding source, client and contractor already impedes an unbiased and sincere assessment which in turn inhibits an adequate reaction to the current erosion issue: the project "Coastal Protection at Gn. Fuvahmulah, Maldives" is initiated by the MEE and funded by an infrastructure development facility of an external government. At the same time, the MEE is also consulted by experts of the funding facility's home country. Together with the local sub-contractor Maldives Energy and Environmental Company (MEECO), these experts carry out an ESIA, which has to be acknowledged by the MEE – being contracting and supervisory authority at the same time. The subsequent ESIA and technical report dismissed a potential soft solution, such as beach nourishment, claiming no sustainable sediment source was available on the island. However, since the construction of the port, the area west of the seaport and south of the airport host large quantities of deposited sand[5], which wasn't present prior to the harbor construction (visible on aerial and satellite images in former reports[43,44] and the EIA[51]). The external contractor disregarded the reef as sediment source and sediment transport patterns on reef islands – even though this information was made available through research in the past decades[9] and is confirmed to be valid for this island as well[5]. This research of the past decades has emphasized the reef's natural protection capacity and its ability to feed the island in the face of sea level rise. Neglecting these vital ecosystem services and transferring generic solutions for mainland coasts beyond their scope of application is another rigorous example of building *in* nature – in contrast to a sustainable building *with* nature[58]. However, the technical report and EIA on the coastal protection project discusses only hard-engineered alternatives in the stakeholder consultation, while the measure less preferred by locals was finally chosen. By framing stakeholder consultation as project risk mitigation instead of equitable part of decision-making, the ESIA epitomizes the deficits of prescribing development projects in a top-down manner and lacks to provide a solution acceptable for the society and suitable for the environment. Somewhat arbitrary compliance with national EIA guidelines is commonplace in the Maldives, leading to questionable assessments regularly[39,40] and ultimately failing to address the root causes of adverse impacts on reef environments. An "unduly overemphasis"[36] on hard protection is a symptom of maladaptation and in the Maldives this symptom originates from the socio-political history of the country and is still prevalent today. Yet such recurring and systematically provoked maladaptive actions go beyond the initial scope of maladaptation. Against this background, this study defines the concept of maldevelopment: maldevelopment is a socio-political phenomenon amplifying maladaptation which is characterized by decisions or policies in constant or deliberate favor of inadequate actions and trade-offs towards future climate change related risk. Structural maldevelopment to date impairs the potential of dealing with additional future changes, such as sea level rise, extreme wave events, and storm surges. Studying maldevelopment in its comprehensiveness leads to the conclusion that sustainable development requires an integrated analysis of political interests and societal demands within the natural boundaries, in order to adequately address future climate change stressors. This study finds structural maldevelopment being the most important human driver of undesired coastal changes on small islands. Risks evolving from maldevelopment add onto the current and future risks of the other significant natural driver, being climate change

and the associated impacts. This is very likely not only true for Fuvahmulah, as many small islands in the Maldives and in the world show symptoms of maldevelopment[22,25,27,28].

The perception analysis and population surveys on Fuvahmulah reveal that island dwellers are well aware of the port's impacts and are doubting the national government's narrative, framing coastal erosion as climate change and sea level rise impact. Local knowledge helps to identify and overcome local challenges, reflected by both the traditional primacy of island development within the borders of the "heylhi" and veritable attribution of the current coastal erosion. Analyzing the wave-induced sediment processes in numerical wave models corroborates the local people's assumption that the harbor infrastructure interrupts coastal dynamics and acts as a sediment barrier for the east coast. Local knowledge conveys a distinct local capacity towards economic development in combination with future environmental protection on the island. Implementing and evolving these capacities into a tailored adaptation pathway plan facilitates to consider smaller and less intrusive responses beyond conventional hard-engineering approaches, discloses their potentials and limitations, and leaves further scope of action for the future. Such a more conceptual approach of adaptation pathway planning requires (1) a greater focus on the underlying process, (2) a systematic collaboration between local, national and external experts and stakeholders on the island, (3) ongoing monitoring on the natural and societal impact of the implemented measures, and based on this (4) frequent adjustments on the pathway roadmap by responsible authorities when an adaptation tipping point is reached and the current status (a policy action or actions) will fail. In this regard, dynamic adaptive policy models represent suitable, modern methods, allowing more conceptual approaches to be embedded into continuous and sustainable coastal planning[59].

Amalgamating coastal and societal aspects leads to a comprehensive understanding of the ongoing natural and socio-political processes behind coastal issues. These interdisciplinary insights underline that adaptation strategies and measures need to be adjusted to local circumstances. This helps to (1) avoid an interruption of the coastal system, (2) undermine vital ecosystem services and (3) sustain natural coastal protection.

## Methods

**Digital elevation model (DEM).** Based on three field campaigns, Unmanned Aerial Vehicles (UAVs) capture the topography of Fuvahmulah and facilitate analyzing shoreline changes. A DJI Phantom 4 multicopter recorded aerial photos within two field campaigns in March and September 2017. A DJI Phantom 4 Pro was used for the coastal survey in March 2019. The aerial surveys covered the entire coast of Fuvahmulah, except the beaches adjacent to the airport, which are a no-flight-zone. After recording the aerial images of Fuvahmulah's coast, Agisoft Photoscan Pro (version 1.4.5 build 7354) processes the photos into DEMs by means of Structure from Motion-MultiView Stereo (SfM-MVS)[60]. DEMs are three dimensional models of the area, containing height and color information, here in about 3.5 cm px$^{-1}$ resolution. The resulting coastal DEMs were recorded at different points in time, facilitate analyzing temporal changes of sediment volumes on the beach, shoreline location and vegetation. The general workflow in Photoscan is to align the photos, georeference the model with Ground Control Points (GCPs), and create a dense point cloud as well as a mesh of the area[61]. Photoscan derives the DEM from the dense point cloud, while the mesh serves as base for the orthophoto (for further information on input parameters of each processing step in Photoscan, see Supplementary Materials). To position the models in the World Geodetic System (WGS84), the March 2019 DEM uses a unique set of GCPs around the island. The 2017 model uses the drone's Global Navigation Satellite System (GNSS) antennas for initial positioning. Afterwards virtual Ground Control Points (vGCPs) of specific landmarks available in both models georeference the 2017 DEMs with the 2019 DEMs. Apart from the erosion, the position of dedicated landmarks between all three field campaigns were reproduced with the expected accuracy[61,62] for SfM-MVS procedures in coastal areas.

In general, the inspected erosion starts at the bottom of the beach profile. Waves and currents carve out the sediment under the roots of the coastal forest or dislocate coral rocks. Owing to the specific erosion process on Fuvahmulah, early erosion remains mostly undetected in the DEMs, mostly until the upper part slips

or collapses. However, adjacent to the seaport, the erosion is advanced enough to be well visible in the DEMs.

**Hydrodynamic boundary conditions.** To evaluate the drivers behind sediment transport, this study uses numerical near-shore wave models. These models require regional wave climate information as hydrodynamic boundary conditions. To derive the wave climate for Fuvahmulah, this study considers data from three global hindcast and reanalysis models from the Collaboration for Australian Weather and Climate Research (CAWCR), National Centers for Environmental Protection (NCEP) of the American National Oceanic and Atmospheric Administration (NOAA), and European Centre for Medium-Range Weather Forecasts (ECMWF) (see Table 1). Data from wave spectra are an appropriate resource to be used as boundary condition in models of finer spatio-temporal resolution[63,64]. To validate the modeled wave climate data, this study compares each data set with wave heights derived from individual harmonized and inter-calibrated SRA measurements of the GFZ Altimeter Data System[45] (see Fig. 4a). The SRA data is taken from latitude 0° and longitude 73° in a radius of 1.5 km between 1993-04-25 and 2018-06-15.

The CAWCR provides hindcast wave data[65] generated with the WAVEWATCH III ocean wave model[66]. The closest output node to Fuvahmulah and the SRA data from GFZ is at 0.0° latitude, 73.2° longitude. The model uses a grid with 0.4° zonal and meridional wide increments. CAWCR also provides estimates for wave heights under different RCP in the twenty-first century. WAVEWATCH III is a numerical model calculating wind wave generation and wave propagation based on the random phase action density balance equation for wave spectra[66]. The waves are forced by surface wind fields from atmospheric models[67], made available by the NCEP as Climate Forecast System Reanalysis (CFSR) and Climate Forecast System Reanalysis v.2 (CFSv2). The Environmental Modeling Center (EMC) of NCEP runs an own hindcast with WAVEWATCH III and their bias corrected wind forcing. EMC maintains WAVEWATCH III and places the code at disposal. WAVEWATCH III is a fork of the ocean wave model (WAM)[68]. The ECMWF uses the WAM model for their ERA5 with wind forcing from their Integrated Forecasting System (IFS)[69]. The data is available through the Copernicus Climate Change Service (C3S)[70]. The closest data node to Fuvahmulah for both NCEP and ECMWF's ERA5 are at 0.0° latitude, 73.5° longitude (see Fig. 1). The SRA data from GFZ were taken from the node at 0.0° latitude, 73.0° longitude (Table 1 juxtaposes the underlying models, key input parameters and output variables of each data set for this study).

To compare the data sets, this study correlates ocean wave time series from all three repositories with a cross-correlation function from the nodes closest to Fuvahmulah. The general solution for a cross-correlation function writes as:

$$C_{AB}(\tau) = \frac{1}{N-m} \sum_{N=0}^{N-m} (A(m\Delta t) - \mu_A)(B(m\Delta t + \tau) - \mu_B) \quad (1)$$

with $\tau = m\Delta t$ as the lag time for $m$ sampling time increments and $N$ the number of values in the correlated time series $A$ and $B$, which are reduced by their respective means $\mu_A$ and $\mu_B$. The normalized cross-correlation follows from considering the standard deviations $\sigma_A$ and $\sigma_B$ of each time series as

$$\rho_{AB}(\tau) = \frac{C_{AB}(\tau)}{\sigma_A(\tau)\sigma_B(\tau)} \quad (2)$$

The normalized cross-correlation values range from 1 for equal signals over 0 for uncorrelated signals to –1 as a reverse or negatively correlated signal. The corresponding lag time $\tau$ of the highest absolute value, here the correlation coefficient $R$ of the cross-correlation, gives information about the time-shift between both signals.

The wave data in each repository contains time series of over 40 years. For the analysis, the time series is split up in annual sections from 1980 until 2019-05 and used separately in the cross-correlation function (Fig. 4b), resulting in 39 annual cross-correlation coefficients. When correlating the hourly data of CAWCR and ERA5 (sample size $n = 345,504$) with the 3-hourly NCEP time series ($n = 115,168$), the hourly time series were reduced to the associated 3-h values of the NCEP time series. To estimate the spatial variability of the data, a correlation of different spatial output nodes of the CAWCR data set (Fig. 4c) uses the entire CAWCR time series from 1980 to 2019 ($n = 350,640$).

This study further evaluates the impact of climate change on the wave climate according to the baseline chosen in the IPCC's fifth assessment report[47] by comparing the mean significant wave heights between the time periods 1986–2005 (historical; $n = 204,467$ undefined "NaN", or "Not a Number", values remain unconsidered) and 2081–2100 (end of the century) under RCP 4.5 and RCP 8.5 projections ($n = 230,203$ for both; undefined "NaN" values remain unconsidered). These projections are part of the CAWCR repositories[71]. The data is computed with the same tools as the hindcast data. Yet, the global grid is coarser with spatial increments of 1.0° (60′) and a temporal resolution of 6 hours. The RCP 4.5 and RCP 8.5 computations for wind waves use surface wind projections of 8 atmospheric models[71].

As the CAWCR data set has the output node closest to Fuvahmulah, boundary conditions for the numerical models used in this study were derived from the CAWCR data node north-west of the island at 0.0° latitude, 73.2° longitude (see Fig. 1). The boundary conditions come from the entire, available hindcast data of

**Table 1 Overview on wave data repositories. Data repositories considered for the local wave climate analysis.**

| CAWCR | | |
|---|---|---|
| Designation | Hindcast | Hindcast | Hindcast |
| Time series | 1979–2010 | 2011-01 to 2013-05 | 2013-06 to present |
| Wave model | WAVEWATCH III v4.08 | | WAVEWATCH III v4.18 |
| Wind forcing | NCEP CFSR hourly winds (Reanalysis) | NCEP CFSv2 hourly winds (Reanalysis) | NCEP CFSv2 hourly winds (Reanalysis) |
| Grid resolution | 0.4° (24′) | 0.4° (24′) | 0.4° (24′) |
| Temporal resolution | Hourly | Hourly | Hourly |
| Spectral output | 3683 points | 3683 points | 3683 points |
| Initializing Phase | 1979-01 | – | – |
| Partitions | 4 | 4 | 4 |
| Citation | CAWCR[65] | CAWCR[65] | CAWCR[65] |
| Extra | RCP 3.5, RCP 4.5 and RCP 8.5 forecast[71] Nested Australian and Western Pacific subgrids | | |

| | NCEP-EMC | | ECMWF ERA5 |
|---|---|---|---|
| Designation | Reanalysis | Hindcast (Phase 2) | Production Hindcast |
| Time series | 1979 to 2009 | 2009-02 to 2019-05 | 1979 to present |
| Wave model | WAM[a] | WAVEWATCH III v5.08 | |
| Wind forcing | NCEP CFSR hourly winds (Reanalysis) | NCEP GFS 6 h winds (Forecast) | IFS atmospheric model (CY41R2) |
| Grid resolution | 0.5° (30′) | 0.5° (30′) | 0.5° (30′) |
| Temporal resolution | 3 h | 3 h | Hourly single levels |
| Spectral output | 2D at each node | 2D at each node | > 2000 points |
| Initializing Phase | – | until 1979-01-04 | – |
| Partitions | 23 | 23 | 3 |
| Citation | NCEP-EMC homepage[b] | NCEP-EMC homepage[b] | C3S Climate Data Store[c] |
| Extra | Arctic Ocean and 13 nested subgrids | Arctic Ocean and 7 nested subgrids | 4D VAR data assimilation |

[a]Adapted by ECMWF within the IFS documentation, no. CY41R2[69]
[b]Polar.ncep.noaa.gov/waves/hindcasts/, last accessed 2020-03-22
[c]see ERA5[70]

significant wave heights $H_s$ and associated peak directions $\theta_p$ between 1979 to 2020-02. The analysis of peak directions $\theta_p$ for each season introduces the range of peak directions $\theta_{p,r}$. This range comes from a discrete probability distribution of the entire time-series and contains the directions, which combined have the highest 33% occurrence probability (see Fig. 5).

In general, the performance of global ocean wave models is good in the Indian Ocean when compared to buoy data[64,72]. The comparison to SRA measurements shows that wave data from global models occasionally underestimates wave heights, due to their relatively coarse spatio-temporal resolution in order of kilometers and hours. This improves with higher spatio-temporal resolutions[73]. Other studies have shown that underpredicted wave heights are especially prominent in extreme events[74]. In addition, non-linear effects can mitigate increased flood risk associated with sea-level rise[75]. Since 1992 and in contrast to sea-level rise, wave heights have only increased by about 0 to ~1 cmyr$^{-1}$ for the Maldives according to a most recent study[76] (see also Fig. 5f). In front of this background, wave heights under the respective RCP scenarios are conservative estimates of hazard levels, as the ocean wave models have a resolution of 1.0° and 6 hour time steps versus 0.4° to 0.5° spatial and hourly or 3 hour temporal resolution for the hindcast models. However, even when (1) combining today's wave heights with a projected sea level rise of about 0.4 m to 0.8 m and (2) comparing it against the topographic height of Fuvahmulah with ~3.5 m above sea level, both factors will lead to more frequent extreme events. In this context, on Fuvahmulah, historically rare extreme events will become common by 2100 under all RCPs[2,3].

**Numerical wave models.** With the wave climate as boundary conditions, numerical wave models can reconstruct waves and wave-induced currents and thus give insights into sediment transport processes on the reef under different scenarios (for example with or without seaport infrastructure). This study uses two numerical models for different purposes. First, D3D helps to study the general sediment transport processes on the fringing reef platform under the characteristic wave climate of the region. The phase-averaging wave module of D3D uses wave spectra in its governing equations and combines them with sediment transport formulations[77]. To study the influence of the port infrastructure on the adjacent coastal areas, this study uses the depth-integrated (2DH) Boussinesq-type model BOSZ[63]. The phase-resolving wave model can directly capture wave transformation and secondary processes such as wave setup and recirculation over the reef. It is therefore suitable to study the current-induced processes in the study area in detail. While the computation of direct sediment transportation is challenging, the general wave-induced current velocities or shear stresses from wave-induced currents serve as the driving forces behind the most prominent bedload and suspended sediment transportation formulations[77]. Thus, acceleration, deceleration and presence of current fields are direct indicators for sediment accretion and sediment pick-up.

D3D has been used successfully for calculating sediment transport and demonstrated good performance in both experimental and hindcast studies[78]. The model is in general capable to outline the morphodynamic processes on the Fuvahmulah reef. To remain numerically stable and efficient in terms of computational resources, the settings in this study contain the following simplifications: the model uses a relatively coarse rectilinear grid off-shore, with a refined resolution of $\Delta x = 67$ m, $\Delta y = 15$ m around Fuvahmulah. To study the morphodynamic processes around the island, D3D uses both the WAVE and the FLOW module. The reef depth and island height are based on topographic and bathymetric surveys within the field campaigns, however the implementation in D3D is idealized: the off-shore section has a constant water depth of $h = 100$ m. The fringing reef has a constant depth of $h = 4$ m, which increases on the southern reef platform to $h = 17.5$ m. In the model, the entire island Fuvahmulah lies at 3 m above mean sea level (MSL). The off-shore reef south of the island has a similar shape in the model compared to the real reef. However, to provide enough cells to transport sediment, the virtual fringing reef in the model is wider than the physical reef around Fuvahmulah. To evaluate sediment transport pathways around the island, the reef was covered with a 1 m layer of erodible sediment at the start of each computation. The underlying assumption is that the reef platform is the main

sediment supplier for the island[5,13]. While the simplifications allow for a larger set of computations, the implementation is still sufficiently detailed to capture the very general interaction between bathymetry, waves and sediment on the reef.

The following simulations in D3D use a PM-spectrum with $H_s = 2.3$ m, $T_p = 17$ s and a peak direction of $\theta_p = 202.5°$ and $\theta_p = 135°$ respectively. Despite the complexity of sediment transport on coral reefs[79], for a first estimation, D3D runs only with the default (suspended and bed-load) sediment transport formulations[77]. All simulations use the same sediment properties: the sediment layer thickness is $d_{sed} = 1$ m, sediment density is $\rho_{sed} = 2.0$ g cm$^{-3}$ (based on in-situ measurements) and sediment grain size (diameter) is $D = 200$ μm. The morphological scale factor is $f_{MOR} = 2$.

The reef bathymetry in the 2DH model comes from records of a single-beam, dual-frequency echo-sounder (Dr. Fahrentholz LituBox 15/200) taken during the first field campaign in 2017. The echo-sounder was able to show water depths up to ~ $80-100$ m. However, as water depths increase rapidly beyond the reef plateau, records from the echo-sounder exceeding the maximum depth of the Fuvahmulah reef plateau at the southern tip of ~ $20-25$ m were discarded. The minimum depth of the echo-sounder was 4 m due to the draft of the boat. The 2017 Photoscan model provides the nearshore (reef) bathymetry for water depths between mean low tide to mean sea level and the island's coastal topography (including the coastal ridges). The Generic Mapping Tool[80] interpolates the combined elevation and bathymetry data on a 7.5 m by 7.5 m grid, which was also used as a computation grid in the BOSZ model. Offshore waters are truncated to $h = 60$ m. Similar to the computations in D3D, the hydrodynamic boundary conditions are characterized by a TMA-wave spectrum with a significant wave height of $H_s = 2.3$ m, a peak period of $T_p = 17$ s and peak directions of $\theta_p = 202.5°$ and $\theta_p = 135°$, respectively.

Since this study focuses on the wave driven longshore currents, ocean circulation around Fuvahmulah and the tidal current velocity was disregarded for sediment transport processes (for further information, see Supplementary Materials). This can be considered standard practice when regarding longshore sediment transport and confirms that longshore sediment transport mainly depends on incident waves, while tidal and ocean currents are of minor importance[81].

Both numerical approaches reveal the impact of the regional wave climate on sediment transport and thus on erosion. However, some limitations apply to the results of the numerical models: the wave input is based on storm wave heights of two peak directions, while moderate seas also have an impact on sediment transport and beach restoration[82]. Furthermore, D3D is able to reconstruct experimental and hindcast data of sandy beaches[78], but application on reef island morphodynamics is scarce[83]. For Fuvahmulah, D3D does not implement port structures in its bathymetry, to mimic the natural sediment pathways. The coarse resolution and the idealized bathymetry give a general concept of sediment movement on the reef rather than quantitative volumes of transported sediment. Still, the coarse model is able to reconstruct typical, process-based morphodynamic behavior for an environment like the Fuvahmulah reef[5]. The 2DH wave model comprises a more detailed study with and without the port structure. However, the results must be interpreted with caution: the 2DH approach calculates current velocities but does not yield transported sediment volumes. Erosion and sedimentation depend on more than only current velocities[5]. For example, research on organism scale shows that sediment transport on coral reef bottoms is complex[79]. However, current velocities are the main driver behind sediment transport phenomena and velocity gradients hint at hotspots of erosion and sedimentation[77]. With this in mind, the computed wave-induced currents are able to illustrate the role of the southeastern reef area under the given wave climate for the sediment transport around Fuvahmulah. They also facilitate a process-based understanding of the seaport's impact on the adjacent coastline, even without increased sea level or varied wave heights.

**Societal aspects**. The analysis of the societal dynamics is based on a mixed-methods approach to convey the social and governance aspects influencing coastal protection and climate change related topics. The approach encompasses (1) a literature review and content analysis of climate change adaptation and coastal protection literature in the Maldives, (2) two surveys with the local population, which are complemented by (3) semi-structured interviews with relevant actors involved in coastal management on the local as well as on the national-level. All participants gave their voluntary consent to participate and have been informed about the objective of the research. The collected data of the second survey and the interviews were analyzed via a structuring content analysis in the software program MAXQDA[84]. This content analysis categorizes the interview's responses and those of open questions in the surveys into codes and subcodes, developed in a concept-driven and data-driven approach[85]. The concept-driven or deductive approach follows a categorization structure established before the field campaign. However, the possibility to respond to open-ended questions also requires post-processing of the responses based on their content – resulting in a data-driven approach.

The first survey and interviews took place from March to April 2017 and were synchronized with the first field campaign, measuring topographic and hydrodynamic data. Intermediate results from both work packages and the associated discussions led to interdisciplinary insights, considered in the second survey (January to February 2019).

**Literature review**. The literature review was conducted by content analysis. The analysis focused on publications relevant to climate change adaptation and coastal protection in the Maldives, including peer-reviewed papers, legislation and regulations, as well as gray literature. Legislation and regulations were primarily identified with the support of actors in the Maldives coastal governance system through the semi-structured interviews, ensuring their relevance and completeness. The literature review also encompasses previous EIAs on coastal projects in the Maldives, as these are required to list the legislations and regulations that are applicable to the projects. Most legislation and regulations are available in an English version on the website of the respective ministry, however, documents in the local language Dhivehi have not been considered. The analyzed gray literature covered EIAs, available through the Maldives Environmental Protection Agency (EPA), as well as reports from the Government of the Maldives, Non-governmental Organizations (NGOs) and intergovernmental organizations, such as the United Nations Development Programme (UNDP) Maldives. Special attention was paid to reports, EIAs, and ESIAs concerning Fuvahmulah's seaport or explicit climate change adaptation measures on the island (see Table 2). The documents were analyzed via a qualitative content analysis[85], which focused on the following thematic codes or categories in the context of the coastal governance system in the Maldives: (1) distribution of responsibilities among actors in the context of coastal protection, (2) history of coastal protection measures, (3) framing of coastal protection in the climate change discussion and (4) framing of coastal protection in the climate change discussion framing of coastal protection in the climate change discussion. The analysis disclosed knowledge gaps in the literature and demonstrated the need for an exploratory and qualitative research approach. Therefore, the study also contains population surveys and semi-structured interviews.

**Survey**. Two surveys with semi-standardized questionnaires were conducted in Fuvahmulah. The first survey focused on four topics: (1) the local's understanding of the environment and the coast, (2) their perception of environmental issues with a focus on climate change impacts and comprehension of responsibilities for action, (3) the locals' perspectives on community life, and (4) their perspectives and attitudes towards coastal protection. The second survey focused on: (1) the locals' perceptions on community life, (2) their potential for engagement in environmental and coastal protection activities, and (3) how politics shape the way coastal issues are addressed.

The surveys involved going from door-to-door in all eight wards of the island and were supported by local research assistants. The sample size of each survey is equivalent to about every eighth household on the island, while household selection was done by previous randomized sampling, yielding an unbiased representation of local households. Interviewees were not working in local coastal management. The first survey had 116 participants between the age of 14 and 86, while the second survey was answered by 98 inhabitants of Fuvahmulah between 14 and 84 years old. The lower threshold of 14 years results from the fact that climate change is a long-term development and coastal protection measures will have an impact for several generations. As in social science studies on future societal development, we have accordingly extended the age limit to 14 years. Furthermore, this decision is based on the experiences made on the island and made in consultation with the local research assistants – people over 14 years of age start to participate in environmental activities without the guidance of their parents and are therefore considered to be sensitive to the environmental situation of Fuvahmulah. With this in mind, and to avoid bias[86], adolescents are considered capable of giving consent to answer the questionnaire themselves. As participants were not divided into experimental groups, the responses of all participants in the survey remain anonymous. However, to further respect their privacy, this minimal-risk study does not include direct verbatim statements from participants aged 14–18.

The questionnaire of each survey contained closed and open questions (see supplementary materials). For closed questions, the questionnaire provides predefined answers, which were analyzed with methods of descriptive analytics. In contrast to closed questions, open questions require the participant to respond freely to the question. The answers were categorized and coded[85] after the survey to quantify the results and identify important response patterns. In both surveys, the individual interview time varied strongly, ranging from 20 to 90 min. This can be explained by the qualitative approach of the surveys, utilizing open questions and the varying length and depth of answers which depended, for example, on the interviewee's interest in the topics.

**Interviews**. In each field trip, the interviews of Fuvahmulah's population were complemented by semi-structured interviews with actors on the national and local level being involved in the management of coastal issues in the Maldives. The selection of interview partners is based on their profession and their expertize on the subject (purposive sampling). Using this non-random technique was necessary due to the limited number of actors in coastal governance, while obtaining the required information to compare the varying opinions and interests among those actors regarding coastal protection.

On the national-level, interview partners were representatives of the Ministry of Environment and Energy (renamed to Ministry of Environment in 2018), members of non-governmental and intergovernmental organizations that are active in the climate change adaptation and coastal management fields, as well as local researchers. Interviews with non-government actors gave in-depth and critical

**Table 2 List of reports, EIAs and ESIAs on the environment, coastal protection, and adaptation on Fuvahmulah. Laws and policy regulations on coastal risks and sea-level rise in the Maldives are assessed according to the framework of Gussmann and Hinkel[40].**

| | Title & Issuing ministry, organization or company | Year |
|---|---|---|
| Title: | Foahmulaku Beach Erosion Survey & Coastal Protection Report[44] | 2006 |
| Issued by: | Ministry of Environment and Energy (MEE) | |
| Title: | Survey of Climate Change Adaptation Measures in Maldives | 2011 |
| Issued by: | Ministry of Housing and Environment (MHE) and United Nations Development Programme (UNDP) | |
| Title: | Wetland Conservation & Coral Reef Monitoring Project | 2011 |
| Issued by: | Ministry of Housing and Environment (MHE) and Climate Change Trust Fund (CCTF) | |
| Title: | Environment & Social Assessment & Management Framework Climate Change Adaptation Project[43] | 2014 |
| Issued by: | Ministry of Environment and Energy (MEE) | |
| Title: | EIA Report For Coastal Protection at Gn. Fuvahmulah, Maldives[51] | 2016 |
| Issued by: | Maldives Energy and Environmental Company (MEECO) and Royal HaskoningDHV | |
| Title: | Report: Coastal protection at Gn. Fuvahmulah[50] | 2016 |
| Issued by: | Royal HaskoningDHV | |
| Title: | Review of the Draft ESIA Report for the Fuvahmulah Coastal Protection | 2016 |
| Issued by: | Netherlands Commission for Environmental Assessment (NCEA) | |
| Title: | Review of the ToR for the ESIA Fuvahmulah Coastal Protection, Maldives | 2016 |
| Issued by: | Netherlands Commission for Environmental Assessment (NCEA) | |
| Title: | Environmental and Social Impact Assessment–Ecotourism Facility Development | 2017 |
| Issued by: | Sandcays, Ministry of Environment and Energy (MEE) | |

perspectives on the way environmental and coastal issues are addressed by government entities. This information is especially important as it helps to identify informal institutions, such as norms and attitudes towards specific responses to coastal problems that become effective and influence the way coastal problems are addressed.

On the local level, interview partners are current and former members of Fuvahmulah's council and representatives of non-governmental organizations.

In summary, the results of this study are based on 32 interviews with 18 interviews with actors on the national-level (10 government, 8 non-governmental or intergovernmental organizations and researchers) and 14 on the local level (8 government, 6 non-governmental organizations). The interviews of the first field trip addressed (1) societal dealings with environmental problems, (2) coastal management with focus on decision-making processes, (3) community involvement and public awareness, and (4) politics and climate change. The second interview guide focused on (1) the community in the Maldives, (2) community engagement, and (3) coastal governance. The interviews lasted between 30 and 120 minutes. Owing to the political situation in the Maldives and the small number of experts in this field, not all interviewees agreed to be recorded, but memos and notes capture the interview's statements. Recorded interviews were transcribed before analysis. In addition, the interviews were fully anonymized – except for the ministry officials – in order to protect the identity of the informants.

**Reporting summary**. Further information on research design is available in the Nature Research Reporting Summary linked to this article.

## Data availability
The field data[87] from the aerial surveys and bathymetry for numerical modeling is publicly available through https://doi.org/10.5281/zenodo.4304049. Wave climate data is publicly available from the respective services and homepages of CAWCR (CSIRO Data Acess Portal: https://data.csiro.au/collections/collection/CIcsiro:39819), ECMWF (Copernicus Climate Data Store: https://cds.climate.copernicus.eu) and NOAA (WAVEWATCH III®Hindcast and Reanalysis Archives: https://polar.ncep.noaa.gov/waves/hindcasts/). For further information of the climate data, see also Table 1. Source data supporting the social sciences' finding are provided within the Supplementary Information File of this paper. Other raw data (for example interview transcripts) on the social sciences part that support the findings of this study are protected and not publicly available as they contain personal information that could compromise research participant privacy. Other data from the social science part can be made available from the social science team (B.M.W.R. and A.H.) upon reasonable request.

## Code availability
The aerial images were processed with the Structure-from-Motion MultiView Stereo (SfM-MVS) algorithm of Agisoft Photscan Pro (version 1.4.5., build 7354). Except for the

bathymetry of the numerical model, geo-data was processed with Quantum GIS 3.10 A Coruña and 3.16 Hannover on Ubuntu Linux 18.04 and 20.04 and/or Python modules rasterio v.1.1.1. to v.1.1.3. together with the module Fiona v.1.8.13. The Boussinesq Ocean and Surf Zone (BOSZ, version 01-2019) model is available through Volker Roeber upon reasonable request. The Generic Mapping Tool (GMT, version 5.4.5) was used to process the geo-data to be used as bathymetry in BOSZ. Other simulations on the morphodynamics around the reef island was carried out with Delft3D. Delft3D is open-source and available through Deltares (oss.deltares.nl/web/delft3d). In the Delft 3D Suite, we used the modules Deltares FLOWD3D 6.02.13.9162M, SWAN III 40.72ABCDE. Other scripts for data post processing and visualization, are written with the open-source software Python 3.7 in Jupyter Notebooks, using the modules Numpy 1.17.4, Matplotlib (pyplot) 3.1.2, Xarray 0.12.1 with NetCDF 1.5.3, Pandas 0.25.3, cartopy 0.18.0 and cdsapi 0.1.3 and later versions. Cartopy uses OpenStreetMap data (available under the Open Database License, see openstreetmap.org) as well as Stamen open-source maps. Data of the second household survey and interviews were analyzed with the software MAXQDA.

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

## Acknowledgements

This study took place in the project "Dealing with change in SIDS: societal action and political reaction in sea level change adaptation in Small Island Developing States (DICES)", grant no. SCHL 503/17-1 (C. Gabriel David, Torsten Schlurmann) and RA 585/19-1 (Arne Hennig, Beate M. W. Ratter). The project is framed within the priority programm (SPP-1889) – regional sea level change and society of the German Research Foundation (Deutsche Forschungsgemeinschaft, DFG). Volker Roeber acknowledges financial support from the Isite program Energy Environment Solutions (E2S), the Communauté d'Agglomération Pays Basque (CAPB) and the Communauté Région Nouvelle Aquitaine (CRNA) for the chair position HPC-Waves; as well as the support from the University of Hawai'i at Mānoa for the Affiliate Graduate Faculty position. Zahid was employed as Deputy Director General Climatology at the Maldives Meteorological Service (MMS) at the time of the project, but retired from his position. The authors would like to thank Ali Ahmed, Pablo Ballesteros, Tatiana Ivanova, René Klein, Nina Kohl, Manò Schütt, Ibrahim Shiyan (Panda), Jailam Zahir as well as Marion and Uwe Zander for their help in the field campaigns. In the Maldives, the authors were supported by the Maldives Meteorological Service (MMS), Fuvahmulah Island Council and Fuvahmulah DIVE School. Pablo Ballesteros and René Klein are staff of Ludwig-Franzius-Institute, Leibniz University of Hanover (LuFI-LUH) and supported the field campaign with their technical support. Nina Kohl and Manò Schütt were student assistants at LuFI, helping to record data on Fuvahmulah. Ali Ahmed and Jailam Zahir are locals from Fuvahmulah and assisted the in the coastal surveys (Ali) and in the interviews and household survey (Jailam). The authors also thank Tilo Schöne of GFZ Potsdam, providing the SRA data as well as Jean Bidlot (ECMWF), Mark Hemer (Australian Commonwealth Scientific and Industrial Research Organisation, CSIRO) and Todd Spindler (NCEP-EMC) for their help in accessing climate reanalysis data. Also, Elisa Casella and Alessio Rovere gave valuable insights and feedback to UAV-based photogrammetry, while Tobias Kersten helped with geodetic questions before the third field campaign and with post-processing the GNSS data. Furthermore, the authors acknowledge Jannek Gundlach for his feedback on and Astrid Kartes and Jonas Briese for their assistance with Delft3D. Both, Astrid and Jonas, were students at LuFI during the project.

## Author contributions

C.G.D. conducted the natural science/engineering part of field campaigns, research and analysis, while A.H. carried out the corresponding social science part. V.R. developed the BOSZ model used in this study and supervised the numerical modeling. T.S. and B.R. (co-)designed the research project, were responsible for funding resources and reporting and provided guidance throughout the entire research. C.G.D. wrote the manuscript with input from A.H., while B.R., V.R., T.S. and Zahid edited and contributed to the final manuscript. Zahid helped in designing the research project, coordinating the field surveys and provided a local perspective to the evaluation and analysis.

## Funding

 This study took place within the German Research Foundation's (DFG) priority programm "regional sea level" (SPP-1889).

## Competing interests

The authors declare no competing interests.

## Ethics statement

All participants gave their informed consent and we complied with all relevant ethical regulations. Ethical affairs have been handled and declared during the funding phase of the project. Ethics committee and institutional review board approval was determined not to be required for this project.
