## [Peer Review File · Nature Communications]

Considering socio-political framings when analyzing coastal climate change effects can prevent maldevelopment on small islandsReviewer comments, first round –

Reviewer #1 (Remarks to the Author):

Climate change induced effects or maldevelopment:
small islands and conflicting attribution of root
causes

This is an interesting manuscript developed in an exotic and vulnerable place in the world:
Fuvahmulah in the south of the Maldives.

This manuscript is mainly focused on the coastal erosion assessment and the impacts on the
natural physical environment. It is a 100% technical manuscript. I must say that after reading the
section "Governmental and societal framing of coastal development," looks strange into the
document (doesn't fit - it's forced).

Introduction, study area, and results are ok. They are descriptives and shows the situation in a
clear way. However, I think the discussion can be more robust. Authors can compare their results
with other islands!

My suggestion here is simple: try to focus on the technical aspects of the manuscript and leave
behind the management component (or improve it - significantly). This component is weak and
generates much noise in the text (it is forced - imposed).

Regards,

Reviewer #2 (Remarks to the Author):

The major claims of the paper are that:

1. Sometimes there is disagreement between local and governmental knowledge concerning how
certain adaptations should be implemented, and oftentimes local residents know more about the
possible consequences of certain (mal)development strategies than non-local planners do not
understand or prepare for. This is important because anthropogenic interventions including coastal
structures and other types of structural mitigation and adaptation can cause additional pressures
on already stressed environments. If such decisions for specific adaptation structures are being
implemented without the consideration of local knowledge and other environmental conditions, the
possibility of implementing maldevelopment that further negatively affects the environment
increases.

2. It is difficult to develop sustainable development in islands that are increasingly threatened by
climate change, but this can be made even worse when (mal)adaptive decisions result in even
greater impacts to already vulnerable areas.

3. Successful sustainability requires the balancing of socioeconomic and political factors within the
natural environment and ongoing changes to the environment due to climate change. The paper
demonstrates that an interdisciplinary approach can provide insight as to how small-islands'
adaptive capacities are typically impaired in dealing with climate change related changes, reveal
the structural challenges of top-down processes, and disclose the potential of local knowledge to
overcome maldevelopment.

Major concerns about the paper – While this paper addresses an important topic in an fairly

interdisciplinary way (which is still missing in a lot of adaptation research), this paper is not well organized and the way the results are reported are frustrating and difficult to process.

This occurs for two reasons: 1) there are two very different sets of methods being addressed, and 2) the authors did a poor job of synthesizing the results from those methods.

Methods

There are also some issues when explain the methods, especially the social science aspects. First, the methods associated with the social science aspect of the work or not well described or well-organized. Problems associated with those methods are described below:

Method 1 - describe the social science methods, including interviews with government officials and surveys distributed to the general population.

Critical information concerning the three main social science methods missing from the paper (which affects the ability for this work to be reproduced) is listed below:

1. Lit review

How was this done? There is literally one sentence describing the methods for this aspect of the paper and the supplemental document does not go into much more detail. How was this done? What sources were used to identify laws and regulations? There are also results from an Environmental Impact Assessment (EIA). Why are these here. Are these results the authors collected for the lit review?

2. General population surveys

What were the questions on the survey?; How were the questions analyzed or coded?; Were any of these participants also interviewed as people who work on local coastal management? This is not explicitly stated in the methods or the supplement. For example, the results section provides a quote from an interviewee that discusses the protective functions of the reef immediately after it reports that "islanders feel safe because of the coastal ridges around the island." Does this apply to interviewees or survey responses?; The paper gives no context as to what people were specifically asked that can be used to directly demonstrate how that information was used in comparison to the physical modeling and DEMs when making arguments that local knowledge better understood oceanic-climatic pressures than high level officials.

3. Interviews with government officials

Methods do not provide enough detail concerning the interviews and how they were conducted. What were the interview questions? How long did they last?

Did the managers indicate on maps where there were issues associated with specific oceanic-climatic pressures or were there key words or place names that were given? How were these chosen? Why did the national level include government and non-government officials? What does that add to the study? How were the interview results analyzed? If qualitative, how were key words coded or highlighted?

Method 2 describes the physical modeling of oceanic-climatic pressures of interest, including modeling of erosion adjacent to the new harbor and sediment dynamics on the reef. Models describe:

Sediment Transport and relocation

Identifying erosion using satellite imagery

Near-shore wave models

DEMs

These sections were much clearer, but the authors disproportionately focused on the modeling results and provided much more detail than the social science activities.

Results:

There are several issues associated with how the paper is organized and how the results are presented. The same ambiguity of the social science methods also occurs in the results.

It is never made explicitly clear when specific results for the social science methods are being

presented. For example, when discussing the literature review regarding relevant laws and regulations associated with coastal management in the results, the authors do clearly explain that these are the results from the lit review. Furthermore, the discussion of the law and regulations is weak and interspersed with results from the surveys.

The results are also organized in a way that does not clearly present the social science results, the physical modeling results, and then the synthesis of both. For example, some of the survey and interview results are reported on the physical modeling sections. While this is meant to demonstrate that local knowledge matches physical modeling and serves as an attempt of synthesizing the results, these sections are disjointed, as the synthesis information is typically provided at the end of the long modeling results sections, and are not tied back together.

A way to potentially improve the structure of the paper would be to 1) present the physical modeling results that demonstrates where there are certain areas that have higher sediment transport or erosion potential that are based on maladaptation, 2) present the results about the laws and regulations associated with those particular pressures from the social science perspective, and 3) include a synthesis section of the results that demonstrates how the physical models match those same results were information and local knowledge that is elucidated from the social science methods.

Discussion

Once connections between physical and social science results have been made clear, they should be further discussed in the discussion section. Additionally, a key discussion point that is incredibly impactful is "This study displays the exploitation of the climate change narrative to cover for maldevelopment on a local level." However, while this claim has potentially substantial implications, it is not well introduced as one of the main drives of the research, nor it is clearly explained how that conclusion derived from the project results. This aspect of the study does not appear anywhere else in the document -if it is, it needs to be more clearly stated, because this statement does address a less well-researched topic. While several studies demonstrate the necessity to incorporate local knowledge into adaptation planning to avoid maldevelopment (a few are listed below the summary), few that I know of purposely examine whether maldevelopment is being explained and justified using the exploitation of climate change narratives by national level officials. IF this was the major claim for this paper, it is not well framed throughout the entire paper and should be refocused on this issue.

Summary: This research attempts to implement a truly interdisciplinary research plan, with the goal of synthesizing physical and social science methods to substantiate and justify the need for incorporating local knowledge into adaptation planning as a way to avoid implementing maladaptation that is potentially destructive in the future. However, due to major organizational issue with the paper and a lack of detail, especially concerning the social science aspects and too much focus on the physical modeling, the paper would benefit from major changes.

Karen Elizabeth McNamara & Ross Westoby (2011) Local knowledge and climate change adaptation on Erub Island, Torres Strait, Local Environment, 16:9, 887-901, DOI: 10.1080/13549839.2011.615304

Reineman, Dan R., Leif N. Thomas, and Margaret R. Caldwell. "Using local knowledge to project sea level rise impacts on wave resources in California." *Ocean & Coastal Management* 138 (2017): 181-191.

Magnan, A.K., Schipper, E., Burkett, M., Bharwani, S., Burton, I., Eriksen, S., Gemenne, F., Schaar, J. and Ziervogel, G. (2016), Addressing the risk of maladaptation to climate change. *WIREs Clim Change*, 7: 646-665. doi:10.1002/wcc.409

Reviewer #3 (Remarks to the Author):

Overall comments:

Title: Climate change induced effects or maldevelopment: small islands and conflicting attribution of root causes

Very good topic. However, major revisions are needed to improve the quality of the paper

Several acronyms are listed in the manuscript. Authors need to give the full form of abbreviations in a tabulated form after the Abstract.

Abstract

Well written Abstract. However, revisions are needed.

Authors need to give information on the coastal length of the Fuvahmulah area in the abstract.

Also, what interdisciplinary perspective has been used for this study (Survey and DEM models).

They should also need to give this information in the abstract.

How are the results useful to the policy and decision makers? Should need to include this information as well.

Introduction

Lines from 25th to 30: References are needed for those statements.

Authors heavily depended on grey literature rather than synthesizing the global literature regarding small islands -coastal erosion or vulnerability. One paragraph is needed regarding this.

Study site

What is the length of the Fuvahmulah coastal area?

How much of population of this area?

How much coastal area was already eroded in the last two decades?

What is the erosion rate in this area?

What are the socio-economic and political characteristics of this area?

Information is needed regarding above mentioned queries for this particular section.

Results

Figure 2 is a nice representation of coastal erosion.

What is the average rate of coastal erosion per year in this site?

How many km of coastline already crumbled into the sea?

Information/explanation is needed?

Impacts on the natural physical environment

Wave climate of the southern Maldives

Well written part

Discussion

Good. But can be expandable with a comparison of some recent global examples.

Conclusion

Good, but too lengthy. Revisions are needed.

What are the study limitations? Information is needed.

Methods:

Why has the survey taken from the age group of 14 ?

Is there any particular reason?

Where is the ethical approval number?

How may interviews have been taken?

Information is needed for above mentioned queries.

DEM and Numerical wave models

Very well explained and well written parts.

Reviewer #4 (Remarks to the Author):

This paper applies an interdisciplinary approach which combines scientific and sociopolitical research to understand if erosion problems in the Maldivian reef island Fuvahmulah are due to climate change or human interventions (maldevelopment).

The problem of erosion in this island is analyzed from different perspectives (political decisions, local population perception, process-based methods), and it has discovered that island community and national government have a diametrically opposed explanation about the erosion problem. Measured coastline erosion and simulations of the sediment patterns corroborate the perception of the local actors.

As very positive aspect of your work is the integration of different data sources:

- The analysis of the environmental governance in the Maldives that are financially supported by different international organizations
- Semi-structured interviews with actors relevant for coastal protection management
- Surveys amount the local inhabitants that allow to find that a low percentage of population is involved in the development projects in the island. Very interesting to highlight the knowledge of the local people about the protective function of the heylhi.
- Measured erosion rates from DEMs produced with the SfM approach
- Characterization of wave climate in the Southern Maldives differentiating between the dry and wet season for the present climate and the projected future changes.
- Application of two process-based models to analyze the sediment movement for two main characteristic storms and the wave-induced currents with and without the seaport.

I think the main conclusion about the study is the people's participation in infrastructure projects would have avoided or reduced the erosion problem generated by the construction of the port because their knowledge about the local environment would have contributed to a better design. It is worthy to emphasize the important of local knowledge when interventions are projected.

On the hand, it seems that the purpose of your results is to highlight that climate change is not always the cause of erosion in Maldives as the national government claims and they use this reason to obtain international funding. Although this might happen, the case you have studied represents the local impact of an infrastructure that was built more than a decade ago and cannot be use an example of false adaptation measure or as an adaptative capacity which deteriorates the adaptation to climate change. I am not sure if you also try to demonstrate that erosion due to climate change (mainly due to sea level rise) will not be a problem is this island because people have the perception of safety against climate change impacts. The climate change effects may not be visible nowadays being one reason why people did not mention it in the survey, and also, because the survey was very focused on the erosion problem related to harbor. It seems that you try to extrapolate what happen with the construction of the port with potential erosion problems along the coast of the island (lines 269-270).

The constructions of the port was demanded by people from Fuvahmulah for decades because previously to the construction, transporting people, goods and cargo were made by small fishing boats (dhonis) from the island to large ships offshore with many boats capsized and people died. I think it is not complete accurate to give maldevelopment the same connotation as maladaptation because both increase vulnerability or diminish welfare. Seaport has been a very important coastal infrastructure for the economic and social development of the island. Besides, although the harbor acts as a barrier to sediment transport, it does not mean that climate change will exacerbate the impact, a local impact assessment should be performed to verify this supposition. Climate change could affect not only the wave height but also the wave period and direction which is a variable more relevant in sediment transport.

I consider that the meaning of several concepts: coastal infrastructures (which can increase quality of life), adaptation measurements (being hard engineering solutions one option, luckily there is a change of mind to a more environmental interventions) and natural self-adaptation measures (being the reef and coastal vegetation natural coastal protections, not the sediment transport

itself) are mixed. It is not comparable an infrastructure as a port with different options to respond to an increased coastal risk and sea level rise (as protection, accommodation, ecosystem-based adaptation, very different from engineering-type hard-coastal protection systems).

Although your work demonstrates that negative consequences from anthropogenic interventions could be similar to those to climate change, the study does not constitute a material to be published in *Natural Communications* journal. Although an interdisciplinary approach has been applied, most of the work is concentrated on demonstrate that the harbor infrastructure is the source of erosion on the coast adjacent to the north of the seaport.

You use the term "top-down implemented infrastructure projects" along the text and even deduce, from the analysis of the problem of erosion close to the harbor, structural challenges of top-down processes. What does this term mean for you? As far as I know, bottom-up or top-down approaches refer to the way the climate impacts are analyzed. Bottom-up approaches usually start with a system or impact (such as a disaster), and then identify all of the underlying variables, processes or phenomena that play a role in shaping the outcome, understanding the nature of the risks before identifying the relevant drivers and hazards. While in top-down or scenario-led approaches, scenarios are generated using climate models and then incorporated into an impact model introducing a lot of uncertainty because it is difficult to integrate multiple interacting drivers and/or hazards (Zscheischler et al., 2018). It should be convenient to describe this term related with infrastructure projects and its drawbacks. Why does this approach usually follow? You should describe in more detail the implications of this approach and how could be improved for future applications.

It is true that any coastal infrastructure will generate disturbances in the natural system, but many different configurations could be studied analyzing the effect in the close beaches or vital ecosystems in the area and the select the less damaging. I think the maldevelopment of coastal infrastructures could be avoid incorporating in the design a study of the changes in the sediment transportation pattern (e.g., in this study case), but this is an aspect that is not usually taken into account. It is also convenient to include climate change (in the drivers or hazards, exposure and vulnerability) in the design of coastal infrastructures to avoid worsening climate change impacts. There could be a better way to implement "top-down implemented infrastructure projects".

The seaport was constructed in 2002, therefore, the erosion problem should have been an impact that evolves in time and becomes more evident as time passes, not just in the recent years. You calculated an erosion rate as the differences between DEMs made in 2017 and 2019. Only two measurements and without related with climate during this period could lead to false conclusions. Climate variability might explain the magnitude of the erosion rate in that period. Maybe some of the erosion problems close the seaport is due to port itself, which interrupts the sediment transport, but also could be attributed to sand mining (as the inhabitants of Fuvahmulah point to it as the most common reason of erosion in the island in the second survey attributed).

It is not clear for me how you apply a DPSIR framework (drivers, pressures, state, impact and response) in this work? Has this framework got specific steps within a methodology?

More specific comments about some sections of your study:

Regarding the wave climate characterization, it is not clear for me how you have calculated the changes in H_s (as the differences between the time series of the 3 RCPs at the end of the century (2090-2099) and the present time period (2010-2019)). Because the percentage of increase seems to be very high (see Hemer et al., 2013). Are you considering the different GCMs available at CAWCR repository? Are you calculating the ensemble changes of H_s ? Why did you consider only the period 2010-2019 and 2090-2099? The climatology needs at least 20 or 30 years to be calculated. I think mention future changes in significant wave height does not significant contribute to the study.

Although the simulations you have performed help to understand the effect of the seaport interrupting the sediment transport generated by the wave currents, you have only analyzed the transport in two characteristic storms. Storms usually have a high effect in the shoreline retreat,

but the shoreline can be recovered between storms being seasonal (as it seems to happen in the north of the island) and interannual changes natural oscillations of the shoreline and they do not necessary imply erosion. It is true that reef environments require special analysis due to the complexity of the environment. There are many references that have been improved the knowledge of the sediment transport in the reef environment and atoll island (most of them have been included in your references) which apply more sophisticated methods. I think your analysis is only for contrast the measured physical processes and the local knowledge about the effect of the seaport but it is not in the state-of-the-art.

Dear respected reviewers,

We would like to express our appreciation for your constructive comments and we believe that addressing these comments have improved the quality of the manuscript significantly.

Please find enclosed the detailed answers to the reviewers' comments as well as the corresponding actions addressed in the revised manuscript. Two revised manuscripts are submitted along with this rebuttal letter - one version allowing the reviewers to track changes and a clean version of the current manuscript.

With the kindest regards,

The authors.

Reviewer #1 (Remarks to the Author):

Comment 1.1:

"Climate change induced effects or maldevelopment: small islands and conflicting attribution of root causes

This is an interesting manuscript developed in an exotic and vulnerable place in the world: Fuvahmulah in the south of the Maldives.

This manuscript is mainly focused on the coastal erosion assessment and the impacts on the natural physical environment. It is a 100% technical manuscript. I must say that after reading the section "Governmental and societal framing of coastal development," looks strange into the document (doesn't fit - it's forced).

Introduction, study area, and results are ok. They are descriptives and shows the situation in a clear way. However, I think the discussion can be more robust. Authors can compare their results with other islands!

My suggestion here is simple: try to focus on the technical aspects of the manuscript and leave behind the management component (or improve it - significantly). This component is weak and generates much noise in the text (it is forced - imposed).

Regards"

Answer to Comment:

We appreciate the critics given by Reviewer #1 as we have been able to outline the core methodological approach and to discuss the main findings in our study. All technical aspects have been discussed and acknowledged at a coastal engineering conference in late 2019. Yet, the crucial question on the root causes of coastal erosion in the island are only partly of technical origin since they are deeply rooted in socio-political dimensions following top-down decision-making processes in the Maldives. Addressing these issues holistically within this manuscript and in an interdisciplinary research attempt was our main goal since we similarly

found that *“the ability of communities to prepare for and respond to prolonged or sudden coastal impacts is determined by socio-political and economic privilege”* - which is in line with the Editorial *“Sea change in coastal sciences”*, *Nat Commun* 11, 4601 (2020), DOI:10.1038/s41467-020-18333-8.

Following the comments of Reviewers #2 to #4, we have profoundly re-organized the structure and added new contents to the manuscript. We are confident now, to have improved to outline both the role and the significance of methodologies developed by our co-authors from the social sciences that is necessary to shed light on a sensitive coastal management problem in the Maldives. By reorganizing the manuscript and rewriting the discussion, we believe to have better synthesized and embedded the non-technical aspects in the manuscript. In addition, discussion on the methods were moved to the associated method sections, allowing for better readability and a better understanding of our main messages.

For further details, please see the other review comments.

Action taken for this comment:

- *Provision of underlying methods more detailed*
- *Role and significance of methodologies in social science outlined and integrated*
- *Reorganization of manuscript*
- *Enhancement of discussion and synthesis of results from technical and socio-political sciences*
- *Challenges of the current situation on the island addressed*

Reviewer #2 (Remarks to the Author):

Comment 2.1:

“The major claims of the paper are that:

1. Sometimes there is disagreement between local and governmental knowledge concerning how certain adaptations should be implemented, and oftentimes local residents know more about the possible consequences of certain (mal)development strategies than non-local planners do not understand or prepare for. This is important because anthropogenic interventions including coastal structures and other types of structural mitigation and adaptation can cause additional pressures on already stressed environments. If such decisions for specific adaptation structures are being implemented without the consideration of local knowledge and other environmental conditions, the possibility of implementing maldevelopment that further negatively affects the environment increases.

2. It is difficult to develop sustainable development in islands that are increasingly threatened by climate change, but this can be made even worse when (mal)adaptive decisions result in even greater impacts to already vulnerable areas.

3. Successful sustainability requires the balancing of socioeconomic and political factors within the natural environment and ongoing changes to the environment due to climate change. The paper demonstrates that an interdisciplinary approach can provide insight as to how small-islands' adaptive capacities are typically impaired in dealing with climate change related changes, reveal the structural challenges of top-down processes, and disclose the potential of local knowledge to overcome maldevelopment.”

Answer to Comment:

This is an excellent summary of the key challenges in order to successfully progress sustainable development in small islands. In parallel, small island developing states (SIDS) are contradicted to cope with climate change and sea level rise in order to determine adequate adaptation strategies that we think we have exemplified in our study to a large degree. We are thankful the main findings have been clearly identified and appreciated by Reviewer #2.

We highly appreciate the time and efforts taken by the Reviewer and we are very thankful for the constructive, fair and helpful comments. We tried to reply to the issues raised to the best of our abilities.

Action taken for this comment:

- None

Comment 2.2:

“Major concerns about the paper – While this paper addresses an important topic in a fairly interdisciplinary way (which is still missing in a lot of adaptation research), this paper is not well organized and the way the results are reported are frustrating and difficult to process.

This occurs for two reasons: 1) there are two very different sets of methods being addressed, and 2) the authors did a poor job of synthesizing the results from those methods.

Methods

There are also some issues when explain the methods, especially the social science aspects. First, the methods associated with the social science aspect of the work or not well described or well-organized. Problems associated with those methods are described below:

Method 1 - describe the social science methods, including interviews with government officials and surveys distributed to the general population.”

Answer to Comment:

We agree with the Reviewer that the social science part needed further clarifications and deepness on the methodological approaches. We extended and restructured it accordingly. The revised manuscript adds not just information on the social science methods, but goes further into the details on how the methods - the literature review, survey and interviews - have been shaped and implemented in this study. For all conducted methods, we also added information on the analysis process. Regarding the literature review, we added a table enumerating the considered grey literature, dealing with reports on climate change adaptation on the island and the seaport on Fuvahmulah in particular. In regard to the semi-structured interviews, we inserted information on the interview

partners, for example the number of participants in the different sectors taking part in the interviews and the duration of the interviews. We further considered adding another table, stating which government and non-government organizations on national and local level were consulted in the interviews. However, as we granted anonymity to all our sources of information, which was particularly important due to the political situation in the country, we refrain from giving further information on the organization's names, because the pool of actors in climate adaptation in the Maldives is small and the actors could have been identified easily. But even without these details, we substantially extended the social science method section.

Actions taken for this comment:

- Subsections “Literature review”, “Survey” and “Interviews” to better organize the social science method section have been added.
- Deeper description of natural & social science methods according to the following comments by Reviewer #2 have been included.

Comment 2.3:

“Critical information concerning the three main social science methods missing from the paper (which affects the ability for this work to be reproduced) is listed below:

1. Lit review

How was this done? There is literally one sentence describing the methods for this aspect of the paper and the supplemental document does not go into much more detail. How was this done? What sources were used to identify laws and regulations? There are also results from an Environmental Impact Assessment (EIA). Why are these here. Are these results the authors collected for the lit review?”

Answer to Comment:

We added information on how the literature review was conducted. This information includes a table with the analyzed grey literature. Information on relevant laws, decrees and regulations were provided to set a larger picture to the policies and governance in coastal protection and infrastructure.

In the meantime a valuable article was published by Gussmann & Hinkel (2021), summarizing the existing policy framework on coastal adaptation

on the Maldives. Essential information from this paper is discussed in our manuscript and reference is given to the work of Gussmann & Hinkel (2021).

In accordance with Reviewer #4, we give deeper insights and a more thorough discussion on the seaport's history in terms of its planning and implementation phases. This decision reveals the original motivation of coastal development that led to arising erosion problem on the island. Moreover, we shed light on the de facto response strategy on national level with international donor agencies and consultants to cope with the coastal erosion problem by implementing a massive structural revetment on the north of the island to alleviate the current situation. Here, it was necessary to review a set of Environmental Impact Assessments (EIAs) and Environmental and Social Impact Assessments (ESIAs). The insights of these reports are most important, as they strengthen our discussion by referring to the Reviewer's statement that *"This study displays the exploitation of the climate change narrative to cover for maldevelopment on a local level"* which the Reviewer finds *"incredibly impactful"* in comment 2.8. It epitomizes the findings discussed in Zubair et al. (2011), but in terms of this very specific and drastic example of Fuvahmulah. Since these documents are especially important from our viewpoint, we added a table to the methods, mentioning the documents dealing with the seaport on the island and mentioning explicitly Fuvahmulah in regard to coastal adaptation.

The article now contains an entire section in both results and methods on the literature review.

Actions taken for this comment:

- Subsections *"Coastal adaptation in practice"* in the results and *"Literature review"* in the methods have been added.
- Table 2 added (page 26).
- Further reference has been made to Gussmann & Hinkel (2021) and Zubair et al. (2011).

Comment 2.4:

"2. General population surveys

What were the questions on the survey?; How were the questions analyzed or coded?; Were any of these participants also interviewed as people who work on local coastal management? This is not explicitly stated in the methods or the supplement. For example, the results section provides a quote from an

interviewee that discusses the protective functions of the reef immediately after it reports that “islanders feel safe because of the coastal ridges around the island.” Does this apply to interviewees or survey responses?; The paper gives no context as to what people were specifically asked that can be used to directly demonstrate how that information was used in comparison to the physical modeling and DEMs when making arguments that local knowledge better understood oceanic-climatic pressures than high level officials.”

Answer to Comment:

The Reviewer raises important points which have been indeed not fully covered nor thoroughly discussed. To this end, we appreciate the Reviewer’s input and did accordingly improve our manuscript touching the following points

- The relevant questions of in the survey are now also in the supplementary materials, together with the associated codes and coding rules.
- We now also mention explicitly that non survey participants have been associated or affiliated with local work on local coastal management.
- Citations in the results now name survey number and participant number, indicating the results originate from the survey.
- Addressing the example of safety perception due to coastal ridges - we indicate now, that these results come from the interviews.

The question in the supplementary materials indicate that our questionnaire did not contain a specific query on the local’s knowledge or perception of coastal erosion in the island, but the open questions allowed them to express their knowledge freely (which they did), for example, coral reef environment as well as drivers and causes for the initiation and progression of coastal erosion. Comparing the local’s answers with the reports on the seaport and revetment shows locals were able to find the root causes to erosion in full contradiction to high level officials and their contracted consultants in their reports. Whether officials and consultants do not know, deliberately overlook or even deny these root causes for coastal erosion is subject to speculation that goes beyond the scope of the manuscript. We therefore concentrate on the information available to us, but refer to this specific aspect on “imported knowledge” in the results and discussions paragraphs.

Actions taken for this comment:

- Added questions from the questionnaire to the supplementary materials.
- Further information on how the surveys were conducted and how the data was analyzed has been added to the method section.
- The methods now contain the following (line 685-687): *The sample size of each survey is equivalent to about every eighth household on the island, while household selection was done by previous randomized sampling, yielding an unbiased representation of local households. Interviewees were not working in local coastal management.*
- Updated citation identifier after each survey citation.
- The results now contain the following (line 278-281): *More than half of the local respondents feel safe on their island with regard to sea level rise (52% of the respondents). The interviews have shown this sense of safety roots from the awareness that the lower-lying center of Fuvahmulah is protected from seaborne extreme events by the island's fringing coastal ridges.*
- Results, synthesis and results now better explain our approach and combination of the interdisciplinary approach.

Comment 2.5:

“3. Interviews with government officials

Methods do not provide enough detail concerning the interviews and how they were conducted. What were the interview questions? How long did they last?

Did the managers indicate on maps where there were issues associated with specific oceanic-climatic pressures or were there key words or place names that were given? How were these chosen? Why did the national level include government and non-government officials? What does that add to the study? How were the interview results analyzed? If qualitative, how were key words coded or highlighted?”

Answer to Comment:

We fully agree with the Reviewer that any in-depth information on the concept and contents of the interview methods with actors of the coastal authorities was lacking depth. We have therefore added further information in the methods section and have for example argued why it

was important to also include non-governmental actors to better understand how coastal management issues are typically dealt with in the country. We furthermore reformulated and gave information on the duration as well as the structure and concept behind the interviews (see methods and supplementary materials). The analysis process as well as the development of the code system has also been added to the methods section.

Regarding the question on how the exposed areas on Fuvahmulah affected or deteriorated by ocean-climatic pressures were identified in the individual interviews, especially on the local level, place names were used to describe the affected areas, for example the names of the affected wards on the east coast (e.g. *Maalegan, Hoadhadu*), or other key words, such as a playground on the east coast. At all times, there has never been a need for further specification - all statements and places referred to were unambiguously understandable. Therefore no maps were used.

Actions taken for this comment:

- Added information on how the interviews were conducted and analyzed in the method section (incl. duration, development of the code system).
- Added information why it was important to conduct interviews with non-governmental workers.

Comment 2.6:

“Method 2 describes the physical modeling of oceanic-climatic pressures of interest, including modeling of erosion adjacent to the new harbor and sediment dynamics on the reef.

Models describe:

- *Sediment Transport and relocation*
- *Identifying erosion using satellite imagery*
- *Near-shore wave models*
- *DEMs*

These sections were much clearer, but the authors disproportionately focused on the modeling results and provided much more detail than the social science activities.”

Answer to Comment:

We enhanced the social sciences result and methods section according to the Reviewer's previous comments. However, despite being an interdisciplinary approach, we are not striving to achieve a 50:50 distribution between natural and social sciences in the manuscript, but to address the research interest raised by the project and manuscript.

Action taken for this comment:

- Improved social science methods and result paragraphs.

Comment 2.7:

"Results:

There are several issues associated with how the paper is organized and how the results are presented. The same ambiguity of the social science methods also occurs in the results.

It is never made explicitly clear when specific results for the social science methods are being presented. For example, when discussing the literature review regarding relevant laws and regulations associated with coastal management in the results, the authors do clearly explain that these are the results from the lit review. Furthermore, the discussion of the law and regulations is weak and interspersed with results from the surveys.

Answer to Comment:

Following earlier responses to the Reviewer, we disintegrated the intermixed results from social science and the natural/engineering findings. The manuscript is now re-organized with clear separation of the two main disciplinary approaches, synthesized in the section following the results. The entire discussion section has been modified to "*Synthesis and Discussion*", as well as re-structured to better address the individual findings and consequences. Therefore, we are confident to have significantly improved the paragraph on discussion on the relevant policies and regulations as well as the overall structure (see also comment 2.8 below) of the manuscript.

Comment 2.8:

The results are also organized in a way that does not clearly present the social science results, the physical modeling results, and then the synthesis of both. For example, some of the survey and interview results are reported on the physical modeling sections. While this is meant to demonstrate that local knowledge matches physical modeling and serves as an attempt of synthesizing the results, these sections are disjointed, as the synthesis information is typically provided at the end of the long modeling results sections, and are not tied back together.

A way to potentially improve the structure of the paper would be to 1) present the physical modeling results that demonstrates where there are certain areas that have higher sediment transport or erosion potential that are based on maladaptation, 2) present the results about the laws and regulations associated with those particular pressures from the social science perspective, and 3) include a synthesis section of the results that demonstrates how the physical models match those same results were information and local knowledge that is elucidated from the social science methods.”

Answer to Comment:

We appreciate this comment and strictly followed the Reviewer’s very meaningful recommendation. We restructured the entire results paragraph starting with the physical modeling results. We removed all social science results from other sections and combined them in a separate section, following the physical modeling. According to Reviewer #4, we also provide further insights into current coastal adaptation practices on the island in this section as part of the literature review.

After writing a synthesis section, we found this section to contain discussion elements already. Yet, keeping a clear distinction between a synthesis section and the following discussion paragraph was worth trying, but we managed to merge this section with the former discussion section, now reading “*Synthesis and Discussion*”. The discussion on suitability and limitation of the methods is now in the associated method parts. To this end, we follow the overall recommendation of the Reviewer and are thankful for pointing this out.

Action taken for this comment:

- Restructured entire results and discussion section according to the Reviewer’s suggestions.

Comment 2.9:

"Discussion

Once connections between physical and social science results have been made clear, they should be further discussed in the discussion section. Additionally, a key discussion point that is incredibly impactful is "This study displays the exploitation of the climate change narrative to cover for maldevelopment on a local level." However, while this claim has potentially substantial implications, it is not well introduced as one of the main drives of the research, nor it is clearly explained how that conclusion derived from the project results. This aspect of the study does not appear anywhere else in the document -if it is, it needs to be more clearly stated, because this statement does address a less well-researched topic. While several studies demonstrate the necessity to incorporate local knowledge into adaptation planning to avoid maldevelopment (a few are listed below the summary), few that I know of purposely examine whether maldevelopment is being explained and justified using the exploitation of climate change narratives by national level officials. IF this was the major claim for this paper, it is not well framed throughout the entire paper and should be refocused on this issue."

Answer to Comment:

The current, revised discussion section dismisses any discussion of methodological approaches. We added contents and migrated this paragraph to the respective method parts which provides a better organization and readability and a clear focus on the actual results.

The revised manuscript additionally refers to Alexandre Magnan's recent publications, which we included in our discussion added to the references list (Petzold and Magnan, 2019; Magnan and Duvat, 2020), as he and his co-authors discuss recent (mal)adaptive developments in the Maldives and compile insights in dealing with climate change on SIDS in general. We include a short literature review and use a number of reports and data sources, which - in our opinion - justifies a larger number of references.

We are confident that by means of re-organizing and adding in-depth contents in our revised manuscript the key statement that *"This study displays the exploitation of the climate change narrative to cover for maldevelopment on a local level"* can be sustained and has been fully exploited by the meaningful comments and recommendations provided by Reviewer #2 and #4.

Petzold, J., Magnan, A.K. Climate change: thinking small islands beyond Small Island Developing States (SIDS). *Climatic Change* 152, 145–165 (2019). <https://doi.org/10.1007/s10584-018-2363-3>

Magnan, A.K., Duvat, V.K.E. Towards adaptation pathways for atoll islands. Insights from the Maldives. *Reg Environ Change* 20, 119 (2020). <https://doi.org/10.1007/s10113-020-01691-w>

Action taken for this comment:

- According to the Reviewer's suggestions, we dismissed methodological discussions and focussed on discussing the studies results. Suitability and limits of our methods were mentioned in the respective method sections.

Comment 2.10:

"Summary: This research attempts to implement a truly interdisciplinary research plan, with the goal of synthesizing physical and social science methods to substantiate and justify the need for incorporating local knowledge into adaptation planning as a way to avoid implementing maladaptation that is potentially destructive in the future. However, due to major organizational issue with the paper and a lack of detail, especially concerning the social science aspects and too much focus on the physical modeling, the paper would benefit from major changes.

Karen Elizabeth McNamara & Ross Westoby (2011) Local knowledge and climate change adaptation on Erub Island, Torres Strait, Local Environment, 16:9, 887-901, DOI: 10.1080/13549839.2011.615304

Reineman, Dan R., Leif N. Thomas, and Margaret R. Caldwell. "Using local knowledge to project sea level rise impacts on wave resources in California." Ocean & Coastal Management 138 (2017): 181-191.

Magnan, A.K., Schipper, E., Burkett, M., Bharwani, S., Burton, I., Eriksen, S., Gemenne, F., Schaar, J. and Ziervogel, G. (2016), Addressing the risk of maladaptation to climate change. WIREs Clim Change, 7: 646-665. doi:10.1002/wcc.409"

Answer to Comment:

We outlined our actions to the Reviewer's previous comments, followed by major reorganizations of the results, discussion and methods section. This has especially benefited the social science section, however it was not our intent to achieve a 50:50 distribution between both disciplines - but rather to fully address and manage a complex research topic reflected in this manuscript and use the social science perspective to fill knowledge gaps from the physical modelling perspective.

But addressing the Reviewer's constructive comments has greatly benefited the manuscript and we would like to appreciate the Reviewer's efforts by responding duly and diligently. Thank you for taking your time!

Reviewer #3 (Remarks to the Author):

Comment 3.1:

“Overall comments:

Title: Climate change induced effects or maldevelopment: small islands and conflicting attribution of root causes

Very good topic. However, major revisions are needed to improve the quality of the paper

Several acronyms are listed in the manuscript. Authors need to give the full form of abbreviations in a tabulated form after the Abstract.”

Answer to Comment:

We appreciate this recommendation, although we have so far not encountered a list of acronyms and abbreviations in a Nature Communications publication. Therefore, we have not attached a list to our submitted manuscript yet. However, we are happy to support the review process by providing a list of acronyms and abbreviations in the revised manuscript.

Action taken for this comment:

- Acronyms and abbreviations list added following the manuscript (last page).

Comment 3.2:

“Abstract

Well written Abstract. However, revisions are needed.

Authors need to give information on the coastal length of the Fuvahmulah area in the abstract.

Also, what interdisciplinary perspective has been used for this study (Survey and DEM models).

They should also need to give this information in the abstract.

How are the results useful to the policy and decision makers? Should need to include this information as well.”

Answer to Comment:

We generally agree to this point highlighted by the Reviewer - the abstract should contain a better overview of the methods used. We also acknowledge that information on coastline length is missing in our article, despite being an important piece of information in small islands. We now mention the coastline length in the study site section, which is also helping to explain the spatial impact of the massive revetment in the results (where we mention the information again). However, in our opinion, giving the length of the coastline already in the abstract is too detailed information. We are also convinced that the last sentence of the abstract *"The results of the interdisciplinary approach demonstrate how small-islands' adaptive capacities are typically impaired in dealing with climate-related changes, they reveal the structural challenges of top-down processes, and disclose the potential of local knowledge to overcome maldevelopment"* already provides a sufficient level of detail on the benefits of our study to researchers, practitioners and decision makers.

Actions taken for this comment:

- Explained methods in the abstract.
- Added coastline length in the main text (line 101).

Comment 3.3:

"Introduction

Lines from 25th to 30: References are needed for those statements.

Authors heavily depended on grey literature rather than synthesizing the global literature regarding small islands -coastal erosion or vulnerability. One paragraph is needed regarding this."

Answer to Comment:

Thanks for outlining this. We used a set of literature to backup the statements in these parts of the introduction.

Action taken for this comment:

- Added references.

Comment 3.4:

"Study site

What is the length of the Fuvahmulah coastal area?

How much of population of this area?

How much coastal area was already eroded in the last two decades?

What is the erosion rate in this area?"

Answer to Comment:

We agree that census data and additional site information are missing, concerning coastline length (11km) and population of Fuvahmulah (8510 according to the latest census), which we added to the study site section in the revised manuscript.

However, we are inclined to exclude magnitudes of historic coastal erosion phenomena on the island. There have been government reports on erosion rates, based on transactional measurements and aerial images. But it is merely impossible to match and correlate these previously acquired transect data of the reef and beach width with our detailed geodata from the field surveys.

Traditionally, data from the Maldives is scarce. This is challenging for current research, but we are confident to give a good and cautious estimation, based on our data, without simplifying or extrapolating processes. We are also confident, our methods are robust in order to outline the potential and address the inherent limitations of our approach. However, we agree with the Reviewer that such data would be helpful and back up the rather punctual but incoherent former measurements.

Actions taken for this comment:

- Added coastline length to study site and result section (line 101).
- Added population according to the 2014 census to study site (line 113).

Comment 3.5:

“(Study site continued)

What are the socio-economic and political characteristics of this area?

Information is needed regarding above mentioned queries for this particular section.”

Answer to Comment:

We added the population number from the most recent census (2014) to the study site section and further information on the socio-economic

characteristics of the island in the beginning of the section “Governmental and societal framing of coastal development” .

We have also defined the political framing of the Maldives and Fuvahmulah to the reworked introduction and results. A further description of the political characteristics in the study site section would be repetitious in our opinion.

Actions taken for this comment:

- Added population number according to the latest census (line 113).
- Gave further information on the socio-economic characteristics in the results chapter.
- Better described the political characteristics in the introduction (see also comment 4.6).

Comment 3.6:

“Results

Figure 2 is a nice representation of coastal erosion.

What is the average rate of coastal erosion per year in this site?

How many km of coastline already crumbled into the sea?

Information/explanation is needed?”

Answer to Comment:

Thanks for pointing out these aspects since we agree that Fig. 2 unveils the ongoing coastal erosion processes quite well.

We added the timeframe (“[measured] *between 2017 and 2019*”) to each acquired dataset resembling the coastal decline, allowing the reader to estimate erosion rates based on data gained from our field surveys. We also added the distance between the port entrance and transects of Fig. 2c and 2d in the text, giving the reader information on the spatial extent of the erosion. Beyond transect 2c, erosion was measured occasionally but not constantly. Together with the rather abrupt coastal decline, we refrain from giving a general measure for the entire east coast or greater parts of it.

Actions taken for this comment:

- In the text, added time frame between measurements (line 130).
- Added distances between transects of Figure 2c and 2d to the seaport’s entrance (line 139).

Comment 3.7:

*"Impacts on the natural physical environment
Wave climate of the southern Maldives
Well written part"*

Answer to Comment:

We do highly appreciate this comment and are satisfied that the Reviewer acknowledges our work. Together with the constructive issues raised throughout the review process, this motivated us to respond to the Reviewer to the best of our capabilities.

Action taken for this comment:

- None.

Comment 3.8:

*"Discussion
Good. But can be expandable with a comparison of some recent global examples."*

Answer to Comment:

In accordance with other reviewer's comments, we have restructured and enhanced the discussions.

Action taken for this comment:

- See other reviewer's comments for more detailed information.

Comment 3.9:

*"Conclusion
Good, but too lengthy. Revisions are needed."*

Answer to Comment:

Thanks for generally appreciating the conclusions. We tried hard to revise the conclusions and made some revision to shorten this section.

Action taken for this comment:

- Minor revisions undertaken.

Comment 3.10:

"What are the study limitations? Information is needed."

Answer to Comment:

We shifted the appropriateness and limitations of each method to the methodology part.

Action taken for this comment:

- Restructured manuscript according to this and other Reviewer's comments.

Comment 3.11:

"Methods:

- 1. Why has the survey taken from the age group of 14 ?
Is there any particular reason?*
- 2. Where is the ethical approval number?*
- 3. How may interviews have been taken?*

Information is needed for above mentioned queries."

Answer to Comment:

Responses to point 1 and 3 are below in the "actions taken" section.

Regarding point 2:

An ethical approval number is not intended in our research system, but our research has to follow a dedicated code of conduct, including ethic guidelines. Ethics assessments are already conducted in the project submission (link, however in German, please find a translated version in the Annex of this document) and any violation leads to a rejection of the research proposal. In turn, a granted proposal has successfully passed the ethics assessment and is also approved from an ethical standpoint. We conduct our research in a research project, settled within a priority program and have passed all necessary evaluation criteria - including the ethical vote. We state our research funding in the acknowledgements and adhere to any regulations, guidelines as well as the code of conduct.

Furthermore, we have a local partner (Dr. Zahid, Maldives Meteorological Service) who assisted us in the Maldives and - amongst other things - to follow all legal regulations applicable in the Maldives. Our research was

further acknowledged and supported by the local government of Fuvahmulah.

As suggested by the editor, we also added Dr. Zahid as co-author, as he gave valuable insights from his local perspective when addressing the Reviewer's comments.

Actions taken for this comment:

1. Added *"The lower threshold of 14 years results from experience gathered on the island and in consultancy with the local research assistants – persons over 14 ys. of age start joining environmental activities without their parents guidance and thus are considered aware of Fuvahmulah's environmental situation."* (line 689-691).
1. Ethics approval according to the "editorial policy checklist" file, submitted to the journal and the editors.
3. Information on interview number and interviewees affiliations are now in the document.

Comment 3.12:

*"DEM and Numerical wave models
Very well explained and well written parts."*

Answer to Comment:

Thank you for acknowledging this section and also thank you for taking your time and efforts, reviewing our manuscript.

Action taken for this comment:

- None.

Reviewer #4 (Remarks to the Author):

Comment 4.1:

“This paper applies an interdisciplinary approach which combines scientific and sociopolitical research to understand if erosion problems in the Maldivian reef island Fuvahmulah are due to climate change or human interventions (maldevelopment).

The problem of erosion in this island is analyzed from different perspectives (political decisions, local population perception, process-based methods), and it has discovered that island community and national government have a diametrically opposed explanation about the erosion problem. Measured coastline erosion and simulations of the sediment patterns corroborate the perception of the local actors.

As very positive aspect of your work is the integration of different data sources:

- The analysis of the environmental governance in the Maldives that are financially supported by different international organizations*
- Semi-structured interviews with actors relevant for coastal protection management*
- Surveys amount the local inhabitants that allow to find that a low percentage of population is involved in the development projects in the island. Very interesting to highlight the knowledge of the local people about the protective function of the heylhi.*
- Measured erosion rates from DEMs produced with the SfM approach*
- Characterization of wave climate in the Southern Maldives differentiating between the dry and wet season for the present climate and the projected future changes.*
- Application of two process-based models to analyze the sediment movement for two main characteristic storms and the wave-induced currents with and without the seaport.”*

Answer to Comment:

Thank you for outlining 6 very positive aspects of our work that actually reflect the uniqueness of our study in direct reference to the recent Editorial “Sea change in coastal sciences”, *Nat. Com.* 11, 4601 (2020), DOI:10.1038/s41467-020-18333-8. However, with the further comments,

we fear the Reviewer has been challenged to grasp the benefits of this unique methodological approach - leading to a very critical assessment of our study so that the Reviewer finally rejects our submitted manuscript. With the overall positive feedback and helpful suggestions provided by Reviewers #1 to #3, together with the introductory evaluation of Reviewer #4, we feel encouraged to give our best to reshape und restructure the manuscript accordingly and improve readability as well as key methodologies. We are sure, this allows Reviewer #4 and readers of the article to gain better access to our research.

Despite some discrepancies, we highly appreciate Reviewer #4's very critical comments, giving us further motivation to improve the text and even enhancing the literature review to current challenges on Fuvahmulah. In the meantime, a set of studies were published, giving more insights into the Maldivian socio-political setting (for example Gussmann & Hinkel, 2021) and the natural dynamics on Fuvahmulah (David and Schlurmann, 2020). The combination of the reviews, the new literature cited and the enhancement of the analysis of our results, gives us confidence - at least motivation - to provide valuable and tangible insights into maldevelopment on small islands, readily understandable to the broad audience of Nature Communications.

References

Gussmann, G. & Hinkel, J. A framework for assessing the potential effectiveness of adaptation policies: Coastal risks and sea-level rise in the Maldives. *Environ. Sci. & Policy* 115, 35 – 42, DOI: <https://doi.org/10.1016/j.envsci.2020.09.028> (2021).

David, C.G. and Schlurmann T. (2020): Hydrodynamic Drivers and Morphological Responses on Small Coral Islands — The Thoondu Spit on Fuvahmulah, the Maldives. *Front. Mar. Sci.* 7:538675. doi: 10.3389/fmars.2020.538675

Comment 4.2:

"I think the main conclusion about the study is the people's participation in infrastructure projects would have avoided or reduced the erosion problem generated by the construction of the port because their knowledge about the

local environment would have contributed to a better design. It is worthy to emphasize the importance of local knowledge when interventions are projected.

On the hand, it seems that the purpose of your results is to highlight that climate change is not always the cause of erosion in Maldives as the national government claims and they use this reason to obtain international funding. Although this might happen, the case you have studied represents the local impact of an infrastructure that was built more than a decade ago and cannot be use an example of false adaptation measure or as an adaptative capacity which deteriorates the adaptation to climate change. I am not sure if you also try to demonstrate that erosion due to climate change (mainly due to sea level rise) will not be a problem is this island because people have the perception of safety against climate change impacts. The climate change effects may not be visible nowadays being one reason why people did not mention it in the survey, and also, because the survey was very focused on the erosion problem related to harbor. It seems that you try to extrapolate what happen with the construction of the port with potential erosion problems along the coast of the island (lines 269-270)."

Answer to Comment:

As we understand it, the Reviewer discusses 3 aspects with the comment:

1. After a short summary, the Reviewer sees the study as an impact assessment of the local port construction which cannot be used as an example of neither maladaptation nor maldevelopment, because it was built over a decade ago.
2. The Reviewer also thinks, our statement and the opinion of the local people is that Fuvahmulah will not suffer from climate change, as people have the perception of safety against climate change impacts - just because they do not see any climate change effects as of now.
3. Furthermore the Reviewer understands that we extrapolate the impact of the port construction to other issues along the coast.

This summary reflects our understanding of the raised issues by the Reviewer. We hope to have understood the comment correctly and are going to elaborate on them in the following:

1. We tend to generally oppose the Reviewer's statement, as the seaport is maladaptive per definition. The definition of maladaptation is an *"action or inaction that may lead to increased risk of adverse climate-related outcomes, increased vulnerability to climate change, or diminished welfare, now or in the future"* as pointed out in lines 44-46 in the manuscript. Of course, the seaport is of commercial and logistical benefit for the island and increases the economic development and human well-being of the local inhabitants. Also, the seaport was highly demanded before its construction and following the comments of Reviewer #4, we added the sentence *"The seaport construction was arguably a trade-off for economic development at the expense of natural conservation"* to the discussion. But from interviews with local decision makers, we got information that the then commissioned international engineering consultants have suggested investigating other sites and prioritized another location for the seaport, but governmental authorities were in favor to invest into an industrial and logistics area in the south of the island - meaning the adverse impacts were already suggested by the international engineering consultants and could have been mitigated. Again, it is not our intend to criticize this decision, but responding to the Reviewer's comments, we now further outline in the discussion *"However, instead of governing this conflict (Bisaro & Hinkel, 2016), future plans are going to respond to erosion associated with the seaport's construction with further generic main-streamed adaptation measures (Hoogduin, 2016; Saleem, 2016) rather than acknowledging and dealing with the root causes of erosion"*. So in context of maladaptation's definition, the seaport construction was an action leading to increased vulnerability to climate change and coastal flooding, as it has a negative influence on the coastline by degrading the natural protection barrier, being the vegetated ridge. This natural protection barrier significantly strengthens the island's resilience to withstand ocean-borne pressures. Impairing the resilience generally (also in this case) leads to increased vulnerability. To this end, the port was never intended as an adaptation effort (neither the local authorities nor us did ever claim it was), but the infrastructure project meant to progress the local economy and sustainable development!
The Reviewer also mentions, *"the local impact of an infrastructure that was built more than a decade ago [...] cannot be use an example of false adaptation measure"*. We also contradict this - just because adverse impacts were not visible from day one, earlier government reports already document erosion a few years after the port construction.

However, erosion is an ongoing process, intensifying over the years. As of today, erosion is distinctly visible and measurable and our results clearly identify the port causes this - and thus being maladaptive.

2. Our study does not aim to project future states of erosion or sediment transport under different climate change pathways - it solely provides information on the projected changes of drivers and pressures in the face of climate change that will surely aggravate the situation in the area - with or without the seaport. Recently, rudimentary assessments on the morphodynamic behavior of reef islands on island and atoll scale have been made (Shope and Storlazzi, 2019) and first findings on cross-sectional changes under rising seas have been published (Masselink et al., 2020). But from our viewpoint, there is no suitable and reliable tool to provide deterministic insights on coupled hydro-morphodynamic processes on island scale over the upcoming decades, because process-based (numerical) tools predicting both long- and cross-shore sediment transports in the special hydrodynamic regime are still under development. Therefore, we do not make any suggestion on how sediment transport processes are likely to alter around Fuvahmulah under rising sea levels or changing climates; although we do fully agree with the Reviewer that these stressors will - in all probability - worsen the coastal erosion problems and will sooner or later lead to an uninhabitable state of the island within decades - as Storlazzi et al. (2015) point out. Yet, this is not an unrealistic scenario, but not the core of our study

But what we can do, is to assess the current challenges the island faces that complement any upcoming future challenges. One of these current challenges is coastal erosion (or shoreline changes in general, because in itself [*escalating rates of island physical change [...] will stress populations*]; see discussion section of Masselink et al., 2020). We also explicitly outlined this in our manuscript, and put emphasis again in this statement that *"Together with today's impact of the seaport, future modifications to the natural reef system will most likely have even more significant adverse effects for greater parts of the coast."* and *"Risks evolving from maldevelopment add onto the current and future risks of the other significant natural driver, being climate change and the associated impacts."*

Concerning the people of Fuvahmulah, they feel in general safe on their island as they are well aware of the unique function and level

of protection based on the (still healthy) reef, but they also understand and anticipate the risk of ongoing but future coastal erosion and are worried about it. At no point, we argue that the people of Fuvahmulah have a false sense of security and are thus unaware of climate change related impacts. But they see both: a.) they have an ongoing coastal erosion problem in the south east of the island and b.) they think the port construction has led to this problem (we enhanced the results section to make it clearer that locals see the protection features, see subsection “Local society’s experience with coastal processes”; we also added more information on our methods in the supplementary material, i.e. important questions of the survey sheet and according procentual responses to the questions or selected statements from participants).

From a literature review on this topic, we know that small islands are projected to be at high risk to suffer from sea level rise and associated impacts, but that shoreline responses do indeed differ among reef islands (McLean and Kench, 2015; Albert et al., 2016; Kench et al., 2018). The local population however attributed the current scales of coastal erosion to the more recently built seaport. This reflects their own observations - nothing more nothing less; and they are right with their perception of causes and effects as our modelling attempts confirm. After our field campaign, we started to assess the reefs sediment transport system (David and Schlurmann, 2020) and from there were able to systematically analyze the role and impacts of the port. Based on this analysis, we were already quite confident that the major forcings leading to coastal erosion on the south-east coast was actually caused by the construction of the seaport - standing against the national authorities declaration, which they released with the support of external donor agencies and consultants. But to back this statement up, we also checked the extent of shoreline recession on the entire east coast, leading to point 3:

3. We do NOT extrapolate the erosion on the port to “*potential erosion problems along the coast*”, but show with Figure 2, that the erosion problem is limited to the south-eastern coast of the island. In fact, there have been reports on erosion due to an earlier measure - a seawall - which we now also discuss in our manuscript. But we

neither extrapolate our measurements in accordance to these historic data nor the experiences of the local people.

Even though we have a partly opposing stand on the reviewer's comment, it allowed us to improve our manuscript, by clarifying the issues raised by the Reviewer according to the above mentioned changes, associated with each of the Reviewer's raised issues.

New references

Storlazzi, C. D., Elias, E. P. & Berkowitz, P. (2015). Many atolls may be uninhabitable within decades due to climate change. *Sci. Reports* 5, DOI: 10.1038/srep14546

McLean, R. and Kench, P. (2015), Destruction or persistence of coral atoll islands in the face of 20th and 21st century sea-level rise?. *WIREs Clim Change*, 6: 445-463. <https://doi.org/10.1002/wcc.350>

Albert, S. et al (2016), Interactions between sea-level rise and wave exposure on reef island dynamics in the Solomon Islands, *Environ. Res. Lett.* 11 054011

Kench, P.S., Ford, M.R. & Owen, S.D. Patterns of island change and persistence offer alternate adaptation pathways for atoll nations. *Nat Commun* 9, 605 (2018). <https://doi.org/10.1038/s41467-018-02954-1>

Action taken for this comment:

- Restructured and overhauled the results and discussion section to make our results and the associated implications in face of climate change and maldevelopment more understandable and plausible. For more detailed info, see other review comments.

Comment 4.3:

"The construction of the port was demanded by people from Fuvahmulah for decades because previously to the construction, transporting people, goods and cargo were made by small fishing boats (dhonis) from the island to large ships offshore with many boats capsized and people died. I think it is not

complete accurate to give maldevelopment the same connotation as maladaptation because both increase vulnerability or diminish welfare. Seaport has been a very important coastal infrastructure for the economic and social development of the island. Besides, although the harbor acts as a barrier to sediment transport, it does not mean that climate change will exacerbate the impact, a local impact assessment should be performed to verify this supposition. Climate change could affect not only the wave height but also the wave period and direction which is a variable more relevant in sediment transport."

Answer to Comment:

In this comment, the Reviewer refers to 2 aspects:

1. The distinction between maladaptation and maldevelopment is not clear enough.
2. Climate change does not necessarily increase the impact of the seaport.

Referring to 1:

Thank you for raising this important point. We acknowledge to give maldevelopment a more distinguishable definition compared to maladaptation. There is a very explicit distinction between maladaptation and maldevelopment which we now explain in more detail in the introduction according to the international literature which goes along with our own definitions (see lines 42-61). In the discussion, we now also better outline how our example epitomizes maldevelopment in the sense of our definition.

Furthermore, referring to 2:

Research dealing with projected sea states anticipate a global change in wave conditions (for example Hemer, et al., 2013 and Dodet et al., 2019) but due to limited numerical modelling capacities and since we cannot fully and confidently prove this, we never state (or wrote) that climate change will add onto the seaports impact - but vice versa, we argued that the maladaptive impact of the sea port will add further stress onto the island and its population beyond those associated with climate change, sea level rise and (increasing) ocean hazards. This is a meaningful difference!

New references:

Dodet, G., Melet, A., Ardhuin, F. et al. The Contribution of Wind-Generated Waves to Coastal Sea-Level Changes. *Surv Geophys* 40, 1563–1601 (2019). <https://doi.org/10.1007/s10712-019-09557-5>

Action taken for this comment:

- Improved the definition of maldevelopment (lines 42-61) in the intro and framed our study within this definition in the discussions.

Comment 4.4:

"I consider that the meaning of several concepts: coastal infrastructures (which can increase quality of life), adaptation measurements (being hard engineering solutions one option, luckily there is a change of mind to a more environmental interventions) and natural self-adaptation measures (being the reef and coastal vegetation natural coastal protections, not the sediment transport itself) are mixed. It is not comparable an infrastructure as a port with different options to respond to an increased coastal risk and sea level rise (as protection, accommodation, ecosystem-based adaptation, very different from engineering-type hard-coastal protection systems)."

Answer to Comment:

Yes, the Reviewer is right with this comment - the different elements vary in their type and effect. In our manuscript, we refer to

1. the seaport as coastal infrastructure for better supplies and logistical access in order to progress the general development of the island,
2. the revetment and other actions to mitigate adverse effects as adaptation and
3. the morphodynamic response on hydrodynamic forcing and the associated sediment transport as natural self-adaptation (see also David and Schlurmann, 2020).

Any of these exemplarily stated constructional elements exists and interacts with each with a certain impact that either sum-up or diminish each other: The (3) natural coastal dynamics are reacting to changing boundary conditions and ocean-borne pressures. In parallel, the ocean supports the livelihoods of island dwellers. However, the spatial circumstances (small aerial surface, isolation by ocean) require (1) coastal

infrastructure development to support living on the island. But research on other SIDS most often unveils the impact of human interferences on the coastline and its adverse effect on the islands' resilience. To mitigate these effects, (2) adaptation measures are implemented and safeguard human activity on the island.

As a result, all these elements and the complex interactions need to be considered in an integrated approach that requires planning efforts and decisions to be taken regarding the development of the island and, at the same time, coping with natural forcings in an attempt of coastal management and coastal protection. This is what we've done in our manuscript and are confident we use the terms and its connotations correctly.

However, the Reviewer's comment significantly helped us in the discussions, where we outline the causal links - that go beyond simple correlations - between nature, infrastructure and adaptation in the case of Fuvahmulah, giving a more comprehensive line of argumentation. Thank you!

Action taken for this comment:

- Restructured and rephrased discussions.

Comment 4.5:

"Although your work demonstrates that negative consequences from anthropogenic interventions could be similar to those to climate change, the study does not constitute a material to be published in Nature Communications journal. Although an interdisciplinary approach has been applied, most of the work is concentrated on demonstrate that the harbor infrastructure is the source of erosion on the coast adjacent to the north of the seaport."

Answer to Comment:

Similar to Reviewer #1, this Reviewer sees the technical aspect as the main contribution of our manuscript - and unlike Reviewer #1 does unfortunately not see our contribution fit for Nature Communications. Already the title of the manuscript tells the reader that the contents of the research goes beyond only technical aspects as it sets the trigger to differentiate between causal relations of coastal erosion affected by

conflicting attribution of origins in line with peoples' perceptions or experiences.

Here, the technical part serves as determination and problem assessment, which clarifies the geophysical background and source of the erosion problem - beyond the technical issues. But the natural climate change dimension is only one of (at least) two dimensions - especially considering the impact of humankind on nature (we are living in the Anthropocene!) and in the face of human-made climate change. Therefore, the other dimension is clearly the socio-political system, as the original source and recipient of the impacts of erosion.

The definition of maladaptation and the DPSIR concept already hint at the complex nature of climate change adaptation in practice and helps to outline the interlinkages within both dimensions. While there has been ample evidence within each community on drivers and responses of human interference within the natural system, there is very scarce literature on an interdisciplinary stand - and to the best of our knowledge no recent interdisciplinary evaluation on small islands in the face of climate change, despite being a strongly addressed topic within the IPCC.

We admit the first draft of the manuscript deserves major revisions as suggested by the valuable comments of both reviewers and the editor, but with these addressed, we think that there couldn't be a better journal to communicate our results in. We are confident, our study will have an impact on a broad audience from many disciplines.

We are puzzled, but think that there might be a kind of misperception by Reviewer #4 on the core idea of our manuscript on our main goals to criticize the harbor construction. We do acknowledge that our research has probably not been introduced correctly in order to outline our research aims and findings more clearly. We worked thoroughly on the respected paragraphs and are now convinced that ideas and methods are better, i.e. clearer addressed. In fact, we aim to reveal structural maldevelopment in coastal adaptation and epitomize this concept using the example of Fuvahmulah. In accordance with later comments of Reviewer #4, we enhanced and updated the literature review and thus the insights into the current developments. Using these results, we then improve our discussion and follow a clearer line of argumentation.

Actions taken for this comment:

- Reorganizing the manuscript.

- Better outline of the research aim in the introduction.
- improving the results section.
- Reformulating the discussion section to further clarify the implications of our findings and thus better highlighting the benefit of an interdisciplinary research approach.

Comment 4.6:

"You use the term "top-down implemented infrastructure projects" along the text and even deduce, from the analysis of the problem of erosion close to the harbor, structural challenges of top-down processes. What does this term mean for you? As far as I know, bottom-up or top-down approaches refer to the way the climate impacts are analyzed. Bottom-up approaches usually start with a system or impact (such as a disaster), and then identify all of the underlying variables, processes or phenomena that play a role in shaping the outcome, understanding the nature of the risks before identifying the relevant drivers and hazards. While in top-down or scenario-led approaches, scenarios are generated using climate models and then incorporated into an impact model introducing a lot of uncertainty because it is difficult to integrate multiple interacting drivers and/or hazards (Zscheischler et al., 2018). It should be convenient to describe this term related with infrastructure projects and its drawbacks. Why does this approach usually follow? You should describe in more detail the implications of this approach and how could be improved for future applications."

Answer to Comment:

This is a clear misunderstanding. Here, "top-down" refers to the way, infrastructure projects are planned and implemented - meaning that responsibilities and decisions are made at the national government level, with a minimum participation of regional stakeholders. In contrast, "bottom-up" approaches are initiated by locals or affected people, which are then supported and ratified by superordinate instances. We clearly missed to introduce this concept in a clear fashion in the introduction and caught up on this. This paragraph should be much clearer now in order not to misperceive this concept.

Action taken for this comment:

- First mention of top-down implementation is now enhanced with a definition, reading:

The Maldives government is forced to address these challenges and tends to top-down implemented infrastructure projects¹⁹, which often lead to anthropogenic disturbances in the natural system, thus undermining vital ecosystem services and leading to increased vulnerability^{7,9}. In a top-down approach to implement infrastructure projects, decisions on these projects are generally made on a national level and implementation of these projects is prescribed to the local level. (lines 38-42)

Comment 4.7:

"It is true that any coastal infrastructure will generate disturbances in the natural system, but many different configurations could be studied analyzing the effect in the close beaches or vital ecosystems in the area and the select the less damaging. I think the maldevelopment of coastal infrastructures could be avoid incorporating in the design a study of the changes in the sediment transportation pattern (e.g., in this study case), but this is an aspect that is not usually taken into account. It is also convenient to include climate change (in the drivers or hazards, exposure and vulnerability) in the design of coastal infrastructures to avoid worsening climate change impacts. There could be a better way to implement "top-down implemented infrastructure projects"."

Answer to Comment:

Yes, the Reviewer is absolutely right. We have confidential information on the basis of personal communication on the then conducted assessment of the seaport location and the design of the breakwater layout from conversations with the consultants at the time as well as with local decision makers on Fuvahmulah. But these technical reports and recommendations are confidential and we cannot cite this important source of information. But different port configurations have been studied and the consultants said, their own report even suggests investigating other sites. This was confirmed by our local sources, but it has been also mentioned that the port location was part of a distinct land-use planning with the (at the time) soon-to-be followed airport construction in mind.

This information was confidential. However, the current challenges of the coastal protection project are better documented and available to the public. We haven't directly discussed this earlier, but Reviewer #4's comment encouraged us to incorporate these hidden but not to be

confirmed insights into the study. We added another section “Coastal adaptation in practice” to the literature review and further address the current developments in the discussions.

This information massively benefitted the discussions and also the argumentation of the entire manuscript. The comment helped us greatly, and we thank the Reviewer for pointing this out!

Actions taken for this comment:

- Adding “Coastal adaptation in practice” subsection.
- Using the subsection’s information in the discussion.

Comment 4.8:

“The seaport was constructed in 2002, therefore, the erosion problem should have been an impact that evolves in time and becomes more evident as time passes, not just in the recent years. You calculated an erosion rate as the differences between DEMs made in 2017 and 2019. Only two measurements and without related with climate during this period could lead to false conclusions. Climate variability might explain the magnitude of the erosion rate in that period. Maybe some of the erosion problems close the seaport is due to port itself, which interrupts the sediment transport, but also could be attributed to sand mining (as the inhabitants of Fuvahmulah point to it as the most common reason of erosion in the island in the second survey attributed).”

Answer to Comment:

In the methods, we now give an overview on reports, EIAs and ESIA's dealing with climate change adaptation and more specifically the seaport on Fuvahmulah. Starting with the “Foahmulaku Beach Erosion Survey & Coastal Protection Report” in 2006, experts have already noticed erosion on the east side of the island. We discuss the quality of the governmental agencies’ assessments in comment 3.4., but there is consistent agreement of the port’s influence on erosion. At first, the erosion was not greatly observed - because it is an ongoing, slowly progressing process. However, erosion became noticeable enough that the national government signed a grant arrangement in 2014 - about a decade after the port construction.

We kindly ask the Reviewer to pay further attention to comment 3.4, where we are hesitant to give a definite erosion rate for the east coast or larger parts of it, as we essentially agree with the Reviewer - longer

monitoring of the situation would benefit the evidence and thus the confidence of drivers behind erosion. But with the presented results and those presented in David and Schlurmann (2020), we are confident in our analysis.

Addressing the Reviewer's suggestions, we further discuss the aspects sand-mining and climate variability in the discussions and outline that these aspects could have a minor impact on the island's erosion problem. But we also argue, based on Figure 2 and 3 in the manuscript, that severe erosion is a very dominant local event, while sand-mining was witnessed around the entire island - despite being illegal. And similarly with climate variability - from our experiences on-site and judging from our model results, changing wave directions would have a broader impact on the entire coastline (for a rough estimate example, see figure below with $\theta_p=40^\circ$). But it is important to address these issues Thank you!

References

David, C.G. and Schlurmann T. (2020): Hydrodynamic Drivers and Morphological Responses on Small Coral Islands—The Thoondu Spit on Fuvahmulah, the Maldives. *Front. Mar. Sci.* 7:538675. <https://doi.org/10.3389/fmars.2020.538675>

Action taken for this comment:

- Addressed the comment's issue in the discussion.

Comment 4.9:

"It is not clear for me how you apply a DPSIR framework (drivers, pressures, state, impact and response) in this work? Has this framework got specific steps within a methodology?"

Answer to Comment:

The DPSIR framework is a purely conceptual framework setting causal relations in between interdependent factors or constituents for describing the interactions between society and the environment. It refers to the way data and information on the different interdisciplinary elements are collected, how possible connections are made and conclusions are drawn along a given - the DPSIR - chain of constituents. In simple words, it helps tracking drivers behind the current, measurable situation (impacts) or responses along a causal chain with distinct categories (*drivers* → *pressures* → *state* → *impact* → *response*). It is a widely recognized and applied concept, for example in assessments by the European Union. In this way, the argumentation remains tangible and comprehensive along the logic chain and allows to transfer the lessons learned to other, similar cases.

Action taken for this comment:

- Reformulated DPSIR specific parts in the introduction and indicated where results or discussion use or address DPSIR causal framework within the text.

Comment 4.10:

"More specific comments about some sections of your study:

Regarding the wave climate characterization, it is not clear for me how you have calculated the changes in Hs (as the differences between the time series of the 3 RCPs at the end of the century (2090-2099) and the present time period (2010-2019)). Because the percentage of increase seems to be very high (see Hemer et al., 2013). Are you considering the different GCMs available at CAWCR repository?

Answer to Comment:

We have chosen the 9 year period according to the latest developments, as mentioned in the ESIA by MEECO and Royal HaskoningDHV. But we agree with the Reviewer, it is better to show a projection in line with other climate studies, framed within the IPCC AR5. We therefore adapted the time-frame in the figure to the baseline projection of 20 years (as by given by the IPCC, 2013 pg. 19; Stocker et al., 2013 pg. 79, Collins et al., 2013 pg. 1031; and Church et al., 2013 pg. 1148). Fig. 5f now shows averaged wave data from 1986-2005 (baseline in the aforementioned IPCC) and compares them with the 2081-2100 period equivalent. Thank you for pointing this out and helping us to produce a more standardized and comparable result!

Initiated by the Reviewer's critical suggestion, we realized that the boxplots (subfigure 5f) show data from all ensembles, however the historical data comes from the reanalysis data (as shown in subfigures 5a-e) and not from the historical ensemble data - and thus not from a similar model setup. We changed this and now the historical data (in Fig. 5f) also shows all data values from the historical ensembles, created under CMIP5.

This leaves us with another challenge, as the available data under the specific CMI-protocols are different. CAWCR uses the CMIP3 protocol for RCP 3.5 projections, while CMIP5 protocols were used for RCP4.5 and 8.5. We therefore dropped the RCP 3.5 following the argumentation of giving a better comparability among the ensembles.

In any way, we feel more comfortable showing only data from a consistent data source and are thankful that the Reviewer's comment allowed us to improve our figure.

New references:

IPCC, 2013: Summary for Policymakers. In: Climate Change 2013: The Physical Science Basis. Contribution of Working Group I to the Fifth

Assessment Report of the Intergovernmental Panel on Climate Change [Stocker, T.F., D. Qin, G.-K. Plattner, M. Tignor, S.K. Allen, J. Boschung, A. Nauels, Y. Xia, V. Bex and P.M. Midgley (eds.)]. Cambridge University Press, Cambridge, United Kingdom and New York, NY, USA.

Stocker, T.F., D. Qin, G.-K. Plattner, L.V. Alexander, S.K. Allen, N.L. Bindoff, F.-M. Bréon, J.A. Church, U. Cubasch, S. Emori, P. Forster, P. Friedlingstein, N. Gillett, J.M. Gregory, D.L. Hartmann, E. Jansen, B. Kirtman, R. Knutti, K. Krishna Kumar, P. Lemke, J. Marotzke, V. Masson-Delmotte, G.A. Meehl, I.I. Mokhov, S. Piao, V. Ramaswamy, D. Randall, M. Rhein, M. Rojas, C. Sabine, D. Shindell, L.D. Talley, D.G. Vaughan and S.-P. Xie, 2013: Technical Summary. In: Climate Change 2013: The Physical Science Basis. Contribution of Working Group I to the Fifth Assessment Report of the Intergovernmental Panel on Climate Change [Stocker, T.F., D. Qin, G.-K. Plattner, M. Tignor, S.K. Allen, J. Boschung, A. Nauels, Y. Xia, V. Bex and P.M. Midgley (eds.)]. Cambridge University Press, Cambridge, United Kingdom and New York, NY, USA.

Collins, M., R. Knutti, J. Arblaster, J.-L. Dufresne, T. Fichet, P. Friedlingstein, X. Gao, W.J. Gutowski, T. Johns, G. Krinner, M. Shongwe, C. Tebaldi, A.J. Weaver and M. Wehner, 2013: Long-term Climate Change: Projections, Commitments and Irreversibility. In: Climate Change 2013: The Physical Science Basis. Contribution of Working Group I to the Fifth Assessment Report of the Intergovernmental Panel on Climate Change [Stocker, T.F., D. Qin, G.-K. Plattner, M. Tignor, S.K. Allen, J. Boschung, A. Nauels, Y. Xia, V. Bex and P.M. Midgley (eds.)]. Cambridge University Press, Cambridge, United Kingdom and New York, NY, USA.

Church, J.A., P.U. Clark, A. Cazenave, J.M. Gregory, S. Jevrejeva, A. Levermann, M.A. Merrifield, G.A. Milne, R.S. Nerem, P.D. Nunn, A.J. Payne, W.T. Pfeffer, D. Stammer and A.S. Unnikrishnan, 2013: Sea Level Change. In: Climate Change 2013: The Physical Science Basis. Contribution of Working Group I to the Fifth Assessment Report of the Intergovernmental Panel on Climate Change [Stocker, T.F., D. Qin, G.-K. Plattner, M. Tignor, S.K. Allen, J. Boschung, A. Nauels, Y. Xia, V. Bex and P.M. Midgley (eds.)]. Cambridge University Press, Cambridge, United Kingdom and New York, NY, USA.

Actions taken for this comment:

- Changed to a more consistent data source for historical wave data in Figure 5f (not in 5a-e!).
- Dropped RCP 3.5 projection data for sake of consistency.

Comment 4.11:

Are you calculating the ensemble changes of H_s ? Why did you consider only the period 2010-2019 and 2090-2099? The climatology needs at least 20 or 30 years to be calculated. I think mention future changes in significant wave height does not significant contribute to the study."

Answer to Comment:

We do not calculate ensemble changes of H_s , but follow a different approach: Usually "by using the average of an ensemble of GCMs, the individual model errors are cancelled out and the ensemble uncertainty decreases as increasingly more models are used" (Sperna Weiland et al., 2012). But we use every available data point of each ensemble and plot these in the violin-type boxplot. This shows the entire data and its range, their distribution and all important statistical values, instead of only averages of all ensembles. This gives the reader an even wider and more robust insight into the database, instead of an ensemble, a set of ensemble data or giving changes in the ensembles.

For us, the manuscript does not scrutinize the methods behind and variability of projected wave data but of climate change and associated impacts. Therefore, the study refers to projected climate ocean pressures. To give a better understanding of these terms, we find it necessary to give the reader a more tangible idea, which dimension of increasing pressures and associated climate change impacts small reef islands will face in the future - and therefore combine all available data in one plot. We mention the projected sea level rise, but as the study shows, waves have also an impact on the island and its morphodynamics, so it is necessary to also show the projected change in H_s .

New references:

F.C. Sperna Weiland, L.P.H. van Beek, A.H. Weerts, M.F.P. Bierkens, 2012: Extracting information from an ensemble of GCMs to reliably assess future

global runoff change, Journal of Hydrology, Volumes 412–413, Pages 66-75, ISSN 0022-1694, <https://doi.org/10.1016/j.jhydrol.2011.03.047>.

Action taken for this comment:

- None.

Comment 4.12:

“Although the simulations you have performed help to understand the effect of the seaport interrupting the sediment transport generated by the wave currents, you have only analyzed the transport in two characteristic storms. Storms usually have a high effect in the shoreline retreat, but the shoreline can be recovered between storms being seasonal (as it seems to happen in the north of the island) and interannual changes natural oscillations of the shoreline and they do not necessary imply erosion. It is true that reef environments require special analysis due to the complexity of the environment. There are many references that have been improved the knowledge of the sediment transport in the reef environment and atoll island (most of them have been included in your references) which apply more sophisticated methods. I think your analysis is only for contrast the measured physical processes and the local knowledge about the effect of the seaport but it is not in the State-of-the-art.”

Answer to Comment:

We understand the Reviewer’s concerns. The insights presented here are a close-up on the seaport, derived from the results in David and Schlurmann (2020) - a study which wasn’t published at the time of submission, but is now available in open access. This new paper shows both: sediment formation under storm conditions and over a longer time period.

So, in this sense, we cannot fully or adequately respond to the Reviewers concerns, stating our methods would not be *“sophisticated enough”* in comparison to other studies, because the Reviewer just states an opposing viewpoint without referring to any source(s) that could help illustrating this perspective or providing an in-depth proof-of-concept. In contrast to the Reviewer’s statement and to the best of our knowledge, we are not aware of any more sophisticated tool, being able to show sediment transport patterns over the entire reef. Surely, other numerical models like XBeach or Funwave contain modules accounting for

morphodynamics, but these are mostly tested for mainland coasts with finer sediment and usually only valid for storm conditions (and certainly not under calmer conditions). We do refrain from applying results from sediment modules and scrutinize the driver for sediment transport being (wave-induced) currents instead. As described in the manuscript, areas with increased velocities will initiate transport and areas with decreased velocities will lead to sediment accumulation. This is also the basic assumption within these sediment modules of the aforementioned numerical models (see fundamental literature of Shields, 1936; Hjulstrom, 1935; Egiazaroff, 1965 or Leo v. Rijn, compiled in van Rijn, 2005. The latter serves as basics for sediment modules in a number of today's numerical models, dealing with intercoupled hydro- and morphodynamic interaction). This knowledge of processes is sufficient to qualitatively assess the situation. Such a procedure also avoids error sources when using sediment modules for quantitative observations in environments and for circumstances beyond their scope - which is clearly the case on reef islands.

The study David and Schlurmann (2020) show the methods used and insights gained were at least appropriate for the situation - even though "more sophisticated" methods might exist.

On the specific comment "*Storms usually have a high effect in the shoreline retreat, but the shoreline can be recovered between storms being seasonal*". This is in general true, but the case here is different. The wave data already hint at the very dominant sediment transport direction on the east coast is towards north, being confirmed by results in David and Schlurmann (2020). But with the seaport blocking the sediment transport from south, no material for recovery is present - no matter if the waves are due to storm waves or calm conditions!

New references:

Egiazaroff, I. V. (1965). Calculation of nonuniform sediment concentrations. Journal of Hydraulics Division, Proceedings of the American Society of Civil Engineers, v.91, p. 225-247

Hjulstrom, F. (1935). Studies of the morphological activity of rivers as illustrated by the River Fyris, Bulletin. Geological Institute Upsalsa, 25, 221-527.

van Rijn, L. C. (2005). Principles of Sedimentation and Erosion Engineering in Rivers, Estuaries and Coastal Seas. Amsterdam: Aqua Publications.

Shields, A. (1936) Application of similarity principles and turbulence research to bed-load movement. California Institute of Technology, Pasadena, CA.

Action taken for this comment:

- The new version of the manuscript now officially refers to David and Schlurmann (2020) and explains the significance of this publication for the current study in the according parts.

Annex comment 3.11

When do I need an ethics vote?

General

If it is planned to carry out studies on humans, identifiable human material or identifiable data, the opinion of the locally competent ethics committee is generally required. This general principle is further specified below for different areas of the social sciences, behavioral sciences and humanities according to the state of discussion reached in the subjects.

I. Notes for social sciences (especially sociology, political science, economics, social and cultural anthropology, educational science and related subjects)

Hints for psychology see below II, hints for projects abroad see below III.

For the social sciences (and projects from neighboring subjects that work with social science methods), the submission of **an ethics vote is generally required if patients are involved in the study**. In the following cases, a **statement is expected in the application**, and in certain **circumstances** a vote of ethics may be required:

- Persons involved in the study are persons for whom there is a special need for protection ("vulnerable groups"), such as persons with limited ability to give consent.
- The study and the material used in it are suitable for triggering strong emotions, severe psychological stress or traumatic experiences that go beyond everyday experiences in the participants (interviewees, informants, project collaborators, researchers and those being researched).
- The investigation implies physical risks for the participants or leads to physical pain.
- Potential participants should not be informed about the examination.
- Potential participants should not be informed about the possible risks of participation and measures to avoid harm.

- Participation in the study implies deception (e.g. in laboratory experiments).
- The investigation exposes the participants (interviewees, informants, project staff, researchers) to special risks (e.g. social risks, risks of criminal or civil liability, financial losses, professional disadvantages or damage to reputation; risk due to difficult security situation in the investigation room).

If you are not sure whether your project requires the opinion of an ethics committee, please contact the responsible department.

[... subsection II. is for psychology]

III. Notes for projects carried out abroad

For projects financed by the DFG and partly or completely carried out abroad, the legal regulations applicable in the respective country must be observed. In some cases, the involvement of ethics committees in the respective target country or partner country may also be required.

An existing statement by a foreign ethics committee does not release the DFG from the obligation to examine whether a statement by a German ethics committee should also be obtained. If necessary, an additional opinion of an ethics committee locally responsible in Germany must be obtained. However, the ethics committee may adopt the opinion of the foreign ethics committee.

Do I have to submit the ethics committee's opinion at the time of application?

As a rule, the ethics vote must be submitted with the application. In exceptional cases, you should submit it as soon as possible to avoid delays. Your application can only be processed and decided upon after the evaluation if an ethics committee has given its positive opinion.

Translated with www.DeepL.com/Translator (free version)

Reviewer comments, second round –

Reviewer #3 (Remarks to the Author):

The authors have done a very good revision from their earlier version and they responded well to my comments.

Reviewer #4 (Remarks to the Author):

Thank you for the detail responses.

The manuscript has improved due to the new information added and the restructuring of the paper. I think most of this new information was missed which was basic to provide support to the purpose of your paper.

From my point of view, the main contributions are the new section about the coastal adaptation in practice and the wider description of the methods regarding societal aspects. They help me to understand better your work. However, I still find a bit difficult to capture your main message of your study. There are two concepts (maldevelopment and climate change-maladaptation) that I do not find perfectly linked in your manuscript:

1) The identification of the root causes of the erosion problem in Fuvahmulah, that from your study is deduced that it is not climate change by it is the maldevelopment (in this case, the construction of the seaport in a conflict area)

2) The adaptation to climate change, especially in the small islands, and in this study, and how conventional engineering solutions could be a maladaptation option.

I understand that maladaptation are the solutions proposed and selected to solve the problem of erosion in the island, which are described in the section "Coastal adaptation in practice", and which considers also the effect of climate change (mainly, sea level rise). It is not clear for me if all these solutions are proposed only to solve the problem of erosion adjacent area to Fuvahmulah's seaport or adaptation solutions are also proposed to stop the expected erosion due to climate change. In this latter case, I wonder if the report includes an evaluation of the shoreline/erosion in the future due to sea level rise, deterioration of the coral reefs or other driver changes (e.g., waves). If they are solutions to solve climate change impacts, I understand that they could be considered maladaptation because they are mainly hard-engineering solutions and, additional conditions regarding the social-political dimensions (top-down developments, and a lacking consideration of people's interests) could contribute even more to exacerbate the negative effect. However, in principle, these reasons are assumptions.

On the other hand, I see the example of erosion in the area around the port as an example of how hard coastal structures have negative effects in the environment (maldevelopment) and this kind of solution should be avoided because it might also increase erosion if they are implemented as adaptation solution. Besides, there is also social aspects that contribute also to a higher impact in the environment (as not considering the local knowledge) and that the reason why you considered it as "a socio-political phenomenon amplifying maladaptation". Although, I consider that the political dimension (top-down processes) is extracted from the analysis of the "coastal adaptation in practice".

Are you trying to transmit this idea with this sentence (in the introduction)? Could it be considered a conclusion from your study?

"In that context, this study investigates maladaptive developments and explicates their

consequences to facilitate developing adequate and sustainable adaptation strategies – especially in the face of increasing risks due to climate change”

Regarding the structure of the introduction, you start talking about the impact of climate change in the small islands and that anthropogenic interventions are usually limited to engineering-type hard-coastal protection. Then, enumerate the diverse options to tackle climate change based on the last report of the IPCC. Therefore, we can consider that you are going to focus on the adaptation strategies usually implemented in small islands and with all the additional limitations in their implementations (in terms of social-political characteristics).

But afterwards, you move directly to infrastructure demands and development decisions which it is not necessarily related with climate change. It is true that anthropogenic interventions in natural dynamics put further pressures on the environment and are driven by political decision-making processes and social interests. Although these decisions are embedded in political structures on multiple levels of governance, could we consider that Sea port is “maladaptative” per definition? Is this a categorical statement regarding small islands? I am not sure if you use maldevelopment/maladaptative indiscriminately. It introduces misunderstanding for me.

For example, in this sentence in the conclusions: “The example of the reef island Fuvahmulah epitomizes local coastal maladaptation in the Maldives, implemented by the national government in a top-down process”.

The example of the reef island Fuvahmulah is referring to the seaport? The solutions to solve the problem of erosion (described in section “Coastal adaptation in practice”)? Or solutions in general to reduce the effect of climate change due to an expected future erosion? Should it be described as “maldevelopment” instead of “maladaptation”?

Regarding this paragraph in the introduction: “Besides the maladaptive impact on nature, maldevelopment emphasizes two socio-political aspects: On the one hand, the societal factors, being the local population’s perception of and their attachment to place, their interest in economic development, and environmental protection. On the other hand, the political aspect of responsible authorities, aiming to balance the need for development with environmental protection and issues of sustainability based on decisions and policies. In this sense, maldevelopment is characterized by decisions or policies which are in constant or deliberate favor of inadequate actions and trade-offs towards future climate change related risk.”

Could it be considered conclusions from your study?

In the discussion, “However, instead of governing this conflict (Bisaro & Hinkel, 2016), future plans are going to respond to erosion associated with the seaport’s construction with further generic main-streamed adaptation measures (Hoogduin, 2016; Saleem, 2016) rather than acknowledging and dealing with the root causes of erosion”.

Could we consider that the root causes of erosion are the seaport itself? In this case, how do you think it would be the best option to solve the problem? Do you consider that they are not taking into account the root causes because they are proposing “generic main-streamed adaptation measures”? Could we consider that these measures to solve the problem of erosion caused by the seaport are “adaptation measures”?

In the discussion, “Structural maldevelopment to date impairs the potential of dealing with further future changes, such as sea level rise, extreme wave events, and storm surges. Studying maldevelopment in its comprehensiveness leads to the conclusion that sustainable development requires an integrated analysis of political interests and societal demands within the natural boundaries, in order to adequately address future climate change stressors”.

My conclusion from this sentence is not related with climate change adaptation, it refers only to the design of coastal structures including the effect of climate change. I mentioned it in the first review (comment 4.7), any coastal structure design should include an evaluation of its environmental impact and the effect of climate change. However, I think this approach is relatively new (maybe it is an issue that it is starting to include in the design of the coastal infrastructures in the last decade) and not yet standardized.

More specific comments:

The reference Dodet et al. 2019 (The Contribution of Wind-Generated Waves to Coastal Sea-Level Changes) is not related with the impact of climate change in waves. It is about the sea level component generated by waves when they approximate to the coast (swash and run-up components), that they are usually added to storm surge (meteorological sea level) and mean sea level (with seasonal fluctuations).

In the same reference where you extracted "Using the average of an ensemble of GCMs, the individual model errors are cancelled out and the ensemble uncertainty decreases as increasingly more models ", they said that "Although often the non-weighted ensemble mean hydrological change is calculated, results of several studies showed that more reliable results are obtained by using projections of a cluster of better performing models (Smith and Chandler, 2010) or calculating a weighted ensemble average, where the individual GCM weights are derived from model performance and future ensemble convergence". This is the reason why proposed a new method to obtain a weighted ensemble. Therefore, I think it is more correct to calculate changes (future-historical) for each ensemble and then, calculate the multi-model ensemble of changes (you can represent the mean value but also the uncertainty). References related with wave climate change performed that (see for example, latest references as Morim et al., 2019). For the historical values, it better to use the hindcst(or reanalysis).

Reviewer #5 (Remarks to the Author):

This paper uses an interdisciplinary approach that combines physical and social sciences to investigate erosion on the island of Fvahmulah, Maldives. The study makes some important conclusions that would be of interest to a wide audience, such as the need for management to consider local circumstances and to consider natural processes. A strength of the manuscript is its interdisciplinary approach – such studies are very limited in this field. Physical sciences sections of the manuscript their accompanying analyses and very good figures were a particular strength of the study. However, I would recommend that major revisions would be needed to this manuscript before its acceptance. In particular:

1. Further explanation of why this particular island was selected would be useful when the site is introduced in line 34 and in the study site section. It would be useful to highlight the broader significance of this site – beyond this one island.
2. It would be useful to further explain / justify why the authors believe there to be maldevelopment in the Maldives. It wasn't entirely clear to me how much this was fact or opinion. For example, line 91 – 'Fuvahmulah is an epitome of maldevelopment on small islands'- why?
3. I would have liked to have seen a more thorough presentation of the 'social science' results, ideally accompanied by figures and tables in the results section. This would, to my mind, create a more balanced interdisciplinary piece. For example, it is mentioned that 36% of responses to an open question mentioned erosion. I'm then wondering what the other responses mentioned – are other issues mentioned more frequently, or are you highlighting erosion as it fits in with the physical science analyses in this paper?
4. The discussion would be strengthened with additional links back to the results section – this would clarify for the reader that this discussion has indeed arisen from your analysis.
5. The conclusion is very long, but much of this material could be relocated into the discussion to strengthen that section (as above).
6. The writing style is not quite at the level that would be expected from a Nature journal. This does not impact on the science, but if accepted, I'd recommend it undergoes some proof reading. A capital letter should not follow every colon.

Specific comments:

Paragraph 1 – This paragraph ought to present a more balanced perspective - consider the large body of shoreline change literature that highlights the potential resilience of reef islands (e.g. Kench's papers). Swell events may also serve to build reef islands (e.g. East et al., 2018 – in the Maldives).

Line 38 – though proximity to the equator means that storms will be rare on this island – worth mentioning. Swell events are likely more important.

Line 39 – I'd like to see more evidence that the top-down approach is used. This is not my area of expertise, but I'd like to see more evidence - what is the role then of local councils in such decisions?

Line 138 and elsewhere - transection = transect

Line 256 – A more detailed description of the role of local councils would be useful in this section to strengthen your narrative.

Line 397 – 'Concluding from past experiences and findings of this study, these measures fail to address processes of the reef's sediment transport and thus drivers behind erosion faithfully.'
This sentence would benefit from further explanation with a clearer link back to your results.

Response to Referees Letter

Manuscript Number: NCOMMS-20-22976B

Title: Climate change induced effects or maldevelopment: small islands and conflicting attribution of root causes

Dear respected reviewers,

We would like to express our appreciation for your constructive comments and we believe that addressing these comments have again improved the quality of the manuscript significantly.

Please find enclosed the detailed answers to the reviewers' comments as well as the corresponding actions addressed in the revised manuscript. The responses below start with our replies to the recent comments (2nd revisions), which are followed by the initial revisions of the 1st review round.

Two revised manuscripts are submitted along with this rebuttal letter - one version allowing the reviewers to track changes and a clean version of the current manuscript.

Sincerely,

The authors.

Table of Contents

Table of Contents	2
Editor's response	4
Answer to the editor's response	5
Response to Referees Letter	1
Table of Contents	2
Reviewer #4:	4
Comment 4a.1:	4
Answer to Comment:	4
Action taken for Comment:	4
Comment 4a.2:	5
Answer to Comment:	5
References:	6
Action taken for Comment:	7
Comment 4a.3:	8
Answer to Comment:	8
Action taken for Comment:	10
Comment 4a.3:	11
Answer to Comment:	11
Action taken for Comment:	11
Comment 4a.4:	12
Answer to Comment:	12
References:	12
Action taken for Comment:	12
Comment 4a.5:	14
Answer to Comment:	14
Action taken for Comment:	14
Comment 4a.6:	15
Answer to Comment:	15
References:	15
Action taken for Comment:	15
Reviewer #5 (New reviewer):	16
Comment 5.1:	16
Answer to Comment:	16
Action taken for Comment:	16
Comment 5.2:	17
Answer to Comment:	17
Actions taken for Comment:	18
Comment 5.3:	20
Answer to Comment:	20
Action taken for Comment:	20
Comment 5.4:	21

Answer to Comment:	21
References:	21
Action taken for Comment:	21
Comment 5.5:	22
Answer to Comment:	22
Action taken for Comment:	23
Comment 5.6:	24
Answer to Comment:	24
Action taken for Comment:	24
Comment 5.7:	25
Answer to Comment:	25
Action taken for Comment:	25
Comment 5.8:	26
Answer to Comment:	26
Action taken for Comment:	26

Reviewer #4:

Comment 4a.1:

Thank you for the detail responses.

The manuscript has improved due to the new information added and the restructuring of the paper. I think most of this new information was missed which was basic to provide support to the purpose of your paper.

Answer to Comment:

We thank the reviewer for helping us in these improvements. With the reviewer's second set of comments, we further streamlined the definition of maladaptation and maldevelopment which is indeed prone to misinterpretation in the very beginning. With these changes and the comments of the new reviewer 5, we are convinced, to have strengthened the clarity of definitions given in the manuscript further and provide sound and interdisciplinary insights on the pitfalls of climate change adaptation on small reef islands and the potential of local knowledge to overcome maldevelopment, which are readily appealing to a wide audience.

Action taken for Comment:

None

Comment 4a.2:

From my point of view, the main contributions are the new section about the coastal adaptation in practice and the wider description of the methods regarding societal aspects. They help me to understand better your work. However, I still find a bit difficult to capture your main message of your study. There are two concepts (maldevelopment and climate change-maladaptation) that I do not find perfectly linked in your manuscript:

- 1) The identification of the root causes of the erosion problem in Fuvahmulah, that from your study is deduced that it is not climate change by it is the maldevelopment (in this case, the construction of the seaport in a conflict area)
- 2) The adaptation to climate change, especially in the small islands, and in this study, and how conventional engineering solutions could be a maladaptation option.

I understand that maladaptation are the solutions proposed and selected to solve the problem of erosion in the island, which are described in the section “Coastal adaptation in practice”, and which considers also the effect of climate change (mainly, sea level rise). It is not clear for me if all these solutions are proposed only to solve the problem of erosion adjacent area to Fuvahmulah’s seaport or adaptation solutions are also proposed to stop the expected erosion due to climate change. In this latter case, I wonder if the report includes an evaluation of the shoreline/erosion in the future due to sea level rise, deterioration of the coral reefs or other driver changes (e.g., waves). If they are solutions to solve climate change impacts, I understand that they could be considered maladaptation because they are mainly hard-engineering solutions and, additional conditions regarding the social-political dimensions (top-down developments, and a lacking consideration of people’s interests) could contribute even more to exacerbate the negative effect. However, in principle, these reasons are assumptions.

Answer to Comment:

In our manuscript, we wrote: *“This project is defined as “the development, implementation and maintenance of sustainable coastal protection to prevent erosion and flooding on [...] Fuvahmulah. The main objective of this project is to decrease erosion and flooding through a possible [...] combination of hard and soft coastal engineering interventions [...] to protect the island of Fuvahmulah against flooding due to ongoing coastal erosion and rising sea levels”*”. This is the official statement taken from the project’s original objective proposed to the funding agency (development assistance), showing that the intended measures are meant to address both the current erosion and future climate change effects.

To understand, if a solution can be considered maladaptive, one has to go back to the IPCC’s definition of maladaptation: maladaptation is an action or inaction, that *“may lead to increased risk of adverse climate-related outcomes, increased vulnerability to climate change, or diminished welfare, now or in the future”* (IPCC, 2017: WGII AR5, Glossary, page 1769). In the revised manuscript, we added the following sentences for clarification: *“Maladaptation aims at reducing adverse effects to the coastal community now, but has negative implications on future response options under different climate change scenarios (Noble et al., 2015). For example, the widespread anthropogenic coastal fortification in the Maldives has lead to*

irreversible changes to the natural coastal system, so that the only remaining future response to rising sea levels and extreme events is further armoring of the reef islands' coasts (see also Duvat and Magnan, 2019)." It is not relevant if the response options are hard-engineered (that doesn't make them maladaptive). It is more important to look at their impact on future options to respond to climate change effects and associated hazards (for further elucidation on the implications of maladaptation on coastal management and risk assessment see additions in the revised manuscript). One main issue of maladaptation is that it has a negative influence on future adaptation pathways. In this sense, armouring the island's coast would eliminate future adaptation options, for example physically, financially, or both, for example by means of complementary nature-based solutions. While hard-protection measures can be a feasible option in urban environments (Nunn et al., 2021), maladaptation "*may also occur if the true potential of [any response] option or [...] technology is unduly over-emphasized, making it over-rated*" (Noble et al., 2015). This is the case in the Maldives, as island fortification is widely adopted and renders alternative or complementary options impossible described by Duvat and Magnan (2019).

In the case of Fuvahmulah, there is already a problem with erosion and it needs to be solved in due time, because with rising sea levels, the erosion will impair the island's natural green belt and protective rim. But in their assessment, the national government and the associated international consultant coastal engineers attempt solving the current erosion problem with a rubble mound revetment (fortification) on the island's east side. Construction material (e.g. granite rock or basalt) for the revetment is absent in the Maldives and needs to be imported from neighboring countries, but following tedious logistics and costly overseas transport. But from experience on other islands (Kench 2016; Duvat and Magnan, 2019, ...), these hard-engineering measures might alleviate the symptoms on site, but will very likely just postpone the erosion and in addition reduce the island's natural capacity to grow and naturally adapt with sea level rise. If the erosion is postponed and future adaptation is again to fortify the island's perimeter (this is likely under the current Maldivian adaptation pathway and their preferences for hard-engineered armouring), then long sections of Fuvahmulah's coast are going to be completely fortified - which has been a problem on other Maldivian islands, where "*anthropogenic tipping points*" have already been reached (Duvat and Magnan, 2019). In this context, the specific amount of erosion is not the focus of and rather irrelevant for the present study. Our focus is rather (a) to analyze the underlying mechanisms leading to coastal erosion, so that helpful approaches could be applied and (b) to reveal the socio-political dimension leading to this repeated false (or mal-)adaptation. An "*unduly over-emphasis*" on hard protection is a symptom of maladaptation, roots in the socio-political history of the Maldives, and is still prevalent today (as outlined by our analysis).

This shows that our assessment is not assumptious, but inclusive as it considers a broader range of factors, contributing to or even aggravating maladaptation on reef islands. Therefore, our key message is that the original concept of maladaptation does not go far enough and with our combined interdisciplinary analysis, we introduce the concept of maldevelopment.

References:

Duvat, V.K.E., Magnan, A.K. Rapid human-driven undermining of atoll island capacity to adjust to ocean climate-related pressures. *Sci Rep* 9, 15129 (2019). <https://doi.org/10.1038/s41598-019-51468-3>

Lempert, R. J. (2019). Robust Decision Making (RDM). In: Marchau, V. A. W. J., Walker, W. E., Bloemen, P. J. T. M., and Popper, S. W., editors, Decision Making under Deep Uncertainty: From Theory to Practice, pages 23–51. Springer International Publishing, Cham. https://doi.org/10.1007/978-3-030-05252-2_2

Action taken for Comment:

Added the following explanation to the introduction:

“Maladaptation aims at reducing adverse effects to the coastal community now, but has negative implications on future response options under different climate change scenarios (Noble et al., 2015). For example, the widespread anthropogenic coastal fortification in the Maldives has led to irreversible changes to the natural coastal system, so that the only remaining future response to rising sea levels and extreme events is further armoring of the reef islands’ coasts (Duvat and Magnan, 2019).”

Comment 4a.3:

On the other hand, I see the example of erosion in the area around the port as an example of how hard coastal structures have negative effects in the environment (maldevelopment) and this kind of solution should be avoided because it might also increase erosion if they are implemented as adaptation solution. Besides, there is also social aspects that contribute also to a higher impact in the environment (as not considering the local knowledge) and that the reason why you considered it as “a socio-political phenomenon amplifying maladaptation”. Although, I consider that the political dimension (top-down processes) is extracted from the analysis of the “coastal adaptation in practice”.

Are you trying to transmit this idea with this sentence (in the introduction)? Could it be considered a conclusion from your study?

“In that context, this study investigates maladaptive developments and explicates their consequences to facilitate developing adequate and sustainable adaptation strategies – especially in the face of increasing risks due to climate change”

Regarding the structure of the introduction, you start talking about the impact of climate change in the small islands and that anthropogenic interventions are usually limited to engineering-type hard-coastal protection. Then, enumerate the diverse options to tackle climate change based on the last report of the IPCC. Therefore, we can consider that you are going to focus on the adaptation strategies usually implemented in small islands and with all the additional limitations in their implementations (in terms of social-political characteristics).

But afterwards, you move directly to infrastructure demands and development decisions which it is not necessarily related with climate change. It is true that anthropogenic interventions in natural dynamics put further pressures on the environment and are driven by political decision-making processes and social interests. Although these decisions are embedded in political structures on multiple levels of governance, could we consider that Sea port is “maladaptive” per definition? Is this a categorical statement regarding small islands? I am not sure if you use maldevelopment/maladaptive indiscriminately. It introduces misunderstanding for me.

For example, in this sentence in the conclusions: “The example of the reef island Fuvahmulah epitomizes local coastal maladaptation in the Maldives, implemented by the national government in a top-down process”.

The example of the reef island Fuvahmulah is referring to the seaport? The solutions to solve the problem of erosion (described in section “Coastal adaptation in practice”)? Or solutions in general to reduce the effect of climate change due to an expected future erosion?

Should it be described as “maldevelopment” instead of “maladaptation”?

Answer to Comment:

The reviewer writes *“On the other hand, I see the example of erosion in the area around the port as an example of how hard coastal structures have negative effects in the environment (maldevelopment) and this kind of solution should be avoided because it might also increase*

erosion if they are implemented as adaptation solution”. Maldevelopment is not the negative effects of coastal adaptation on the environment, but the socio-political circumstances leading to repeated maladaptive actions. This statement was already taken in the former manuscripts in lines 53-61:

“In contrast to maladaptation, maldevelopment is not an inadequate adaptation action leading to climate change related risks, but a socio-political phenomenon amplifying maladaptation. [...] In this sense, maldevelopment is characterized by decisions or policies which are in constant or deliberate favor of inadequate actions and trade-offs towards future climate change related risk.”

Fuvahmulah’s island dwellers require(d) this port for giving shelter to small boats and vessels and enable them to improve their local economic situation by services granted by port infrastructure. Within the interplay and well-being of society, economy and environment, it improved (only) two dimensions at the expense of the third. However, the port triggered maldevelopment when analyzing how the negative consequences on the environment are being dealt with..

We strengthened the definition to:

“While recent studies found deficient coastal adaptation policies and policy compliance in the Maldives (Zubair, 2011; Gussmann and Hinkel 2021), this study scrutinizes the impact of such deficits in practice and elucidates the background of recurrent and ongoing maladaptation, which is structurally embedded within the socio-political system. In contrast to maladaptation, this study defines maldevelopment, which is not an inadequate adaptation action leading to climate change related risks, but the socio-political driver behind repeated maladaptation. Besides the maladaptive impact on the natural system, maldevelopment emphasizes the socio-political aspects of recurrent maladaptive actions”.

and further elucidate in the discussions:

“such recurring and systematically provoked maladaptive actions go beyond the initial scope of maladaptation. Against this background, this study defines the concept of maldevelopment, addressing the socio-political framing, provoking repeated maladaptive actions.”

as well as (see bold and underlined section):

*“Together with today’s impact of the seaport, future modifications to the natural reef system will most likely have even more significant adverse effects for greater parts of the coast – especially in spatio-temporal dimensions of the “Coastal Protection at Gn. Fuvahmulah, Maldives”. **This study has shown that in the case of Fuvahmulah, there are distinct hot-spots, which are important for the sediment supply of the island’s coast. The seaport is located at one of these hotspots, disturbing the natural sediment transport around the island (Figure 7). The seaport construction was arguably a trade-off for economic development and societal well-being at the expense of natural conservation. Nevertheless it has triggered maldevelopment on Fuvahmulah, when assessing how the negative consequences on the environment are being dealt with:** instead of governing this conflict (Bisaro and Hinkel, 2016), future plans are going to respond to erosion associated with the seaport construction with further generic main-streamed adaptation measures (Hoogduin, 2016; Saleem, 2016) rather than acknowledging and compensating for the root causes of erosion (also being recognized by Fuvahmulah’s island dwellers).”*

In line with the above comments of this reviewer and suggestions of reviewer 5, we also adjusted the first paragraph in the introduction and supported reframing maladaptation and maldevelopment within the last paragraph of the introduction.

Action taken for Comment:

- Improved explanation of maldevelopment and maladaptation with a more straightforward definition and addressing the distinction of both concepts.
 - Revised and strengthened all according sections in the introduction
- (please see track changes document)

Comment 4a.3:

Regarding this paragraph in the introduction: “Besides the maladaptive impact on nature, maldevelopment emphasizes two socio-political aspects: On the one hand, the societal factors, being the local population’s perception of and their attachment to place, their interest in economic development, and environmental protection. On the other hand, the political aspect of responsible authorities, aiming to balance the need for development with environmental protection and issues of sustainability based on decisions and policies. In this sense, maldevelopment is characterized by decisions or policies which are in constant or deliberate favor of inadequate actions and trade-offs towards future climate change related risk.”

Could it be considered conclusions from your study?

Answer to Comment:

In our study, we define the term “maldevelopment” - to our knowledge - for the first time in the context of climate change science. This definition is based on the interdisciplinary approach of this study and findings stemming from this approach. In this sense, the definition, of course, already builds on the insights we gained throughout the study. Still, we aim to give a comprehensive definition for readers encountering this concept for the first time and in our opinion this requires to elucidate the socio-political factors emphasized by maldevelopment.

Action taken for Comment:

None

Comment 4a.4:

In the discussion, “However, instead of governing this conflict (Bisaro & Hinkel, 2016), future plans are going to respond to erosion associated with the seaport’s construction with further

generic main-streamed adaptation measures (Hoogduin, 2016; Saleem, 2016) rather than acknowledging and dealing with the root causes of erosion”.

Could we consider that the root causes of erosion are the seaport itself? In this case, how do you think it would be the best option to solve the problem? Do you consider that they are not taking into account the root causes because they are proposing “generic main-streamed adaptation measures”? Could we consider that these measures to solve the problem of erosion caused by the seaport are “adaptation measures”?

Answer to Comment:

Yes, the reviewer is right: the root-cause of erosion is the seaport. Drivers and processes that led to this finding have been discussed in David et al. (2019, 2020). Adaptation is in general defined as a “*processes of adjustment by natural or human systems to actual or expected climate and its effects, intended to moderate harm or exploit beneficial opportunities.*” (Abram et al., 2019). So as long as the intention behind the structural measure is to solve erosion, they are adaptation measures. In our opinion, the best and - resulting from our assessment - most sustainable solution would be to regularly bypass sediment by means of common dredging technologies, adapted to the local circumstances, which is believed to activate or even restore the natural dynamics in the reef island and common practice in coastal engineering. We added this to the discussion section.

References:

Abram, N., Gattuso, J.-P., Prakash, A., Cheng, L., Chidichimo, M., Crate, S., Enomoto, H., Garschagen, M., Gruber, N., Harper, S., Holland, E., Kudela, R., Rice, J., Steffen, K., and von Schuckmann, K. (2019). Framing and context of the report. In Pörtner, H.-O., Roberts, D., Masson-Delmotte, V., Zhai, P., Tignor, M., Poloczanska, E., Mintenbeck, K., Alegría, A., Nicolai, M., Okem, A., Petzold, J., Rama, B., and Weyer, N., editors, IPCC Special Report on the Ocean and Cryosphere in a Changing Climate, page 71–130. In Press. (link)

David, G., Schlurmann, T., & Roeber, V. (2019). Coastal Infrastructure on Reef Islands – the Port of Fuvahmulah, the Maldives as Example of Maladaptation to Sea-Level Rise? https://doi.org/10.18451/978-3-939230-64-9_087

David, C. G., & Schlurmann, T. (2020). Hydrodynamic Drivers and Morphological Responses on Small Coral Islands—The Thoнду Spit on Fuvahmulah, the Maldives. *Frontiers in Marine Science*, 7. <https://doi.org/10.3389/fmars.2020.538675>

Action taken for Comment:

Revised the discussion section.

Incorporating the topographic and reef measurements as well as the wave climate in numerical wave models helps to scrutinize the root cause behind the measured erosion on Fuvahmulah’s

east coast: The seaport intervenes with the natural sediment transport and acts as barrier, deflecting suspended sediment off the reef (Figure 7). Bypassing this barrier would be a suitable remedy resurrecting the natural sediment transport along the east side. A bypass would nourish the beaches and act as "low-regret" adaptation to the anthropogenic disturbance, because it limits the impact on future response options under different climate change scenarios. But instead of providing a more robust and flexible approach towards coastal adaptation, the current adaptation plan is in favor of fortifying the entire coastline -- a wide-spread approach in the Maldives and common pitfall, which is known to undermine the islands' natural capacity to adjust to ocean climate-related pressures (Magnan and Duvat, 2020; Nunn et al., 2021) and a crucial step towards reaching an "anthropogenic tipping point" (Duvat and Magnan, 2019).

Comment 4a.5:

In the discussion, “Structural maldevelopment to date impairs the potential of dealing with further future changes, such as sea level rise, extreme wave events, and storm surges. Studying maldevelopment in its comprehensiveness leads to the conclusion that sustainable development requires an integrated analysis of political interests and societal demands within the natural boundaries, in order to adequately address future climate change stressors”.

My conclusion from this sentence is not related with climate change adaptation, it refers only to the design of coastal structures including the effect of climate change. I mentioned it in the first review (comment 4.7), any coastal structure design should include an evaluation of its environmental impact and the effect of climate change. However, I think this approach is relatively new (maybe it is an issue that it is starting to include in the design of the coastal infrastructures in the last decade) and not yet standardized.

Answer to Comment:

We thank the reviewer for sharing the meaningful thoughts on environmental impact assessments which are indeed not yet standardized on global scale; especially in developing countries. But, these EIAs are in fact mandatory since 2012 on the Maldives as an emerging economy, as we have written in the manuscript:

“Since 2012, the EIA Regulation of the Maldives requires to assess the (adverse) impact of such infrastructure projects on the environment”.

Action taken for Comment:

None

Comment 4a.6:

More specific comments:

The reference Dodet et al. 2019 (The Contribution of Wind-Generated Waves to Coastal Sea-Level Changes) is not related with the impact of climate change in waves. It is about the sea level component generated by waves when they approximate to the coast (swash and run-up components), that they are usually added to storm surge (meteorological sea level) and mean sea level (with seasonal fluctuations).

In the same reference where you extracted “Using the average of an ensemble of GCMs, the individual model errors are cancelled out and the ensemble uncertainty decreases as increasingly more models “, they said that “Although often the non-weighted ensemble mean hydrological change is calculated, results of several studies showed that more reliable results are obtained by using projections of a cluster of better performing models (Smith and Chandler, 2010) or calculating a weighted ensemble average, where the individual GCM weights are derived from model performance and future ensemble convergence”. This is the reason why proposed a new method to obtain a weighted ensemble. Therefore, I think it is more correct to calculate changes (future-historical) for each ensemble and then, calculate the multi-model ensemble of changes (you can represent the mean value but also the uncertainty). References related with wave climate change performed that (see for example, latest references as Morim et al., 2019). For the historical values, it better to use the hindcast(or reanalysis).

Answer to Comment:

We have been looking through our references once more, but haven't found a study by Dodet et al. (2019) in our reference list. The only study of Dodet we refer to is the paper by Timmermans et al. (2020). However, we thank the reviewer for commenting again on this methodological approach and giving us a broader perspective on this matter. We are happy to consider these approaches in future analyses.

References:

Timmermans, B. W., Gommenginger, C. P., Dodet, G. & Bidlot, J.-R. Global wave height trends and variability from new multi mission satellite altimeter products, reanalyses, and wave buoys. *Geophys. Res. Lett.* 47,e2019GL086880, DOI: <https://doi.org/10.1029/2019GL086880> (2020).

Action taken for Comment:

None

Reviewer #5 (New reviewer):

Comment 5.1:

This paper uses an interdisciplinary approach that combines physical and social sciences to investigate erosion on the island of Fvahmulah, Maldives. The study makes some important conclusions that would be of interest to a wide audience, such as the need for management to consider local circumstances and to consider natural processes. A strength of the manuscript is its interdisciplinary approach – such studies are very limited in this field. Physical sciences sections of the manuscript their accompanying analyses and very good figures were a particular strength of the study.

Answer to Comment:

We are happy that the reviewer was able to summarize our study's main topic and finds our insights to be appealing to a wide audience.

Action taken for Comment:

None

Comment 5.2:

However, I would recommend that major revisions would be needed to this manuscript before its acceptance. In particular:

1. Further explanation of why this particular island was selected would be useful when the site is introduced in line 34 and in the study site section. It would be useful to highlight the broader significance of this site – beyond this one island.
2. It would be useful to further explain / justify why the authors believe there to be maldevelopment in the Maldives. It wasn't entirely clear to me how much this was fact or opinion. For example, line 91 – 'Fuvahmulah is an epitome of maldevelopment on small islands' – why?
3. I would have liked to have seen a more thorough presentation of the 'social science' results, ideally accompanied by figures and tables in the results section. This would, to my mind, create a more balanced interdisciplinary piece. For example, it is mentioned that 36% of responses to an open question mentioned erosion. I'm then wondering what the other responses mentioned – are other issues mentioned more frequently, or are you highlighting erosion as it fits in with the physical science analyses in this paper?
4. The discussion would be strengthened with additional links back to the results section – this would clarify for the reader that this discussion has indeed arisen from your analysis.
5. The conclusion is very long, but much of this material could be relocated into the discussion to strengthen that section (as above).
6. The writing style is not quite at the level that would be expected from a Nature journal. This does not impact on the science, but if accepted, I'd recommend it undergoes some proof reading. A capital letter should not follow every colon.

Answer to Comment:

1. Motivated by the reviewer's valuable comments, we adjusted the explanations of maldevelopment and maladaptation with more straightforward definitions and by addressing the distinction of both concepts which is indeed prone to misinterpretation. This slight reframing strengthens the according sections in the introduction and better clarifies both concepts and their distinct features.
2. In line with the previous issue under point 1., we adapted the last section of the introduction to clarify the issue raised by the reviewer here.
3. We understand the reviewer's valid concern, but already considered to raise the visibility of the social science section in an earlier stage of the revision phase, where we stated:
"However, despite being an interdisciplinary approach, we are not striving to achieve a 50:50 distribution between natural and social sciences in the manuscript, but to address the research interest raised by the project and manuscript."
and
"..., it was not our intent to achieve a 50:50 distribution between both disciplines – but rather to fully address and manage a complex research topic reflected in this manuscript and use the social science perspective to fill knowledge gaps from the physical modelling perspective."

In the previous stages of the reviewing process we already commented on this remark, but now we decided to add another figure, displaying the relevant questions and answers from the questionnaire (Figure 8). This increases the visibility of the social sciences and thus adding to the overall balance and to progress the transdisciplinary nature of our research approach.

Further in-depth information and results from the empirics have been compiled for the Supplementary material and made available to the reader. In the comment, the reviewer cites a result from the study from the manuscript. About 10 lines before this citation in the manuscript, we write: “*In the household survey, the local community perceives erosion as the most pressing issue (closed question, 27% of 345 mentions, see supplementary materials)*”. Here, we redirect the reader to the supplementary materials for further insights and additional information. However, a further link to the supplementary materials seems to be required, so that we added another “(see supplementary materials)” after the sentence the reviewer quoted.

4. This concern was also raised later in Comment 5.8, where the reviewer directly refers to a sentence in the section. The specific reply is fully dedicated in Comment 5.8. However, this comment here also encouraged us to revise the entire discussion section and add another reference to our results (bold and underlined section “actions taken for comment” below), so that we are thankful to the reviewer for pointing this out.
5. We acknowledged the concern and incorporated the conclusion section into the “Synthesis & Discussion” section – it is also in line with the editor’s request to incorporate the conclusion section into the discussion sections.
6. We thank the reviewer for outlining these opportunities to improve. We have tried our best with the expertise of our university’s language department as non-native speakers. But we are happy to discuss these matters upon acceptance of our manuscript with the editor or proof reading office at Springer Nature again.

Actions taken for Comment:

1. Reworked the paragraph in lines 49–79, introducing the concept of maldevelopment.
2. Added a more direct reference to the definition of maldevelopment in lines 49–79.
3. Added Figure 8 and references to the supplementary materials
4. Together with today’s impact of the seaport, future modifications to the natural reef system will most likely have even more significant adverse effects for greater parts of the coast -- especially in spatio-temporal dimensions of the “*Coastal Protection at Gn. Fuvahmulah, Maldives*”. The seaport construction was arguably a trade-off for economic development and societal well-being at the expense of natural conservation. This study has shown that in the case of Fuvahmulah, there are distinct hot-spots, which are important for the sediment supply of the island's coast. The seaport is located at one of these hotspots, disturbing the natural

sediment transport around the island (Figure 7). However, instead of governing this conflict (Bisaro et al., 2016), future plans are going to respond to erosion associated with the seaport's construction with further generic main-streamed adaptation measures (Hoogduin, 2016; MEECO 2016) rather than acknowledging and dealing with the root causes of erosion (also being recognized by Fuvahmulah's island dwellers). Concluding from other studies' insights (Duvat and Magnan, 2019; Magnan and Duvat, 2020; Nunn et al., 2021) and findings from this study, these measures fail to address processes of the reef's sediment transport and thus drivers behind erosion adequately. Incorporating the topographic and reef measurements as well as the wave climate in numerical wave models helps to scrutinize the root cause behind the measured erosion on Fuvahmulah's east coast: †the seaport intervenes with the natural sediment transport and acts as barrier, deflecting suspended sediment off the reef (Figure 7). Bypassing this barrier is a suitable remedy resurrecting the natural sediment transport along the east side. A bypass would nourish the beaches and act as "low-regret" adaptation to the anthropogenic disturbance, because it limits the impact on future response options under different climate change scenarios. But instead of providing a more robust and flexible approach towards coastal adaptation, the current adaptation plan is in favor of fortifying the entire coastline -- a wide-spread approach in the Maldives and common pitfall, which is known to undermine the islands' natural capacity to adjust to ocean climate-related pressures and a crucial step towards reaching an "anthropogenic tipping point" (Duvat and Magnan, 2019).

5. Added "(see supplementary materials)" at the end of the sentence quoted by the reviewer.
6. Changed capitalization after colons.

Comment 5.3:

Specific comments:

Paragraph 1 – This paragraph ought to present a more balanced perspective – consider the large body of shoreline change literature that highlights the potential resilience of reef islands (e.g. Kench’s papers). Swell events may also serve to build reef islands (e.g. East et al., 2018 – in the Maldives).

Answer to Comment:

We enhanced the literature review on the resilience of reef islands in accordance with the reviewer’s comments.

Action taken for Comment:

Coastal environments react to hydrodynamic pressures with a morphodynamic response, such as wave-induced currents on the reef platform being a driver of sediment transport. Therefore, waves, sea level, reef platform, and sediment production are major elements of reef island genesis and evolution (Kench et al 2006a; Kench et al. 2006b, Kench and Mann 2017; David and Schlurmann, 2020): For example, when looking at the Maldives, monsoonal wind-wave events have an impact on locally characteristic island morphology, but, in general, distant-source swell waves are dominant in the regional wave climate (Kench et al 2006b; Wadey et al. 2017; David and Schlurmann 2020). Swell waves are characterized by a long wave period and play a key role in the island development on reef platforms (East et al. 2018, Masselink et al. 2020). Together with constant sediment provision by the coral reef (Ryan et al. 2019), the wave-induced sediment transport allows reef islands to mitigate erosion (Tuck et al. 2021) and accrete vertically in response to sea level rise (East et al. 2018; Masselink et al. 2020). Coral reefs are important coastal protection assets (Beck et al., 2018), because the reef dissipates most of the incoming wave energy (Ferrario et al., 2014). There is evidence that reefs have grown and can continuously provide protection under rising sea levels (Beetham et al., 2017; East et al., 2018) if they remain healthy. As a consequence, healthy coral reefs are a very important – if not the most important – factor of the island’s resilience to withstand sea level rise and marine extreme events. However, the health of coral reefs is threatened by global warming and ocean acidification (Hughes et al., 2017) and with that also their important ecosystem services.

Comment 5.4:

Line 38 – though proximity to the equator means that storms will be rare on this island – worth mentioning. Swell events are likely more important.

Answer to Comment:

Yes, the reviewer is right that equatorial areas are less prone to be harmed by cyclones and storm activity (there are nice data sets of NASA, depicting the cyclone activity away from the equator towards the mid-latitudes). However, our own impressions of the Monsoon seasons as well as the data taken and processed on site compared with our experiences of the wind-wave-dominated wave climate in the North and Baltic Sea still make us hesitant to put too large emphasis on this. In our previous publication, we show that swell waves from Southern directions are dominant most of the time, but wind-waves govern local events and control transport of sediment. This is in line with other publications by Kench's group and Wadey et al. (2017). Still, we acknowledge the reviewer's comment and adjust the results section accordingly, where we further describe the local wave climate.

References:

Wadey, M., Brown, S., Nicholls, R. J. & Haigh, I. Coastal flooding in the Maldives: an assessment of historic events and their implications. *Nat. Hazards* 89, 131–159, DOI: 10.1007/s11069-017-2957-5 (2017).

Action taken for Comment:

Changed manuscript in “Introduction” section:

Therefore, waves, sea level, reef platform, and sediment production are major elements of reef island genesis and evolution (Kench et al 2006a; Kench et al. 2006b, Kench and Mann 2017; David and Schlurmann, 2020): *For example, when looking at the Maldives, monsoonal wind-wave events have an impact on locally characteristic island morphology, but, in general, distant-source swell waves are dominant in the regional wave climate (Kench et al 2006b; Wadey et al. 2017; David and Schlurmann 2020). Swell waves are characterized by a long wave period and play a key role in the island development on reef platforms (East et al. 2018, Masselink et al. 2020). Together with constant sediment provision by the coral reef (Ryan et al. 2019), the wave-induced sediment transport allows reef islands to mitigate erosion (Tuck et al. 2021) and ...*

Changed manuscript in “Study Site” section:

The isolated location and with the reef and vegetated coastal ridges serve as the only natural coastal protection, Fuvahmulah is particularly susceptible to environmental forcing. *Fuvahmulah's location close to the equator makes it less likely to experience cyclones (Trigo and Gimeno, 2009; Wadey et al., 2017) but the island is considered to be highly exposed to monsoon winds (MEE, 2014} and associated wind-waves, as well as from distant-source swells (Kench et al., 2006b, Wadey et al. 2017; David and Schlurmann, 2020; Magnan and Duvat, 2020).*

Comment 5.5:

Line 39 – I'd like to see more evidence that the top-down approach is used. This is not my area of expertise, but I'd like to see more evidence - what is the role then of local councils in such decisions?

Answer to Comment:

This is actually further described in the section “National politic perspective on coastal infrastructure” and a genuine result of this study that rather renders it unique in coastal research in accordance to “Sea change in coastal science”(an editorial by Nature Communications):

For example, when regarding the implementation of coastal development projects in the Maldives, the planning, implementation and decision-making process are centrally executed and ministerially anchored in the national government without significant involvement of local capacities on the islands. This stands in contrast to the republic's decentralization efforts and the corresponding strengthening of local communities. In the Maldives, actors involved in coastal governance report coastal infrastructure projects, such as seaports as well as coastal protection structures, are predominantly implemented in a top-down process. Such infrastructure projects are usually supported by different international organizations and projects, providing required external financial and knowledge resources.

In addition, the section “Coastal adaptation in practice” describes how decision-makers on a national level undermine a participatory approach of coastal adaptation with an inadequate ESIA. So as a result, together with the findings in Ratter et al. (2019), our study already provides sufficient evidence of a top-down implementation.

The reviewer already acknowledges the reference to the study of Ratter et al. (2019). This study was the complementary social-science study in the transdisciplinary project the research was set in. One of the main objectives of the study by Ratter et al. (2019) was to analyze the government structure with respect to climate change adaptation. The centralized perspective of the Maldives and the historic background was also described by Magnan and Duvat (2020), which we also cited throughout the manuscript. For more evidence and to acknowledge the reviewer's comment, we now also refer to Duvat (2020), who further elucidated this.

More insights into the evolution of top-down policy making in the Maldives is also given by Gussmann and Hinkel (2020). This is further evidence to our statement in the sense of the reviewers request. In contrast to the relocation focus in Gussmann and Hinkel (2020), our case focuses more on the infrastructure aspect. We refrain from citing this study here.

References:

Duvat, V. K. Human-driven atoll island expansion in the Maldives. *Anthropocene* 32, 100265, DOI: <https://doi.org/10.1016/j.ancene.2020.100265> (2020).

Gussmann, G. & Hinkel, J. What drives relocation policies in the Maldives? *Clim. Chang.*163, 931–951, DOI:<https://doi.org/10.1007/s10584-020-02919-8> (2020)

Ratter, B., Hennig, A. & Zahid. Challenges for shared responsibility – Political and social framing of coastal protection transformation in the Maldives. *DIE ERDE - Journal of the Geographical Society of Berlin* 150,169–183 (2019).

Sea change in coastal science (2020). *Nature Communications*, 11(1). DOI: <https://doi.org/10.1038/s41467-020-18333-8>

Action taken for Comment:

Added references to the referred sentence:

“The Maldives government is forced to address these challenges and tends to top-down implemented infrastructure projects (Ratter et al., 2019; **Duvat, 2020; Magnan and Duvat, 2020**), which ...”

Changed manuscript section to:

“**This is confirmed by interviewees** involved in coastal governance, reporting coastal infrastructure projects ~~, such as seaports as well as coastal protection structures,~~ are predominantly implemented in a top-down process.”

Added reference:

Duvat, V.K.E. (2020). Human-driven atoll island expansion in the Maldives. doi.org/10.1016/j.ancene.2020.100265

Comment 5.6:

Line 138 and elsewhere - transection = transect

Answer to Comment:

We thank the reviewer for pointing this out.

Action taken for Comment:

Changed accordingly throughout the document

Comment 5.7:

Line 256 – A more detailed description of the role of local councils would be useful in this section to strengthen your narrative.

Answer to Comment:

The reviewer is right. Such information further strengthens the result section. We changed the former section “*National politics perspective on coastal infrastructure*” accordingly (Please be aware that we needed to reduce the header levels, as the journal’s formatting instructions do not allow for secondary headers. This (sub-)section, as well as the (sub-)section “*Local society's experience with coastal processes*” is now merged under the higher level section title “*Governmental and societal framing of coastal development*”).

Action taken for Comment:

Added and adopted text in the results section:

... in the capital Male’. **Decisions are made at the highest level and have an impact down to the local scale.** When regarding the implementation of coastal development projects in the Maldives, the planning, implementation and decision-making process are centrally executed and ministerially anchored in the national government without significant involvement of local capacities on the islands. **According to actors involved in coastal governance, in general, such top-down processes in decision and implementation are applied for coastal infrastructure projects, for example seaports as well as coastal protection structures. Regional and local government institutions, such as the city council on Fuvahmulah, lack influencing power in the decision-making process. According to the interviewees, the role of local government institutions is limited to informing national-level actors about coastal problems on their island. The lack of power is also expressed through a lack of financial resources for coastal projects at a council-level (interview with a representative of a state environmental agency in 2017). In addition, infrastructure projects are usually supported by international organizations, providing required external financial and knowledge resources. International organizations, however, are legally bound to use the national government as an entry point and cannot initiate projects below the central national level. Altogether, this stands in contrast to the republic’s decentralization efforts and the corresponding strengthening of local communities.**

Comment 5.8:

Line 397 – ‘Concluding from past experiences and findings of this study, these measures fail to address processes of the reef’s sediment transport and thus drivers behind erosion faithfully.’ This sentence would benefit from further explanation with a clearer link back to your results.

Answer to Comment:

This issue was also brought up in a previous comment (Comment 5.2.4). We thank the reviewer for specifying the location in our manuscript and added a distinct link back to our results.

Action taken for Comment:

Together with today’s impact of the seaport, future modifications to the natural reef system will most likely have even more significant adverse effects for greater parts of the coast -- especially in spatio-temporal dimensions of the “*Coastal Protection at Gn. Fuvahmulah, Maldives*”. The seaport construction was arguably a trade-off for economic development and societal well-being at the expense of natural conservation. This study has shown that in the case of Fuvahmulah, there are distinct hot-spots, which are important for the sediment supply of the island's coast. The seaport is located at one of these hotspots, disturbing the natural sediment transport around the island (Figure 7). However, instead of governing this conflict (Bisaro et al., 2016), future plans are going to respond to erosion associated with the seaport’s construction with further generic main-streamed adaptation measures (Hoogduin, 2016; MEECO 2016) rather than acknowledging and dealing with the root causes of erosion (also being recognized by Fuvahmulah's island dwellers). Concluding from other studies’ insights (Duvat and Magnan, 2019; Magnan and Duvat, 2020; Nunn et al., 2021) and findings from this study, these measures fail to address processes of the reef's sediment transport and thus drivers behind erosion faithfully. Incorporating the topographic and reef measurements as well as the wave climate in numerical wave models helps to scrutinize the root cause behind the measured erosion on Fuvahmulah’s east coast: the seaport intervenes with the natural sediment transport and acts as barrier, deflecting suspended sediment off the reef (Figure 7). Bypassing this barrier is a suitable remedy resurrecting the natural sediment transport along the east side. A sediment bypass would nourish the beaches and act as “low-regret” adaptation to the anthropogenic disturbance, because it limits the impact on future response options under different climate change scenarios. But instead of providing a more robust and flexible approach towards coastal adaptation, the current adaptation plan is in favor of fortifying the entire coastline -- a wide-spread approach in the Maldives and common pitfall, which is known to undermine the islands' natural capacity to adjust to ocean climate-related pressures and a crucial step towards reaching an “anthropogenic tipping point” (Duvat and Magnan, 2019).

Reviewer comments, third round –

Reviewer #4 (Remarks to the Author):

Thank you for your explanations.

Most of my comments were related with the definitions of maladaptation and maldevelopment and they have been solved.

Reviewer #5 (Remarks to the Author):

The paper is certainly stronger than the previous version and the authors have responded well to the points raised. I have several minor remaining comments:

- The introduction is much stronger for the additional information, but it is now very long. I wonder whether the information could be condensed to be slightly more punchy for the reader.
- There are still several instances where it's unclear to me whether you are stating fact or opinion.

For example:

- o 'As a consequence, the resulting planning efforts and constructive responses are mostly based on external design guidelines and building codes. In general, experiences from the past are mainstreamed into international design guidelines and building codes, giving best-practice examples under certain environmental conditions – in case of coastal structures these conditions are those of mainland coasts.' References would be useful to support these claims.

- o 'Decisions are made at the highest level and have an impact down to the local scale. When regarding the implementation of coastal development projects in the Maldives, the planning, implementation and decision-making process are centrally executed and ministerially anchored in the national government without significant involvement of local capacities on the islands. According to actors involved in coastal governance, in general, such top-down processes in decision and implementation are applied for coastal infrastructure projects, for example seaports as well as coastal protection structures. Regional and local government institutions, such as the city council on Fuvahmulah, lack influencing power in the decision-making process.' This quote is from the section on interviews and so I assume that such information has come from the interviews? If so, could you substantiate this with quotes and/ or data? If not, can you use references to support these claims?

- o 'Fuvahmulah has gained more attention by the national government in the last two decades, manifested through numerous development projects in that time, thus erosion – being a high priority issue on the island – is a state affair nowadays.' – I'm unsure what you mean by a state affair and am unclear if this is your opinion?

- o 'The example of Fuvahmulah epitomizes planning inadequate coastal structures in a reef environment underlies special environmental features and requirements beyond conventional, generic infrastructure design rules and solutions.' - could you refer back to some data to support this claim?

- o 'many small islands in the Maldives and in the world show symptoms of maldevelopment.' References would be useful to support this claim.

- Figure 8 – it is good to see the data from the population survey. It is unclear to me why the x axis on panel d is reversed. Also, on panel d, is that the exact wording of your question, or are there some missing words?

- I am a little surprised that neither of the existing sediment transport studies undertaken in the Maldives have been referenced. Please see: <https://doi.org/10.1016/j.geomorph.2014.02.013>
<https://doi.org/10.1016/j.gloplacha.2020.103196>

- If formatting rules permit, a short conclusion section would be useful.

- I applaud the authors for their writing (I cannot imagine writing a paper in anything other than

my first language), but at times the writing style is still not quite at the level that would be expected from a Nature journal. This should not count against the authors and I would hope that Nature may be able to offer some support to ensure that all of the wording and grammar is as clear as possible.

Response to Referees Letter

Manuscript Number: NCOMMS-20-22976B

Title: Climate change induced effects or maldevelopment - small islands and conflicting attribution of root causes

Dear respected reviewers,

We would like to express our appreciation for your constructive comments and we believe that addressing these comments have again improved the quality of the manuscript significantly.

Please find enclosed the detailed answers to the reviewers' comments as well as the corresponding actions addressed in the revised manuscript. The responses below start with our replies to the recent comments (3rd revisions), which are followed by the initial revisions of the 1st and subsequent 2nd review round.

Two revised manuscripts are submitted along with this rebuttal letter - one version allowing the reviewers to track changes and a clean version of the current manuscript.

Sincerely,

The authors.

Table of Contents

Response to Referees Letter	1
Table of Contents	2
Reviewer #5 (New reviewer):	3
Comment 5a.1:	3
Answer to Comment:	3
Action taken for Comment:	3
Comment 5a.2:	4
Answer to Comment:	4
Action taken for Comment:	4
Comment 5a.2a:	5
Answer to Comment:	5
Action taken for Comment:	5
Comment 5a.2b:	6
Answer to Comment:	6
Action taken for Comment:	6
Comment 5a.2c:	7
Answer to Comment:	7
Action taken for Comment:	7
Comment 5a.2d:	8
Answer to Comment:	8
Action taken for Comment:	8
Comment 5a.2e:	9
Answer to Comment:	9
Action taken for Comment:	9
Comment 5a.3:	10
Answer to Comment:	10
Action taken for Comment:	10
Comment 5a.4:	11
Answer to Comment:	11
Action taken for Comment:	11
Comment 5a.5:	12
Answer to Comment:	12
Action taken for Comment:	12
Comment 5a.6:	13
Answer to Comment:	13
Action taken for Comment:	13
Appendix	14

Reviewer #5 (New reviewer):

Comment 5a.1:

The paper is certainly stronger than the previous version and the authors have responded well to the points raised. I have several minor remaining comments:

The introduction is much stronger for the additional information, but it is now very long. I wonder whether the information could be condensed to be slightly more punchy for the reader.

Answer to Comment:

We understand that the introduction is rather long (as is the paper itself). However, this is due to several reasons:

1. We are introducing and discussing a new aspect to climate change adaptation, which is the concept of “*maldevelopment*”. To give the reader a clear understanding of this concept, it needs to be well explained and set into context of its cognate “*maladaptation*” to avoid any misinterpretation. The definition of “*maldevelopment*” - in contrast to “*maladaptation*” - was the pivotal discussion with reviewer 4. We therefore do not see any chances to condense this section of the introduction, without impairing the valuable input of reviewer 4, which helped significantly to improve the quality of our manuscript.
2. As an interdisciplinary study, published in Nature Communication, we aim at making our research accessible and communicating our findings as well as its implications for a wide audience with a diverse background. While working together as an interdisciplinary team in this study, both disciplines have learned to - at times - avoid shorter descriptions of certain aspects, which are probably more “punchy” for experts in one discipline, but less tangible for readers coming from another scientific community. Creating this common ground for everyone is a key to understanding the study, its structure and its implications beyond the dimension of only one discipline.

Even though we share the reviewer’s perspective and would endorse a shorter introduction, we feel this is not possible without compromising the two mentioned points and therefore refrain from making changes here.

Action taken for Comment:

None

Comment 5a.2:

There are still several instances where it's unclear to me whether you are stating fact or opinion. For example: ...

Answer to Comment:

Before answering the following comments by the reviewer, one general remark: As outlined in point 2 in our reply to Comment 5a.1, we present our study and results with descriptive elements - on the one hand, to allow a wide range of readers to follow the interdisciplinary study, on the other hand because of the methodological approach developed and applied by the social scientists in our team. They have been doing parts of their field work with qualitative surveys. For example, the Deakin University, Australia, describes qualitative questionnaires (as part of qualitative study design) as

"... open-ended questions to produce long-form written/typed answers. Questions will aim to reveal opinions, experiences, narratives or accounts. Often a useful precursor to interviews or focus groups as they help identify initial themes or issues to then explore further in the research (link to the text)."'

As such, there are several text sections that are not an opinion, but facts, which we describe from our observation. We understand that the reviewer would like to have seen such observations backed up by "hard" measured data or based on quantitative analysis as a common basis in natural or engineering sciences, but here, we make use of a qualitative research approach, as common practice in social sciences:

"Whereas quantitative research aims to develop objective theories by generating quantifiable numerical data, qualitative research aims to understand meaning. This might be the meanings that people attribute to their work, their behaviours or beliefs, or their attitudes or perceptions. Qualitative research is often based on methods of observation and enquiry; qualitative research "explores the meaning of human experiences and creates the possibilities of change through raised awareness and purposeful action" (Taylor & Francis, 2013²). Qualitative research focuses on life experiences; they are more about the "why" and "how" rather than the "how many", or "how often" (link to the text³)."'

It is this combination of approaches - the quantitative natural or engineering sciences with the qualitative approaches in social sciences - which gives unique insights into climate change adaptation on small islands and gives us the confidence to define the concept of "maldevelopment" that is the centre of manuscript. A few of the following issues raised in this respect refer in general to this mixed approach.

Action taken for Comment:

None

¹ <https://deakin.libguides.com/qualitative-study-designs/surveys>

² doi.org/10.4324/9780203777176

³ <https://deakin.libguides.com/qualitative-study-designs/about>

Comment 5a.2a:

'As a consequence, the resulting planning efforts and constructive responses are mostly based on external design guidelines and building codes. In general, experiences from the past are mainstreamed into international design guidelines and building codes, giving best-practice examples under certain environmental conditions – in case of coastal structures these conditions are those of mainland coasts.' References would be useful to support these claims.

Answer to Comment:

Following the above stated methodological approaches, these findings are not “*claims*”, but a description on how building codes and design guidelines are being practically implemented and how the lack of technical capacities and expertise leads to the acquisition and provision of external knowledge. This is chiefly executed by international consulting companies in the Maldives.

For example, the Maldives do not have a dedicated university, teaching technical subjects like coastal engineering. We observed the impact of lacking technical engineering expertise also in other situations in the country, for example how the local government was handling tender procedures of, for example, road repairs. Of course, if technical expertise is missing, building codes are required. Therefore, there is a need to use, refer to and employ external solutions or consultancy.

And again, this is not an opinion, but an original observation we – as an interdisciplinary team – have made on the island, also facilitated through the contact with local actors and national authorities. To set our results in context and explain our findings, we are describing and sharing these circumstances here.

Action taken for Comment:

Added reference to USACE’s Coastal Engineering Manual and CERC’s Shore Protection Manual:

*'As a consequence, the resulting planning efforts and constructive responses are mostly based on external design guidelines and building codes (CERC 1984, USACE 2002). In general, experiences from the past are mainstreamed into international design guidelines and building codes, giving best-practice examples under certain environmental conditions – in case of coastal structures these conditions are **predominantly** those of mainland coasts.'*

Comment 5a.2b:

'Decisions are made at the highest level and have an impact down to the local scale. When regarding the implementation of coastal development projects in the Maldives, the planning, implementation and decision-making process are centrally executed and ministerially anchored in the national government without significant involvement of local capacities on the islands. According to actors involved in coastal governance, in general, such top-down processes in decision and implementation are applied for coastal infrastructure projects, for example seaports as well as coastal protection structures. Regional and local government institutions, such as the city council of Fuvahmulah, lack influencing power in the decision-making process.' This quote is from the section on interviews and so I assume that such information has come from the interviews? If so, could you substantiate this with quotes and/ or data? If not, can you use references to support these claims?

Answer to Comment:

This, again, is not a claim, but a description of the political system based on local experiences and perceptions of the governance in the Maldives, following our original evaluation and analysis of the situation through our observations as well as the interviews. For a social scientist, describing such insights from a qualitative survey is as straightforward and meaningful as classifying Fuvahmulah as a reef island for a natural or engineering scientist - which is obviously given following the described characteristics of the geology, geomorphology and environmental settings.

Action taken for Comment:

None

Comment 5a.2c:

'Fuvahmulah has gained more attention by the national government in the last two decades, manifested through numerous development projects in that time, thus erosion – being a high priority issue on the island – is a state affair nowadays.' – I'm unsure what you mean by a state affair and am unclear if this is your opinion?

Answer to Comment:

We apologize for the misperception, but this again is not an opinion, but an obvious conclusion regarding the policies and programs enacted by the central government in the context of the study. As outlined in other studies (see for example research by Virginie Duvat et al.⁴ or Geronimo Gussmann and Jochen Hinkel⁵), and described in the results section the Maldives try to develop regional centers (following the Maldives Decentralization Act in 2010⁶). Fuvahmulah benefits from these developments noticeably, as there have been several infrastructure projects in the past two decades until today (airport, seaport, water supply and water treatment system and waste disposal service). This shows that Fuvahmulah is in the focus of the national government's politics. In the same way, the huge effort made to deal with erosion (external consultants and a coastal development project with a projected cost of almost 20 million Euro - see homepage of Rijksdienst voor Ondernemend Nederland or Netherlands Enterprise Agency, or screenshot in Appendix of this document) substantiates this sentence. It is not an opinion, it's an observed fact.

However, we understand that the term “state affair” might be confusing or gives room for misinterpretation and changed the term accordingly.

Action taken for Comment:

'Fuvahmulah has gained more attention by the national government in the last two decades, manifested through numerous development projects in that time, thus erosion – being a high priority issue on the island – is a matter dealt with on the highest governmental level nowadays.'

⁴for example <https://doi.org/10.1007/s10113-020-01691-w>

⁵for example <https://doi.org/10.1007/s10584-020-02919-8>

⁶further discussed in Gussmann and Hinkel (2020, <https://doi.org/10.1007/s10584-020-02919-8>)

Comment 5a.2d:

'The example of Fuvahmulah epitomizes planning inadequate coastal structures in a reef environment underlies special environmental features and requirements beyond conventional, generic infrastructure design rules and solutions.' – could you refer back to some data to support this claim?

Answer to Comment:

Yes, the reviewer is right - this comment is valuable and adapt the manuscript accordingly.

Action taken for Comment:

The example of Fuvahmulah epitomizes planning inadequate coastal structures in a reef environment underlies special environmental features and requirements beyond conventional, generic infrastructure design rules and solutions (**see the results section “Natural morphodynamics on the reef and anthropogenic interventions” and Figure 7**).

Comment 5a.2e:

'many small islands in the Maldives and in the world show symptoms of maldevelopment.'
References would be useful to support this claim.

Answer to Comment:

Yes, the reviewer is right - we added references to this sentence where effects of (coastal) maldevelopment took place and aggravated the situation by intervening into natural processes without understanding the physical drivers and processes.

Action taken for Comment:

... many small islands in the Maldives and in the world show symptoms of maldevelopment (Ratter et al., 2016; Petzold et al., 2018; Petzold and Magnan, 2019; Duvat and Magnan, 2019).

Comment 5a.3:

Figure 8 – it is good to see the data from the population survey. It is unclear to me why the x axis on panel d is reversed. Also, on panel d, is that the exact wording of your question, or are there some missing words?

Answer to Comment:

We changed the figure according to the reviewer's suggestions and provided the initial question in full.

Action taken for Comment:

Changed figure design and wording.

Comment 5a.4:

I am a little surprised that neither of the existing sediment transport studies undertaken in the Maldives have been referenced. Please see:

<https://doi.org/10.1016/j.geomorph.2014.02.013>

<https://doi.org/10.1016/j.gloplacha.2020.103196>

Answer to Comment:

Before submitting, the manuscript had a far more extensive list of references, including both references mentioned by the reviewer. However, the journal's "Brief guide for submission to Nature Communications" writes: "as a guideline, Articles allow up to 70 references.", which we complied to by removing a set of references before submitting the manuscript. Following reviewer 5's comments in the second round of revisions, we already added another set of references, exceeding the 70 references threshold (in consultation and in accordance with the editor).

In this sense, if we cited all the references we would have liked to refer to, the list would have been beyond 100 studies and papers. With this being said, we acknowledge the study of East et al. (2020) is worth mentioning and a valuable contribution for setting the scene in the introduction.

Action taken for Comment:

Included a reference to East et al. (2020) in the introduction.

Comment 5a.5:

If formatting rules permit, a short conclusion section would be useful.

Answer to Comment:

This point has been discussed in the previous round of reviews, where the editor stated: *“Please note regarding reviewer #5 request about the conclusion section, we do not permit a conclusion section in our style so therefore I recommend that you do follow the suggestion of incorporating this into the discussion to strengthen this.”*

and where we accordingly replied to the reviewer, upon the request to shift parts of the conclusion into the discussion following the rules and recommendations of the journal:

“We acknowledged the concern and incorporated the conclusion section into the “Synthesis & Discussion” section - it is also in line with the editor’s request to incorporate the conclusion section into the discussion sections.”

Therefore, we will continue to follow the editor’s advice to not include a “Conclusion” section.

Action taken for Comment:

None

Comment 5a.6:

I applaud the authors for their writing (I cannot imagine writing a paper in anything other than my first language), but at times the writing style is still not quite at the level that would be expected from a Nature journal. This should not count against the authors and I would hope that Nature may be able to offer some support to ensure that all of the wording and grammar is as clear as possible.

Answer to Comment:

As we have replied to the reviewer before, *“we thank the reviewer for outlining the opportunities to improve. We have tried our best with the expertise of our university's language department as non-native speakers. But we are happy to discuss these matters upon acceptance of our manuscript with the editor or proof reading office at Springer Nature again.”* If the editor advises us to use any proof-reading service (e.g. by Springer Nature), we will consider using it.

Action taken for Comment:

None

Appendix

Screenshot of rvo.nl⁷, taken on July 19th, 2021.

⁷ <https://www.rvo.nl/subsidies-regelingen/projecten/coastal-protection-gn-fuvahmulah>